# Spatial profiling of chromatin accessibility in mouse and human tissues

Yanxiang Deng[1,2], Marek Bartosovic[3], Sai Ma[4], Di Zhang[1], Petra Kukanja[3], Yang Xiao[5], Graham Su[1,2], Yang Liu[1,2], Xiaoyu Qin[1,2], Gorazd B. Rosoklija[6,7,8], Andrew J. Dwork[6,7,8,9], J. John Mann[6,7,10], Mina L. Xu[11], Stephanie Halene[2,12,13], Joseph E. Craft[14], Kam W. Leong[5,15], Maura Boldrini[6,7], Gonçalo Castelo-Branco[3,16 ✉] & Rong Fan[1,2,11,17 ✉]

Cellular function in tissue is dependent on the local environment, requiring new methods for spatial mapping of biomolecules and cells in the tissue context[1]. The emergence of spatial transcriptomics has enabled genome-scale gene expression mapping[2–5], but the ability to capture spatial epigenetic information of tissue at the cellular level and genome scale is lacking. Here we describe a method for spatially resolved chromatin accessibility profiling of tissue sections using next-generation sequencing (spatial-ATAC-seq) by combining in situ Tn5 transposition chemistry[6] and microfluidic deterministic barcoding[5]. Profiling mouse embryos using spatial-ATAC-seq delineated tissue-region-specific epigenetic landscapes and identified gene regulators involved in the development of the central nervous system. Mapping the accessible genome in the mouse and human brain revealed the intricate arealization of brain regions. Applying spatial-ATAC-seq to tonsil tissue resolved the spatially distinct organization of immune cell types and states in lymphoid follicles and extrafollicular zones. This technology progresses spatial biology by enabling spatially resolved chromatin accessibility profiling to improve our understanding of cell identity, cell state and cell fate decision in relation to epigenetic underpinnings in development and disease.

Single-cell sequencing presents a tangible way to define cell types and states[7], but the tissue dissociation process leads to the loss of spatial context. Moreover, the method of isolation in single-cell technologies may preferentially select certain cell types or perturb cellular states as a result of the dissociation or other environmental stresses[8]. Spatial transcriptomics emerged to address these challenges and to transform how we delineate cellular functions and states in the native tissue environment[1–5]. To investigate the mechanisms underlying the spatial organization of different cell types and functions in the tissue context, it is highly desired to examine not only gene expression but also epigenetic underpinnings such as chromatin accessibility[9] in a spatially resolved manner. Spatial epigenetic mapping would help us to uncover the causative relationship that determines what drives tissue organization and function. To date, the ability to spatially map epigenetic states, such as chromatin accessibility, directly in a tissue section at the genome scale and cellular level is lacking.

Assay for transposase-accessible chromatin using sequencing (ATAC-seq) was developed and was further applied to single cells[9,10]. Imaging chromatin accessibility in fixed cells using fluorescence-labelled DNA oligomers assembled in Tn5 (ATACsee)[11] suggests that it is feasible to profile chromatin accessibility in situ. Microdissecting tissues from specific regions followed by scATAC-seq enables the profiling of accessible chromatin from a region of interest[12]. However, spatially resolved chromatin accessibility mapping over a tissue section at the cellular level has not been possible. Here we applied a spatial barcoding scheme to DNA oligomers that were inserted into the accessible genomic loci by Tn5 transposition to realize spatial-ATAC-seq: high-spatial-resolution genome-wide mapping of chromatin accessibility in tissue at the cellular level. The results from mouse embryos delineated the region-specific epigenetic landscapes and gene regulators involved in the development of the central nervous system (CNS). We also applied spatial epigenomics to human tissues, including tonsils and the hippocampus. Spatial-ATAC-seq revealed a spatially distinct organization of immune cell types and states in relation to lymphoid follicles and extrafollicular zones. This technology adds a new dimension to spatial biology by bringing spatial chromatin accessibility to the field and may offer a wide range of applications in the study of normal development and pathogenesis.

[1]Department of Biomedical Engineering, Yale University, New Haven, CT, USA. [2]Yale Stem Cell Center and Yale Cancer Center, Yale School of Medicine, New Haven, CT, USA. [3]Laboratory of Molecular Neurobiology, Department of Medical Biochemistry and Biophysics, Karolinska Institutet, Stockholm, Sweden. [4]Klarman Cell Observatory, Broad Institute of MIT and Harvard, Cambridge, MA, USA. [5]Department of Biomedical Engineering, Columbia University, New York, NY, USA. [6]Department of Psychiatry, Columbia University, New York, NY, USA. [7]Division of Molecular Imaging and Neuropathology, New York State Psychiatric Institute, New York, NY, USA. [8]Macedonian Academy of Sciences & Arts, Skopje, Republic of Macedonia. [9]Department of Pathology and Cell Biology, Columbia University, New York, NY, USA. [10]Department of Radiology, Columbia University, New York, NY, USA. [11]Department of Pathology, Yale University School of Medicine, New Haven, CT, USA. [12]Section of Hematology, Department of Internal Medicine, Yale University School of Medicine, New Haven, CT, USA. [13]Yale Center for RNA Science and Medicine, Yale University School of Medicine, New Haven, CT, USA. [14]Department of Immunobiology, Yale University School of Medicine, New Haven, CT, USA. [15]Department of Systems Biology, Columbia University Irving Medical Center, New York, NY, USA. [16]Ming Wai Lau Centre for Reparative Medicine, Stockholm node, Karolinska Institutet, Stockholm, Sweden. [17]Human and Translational Immunology Program, Yale School of Medicine, New Haven, CT, USA. ✉e-mail: goncalo.castelo-branco@ki.se; rong.fan@yale.edu

## Spatial-ATAC-seq design and workflow

The workflow for spatial-ATAC-seq is shown in Fig. 1a,b and Extended Data Fig. 1. Tn5 transposition was performed in a fixed tissue section and adapters containing a ligation linker were inserted into accessible genomic loci. Next, barcodes A (A1–A50) and B (B1–B50) with linkers were introduced using microchannels and were ligated to the 5′ end of the Tn5 oligo through successive rounds of ligation, resulting in distinct combinations. The tissue slides were imaged such that spatially barcoded accessible chromatin can be correlated with the tissue morphology. After forming a spatially barcoded tissue mosaic (2,500 tiles), reverse cross-linking was performed to release barcoded DNA fragments, which were amplified by PCR for library preparation. To evaluate the performance of in situ transposition and ligation, the cells stained with 4′,6-diamidino-2-phenylindole (DAPI) were fixed on a glass slide, followed by Tn5 transposition and ligation of a barcode A with fluorescein isothiocyanate (FITC). The resulting images revealed a strong overlap between the nucleus (blue) and FITC (green) signals, indicating the successful insertion of adapters into accessible chromatin loci with ligated barcode A in nuclei only (Fig. 1c).

As we proceeded to develop spatial-ATAC-seq, we went through several versions of chemistry to achieve a high yield and a high signal-to-noise ratio (Supplementary Fig. 1a). We then applied the optimized protocol to different tissue types and assessed the data quality (Fig. 1d–k) (spatial-ATAC-seq pixels may contain more than one nucleus, which is based on the tissue type and cell size; for the embryonic day 10 (E10) mouse embryo, on average, there were 1–2 cells per 10 μm pixel and 25 cells per 50 μm pixel[5]). In a 50 μm E13 mouse embryo experiment, aggregate profiles accurately reproduced the bulk measurement. Furthermore, signals around *Slc4a1* were specifically enriched in the liver region but not in the brain (Fig. 1d). In other experiments, we obtained a median of 36,303 (E11) and 100,786 (E13) unique fragments per pixel of which 15% (E11) and 14% (E13) of the fragments overlapped with transcription start site (TSS) regions, and 10% (E11) and 8% (E13) were in peaks. Moreover, the proportion of mitochondrial fragments was low for both E11 and E13 (1%). For the 20 μm experiment with mouse postnatal day 21 (P21) brain and human tonsil, we obtained a median of 7,647 (brain) and 14,939 (tonsil) unique fragments per pixel of which 18% of fragments fell within the TSS regions, and 24% (brain) and 14% (tonsil) were in peaks. The fraction of read pairs mapping to the mitochondria was 13% (brain) and 3% (tonsil), and the variability in the percentage of mitochondrial fragments may come from a different type of tissue (Fig. 1e–h; as a reference, 10× non-spatial scATAC-seq obtained a median of 17,321 unique fragments per cell, 23% TSS fragments and 0.4% mitochondrial reads). We also found that pixels that are not in the tissue had significantly fewer unique fragments compared with pixels in the tissue (Supplementary Fig. 1c,d). Moreover, the insert size distribution was consistent with the capture of nucleosomal and subnucleosomal fragments for all of the tissue types (Fig. 1i–k). We also performed correlation analysis between replicates, which showed high reproducibility ($r = 0.95$) (Supplementary Fig. 1b).

## Spatial mapping of the E13 mouse embryo

We next sought to identify cell types de novo from the E13 mouse embryo. Unsupervised clustering identified eight main clusters, that revealed distinct spatial patterns that agreed with the tissue histology (Fig. 2a–c and Extended Data Fig. 2). For example, cluster 1 represents the fetal liver and cluster 2 is specific to the spine region, including dorsal root ganglia (DRG) with chromatin accessibility for *Sox10* (Supplementary Fig. 2a,b,i,j). Cluster 3 to cluster 5 are associated with the peripheral nervous system and CNS. Cluster 6 includes several cell types that are present in the developing limbs and cluster 8 encompasses several developing internal organs. To benchmark spatial-ATAC-seq data, we projected the ENCODE organ-specific

ATAC-seq data onto our uniform manifold approximation and projection (UMAP) embedding[13]. In general, the cluster identification matched well with the bulk ATAC-seq projection (Extended Data Fig. 2d–g) and distinguished all of the major developing tissues and organs (some inconsistency between bulk data and spatial-ATAC-seq data is probably attributed to the pixels that may contain multiple cell types). We further examined cell-type-specific marker genes, identified as differential between clusters (false-discovery rate (FDR) < 0.05, $\log_2$-transformed fold change (FC) ≥ 0.25) and estimated the expression of these genes from our chromatin accessibility on the basis of the overall signal at a given locus[14] (Fig. 2c, Extended Data Fig. 2c,h and Supplementary Table 1). *Sptb*, which has a role in the stability of erythrocyte membranes[15], was activated extensively in the liver. *Syt8*—belonging to the synaptotagmin family, which is important in exocytosis during neurotransmission[16]—had a high level of gene activity in the spine and DRG regions. *Ascl1*, which encodes a transcription factor that is involved in the commitment and differentiation of neuron and oligodendrocytes[17], showed strong enrichment in the brain and also the dorsal spinal cord (Fig. 2c and Supplementary Fig. 2e,f). *Sox10* marks oligodendroglia and Schwann cells and presented high accessibility in the DRGs, but also in the brain and spinal cord region (Supplementary Fig. 2a,b). *Olig2* is a marker of neural progenitors and oligodendroglia[18] and is expressed in a limited region of the ventral spinal cord, in the ventral domains of the forebrain and in some posterior regions (brain stem, midbrain and hindbrain)—regions that also present high chromatin accessibility at the *Olig2* locus (Supplementary Fig. 2c,d). Interestingly, *Olig2* chromatin accessibility occurs in the dorsal forebrain at E13 (Supplementary Fig. 2c), suggesting the possibility of epigenetic priming at this region and stage without the activation of gene expression[19]. *Ror2* correlates with the early formation of the chondrocytes and cartilage, and it was highly expressed in the limb[15]. Pathway analysis revealed that cluster 1 was associated with erythrocyte differentiation, cluster 5 corresponded to forebrain development and cluster 6 was involved in limb development, consistent with anatomical annotations (Supplementary Fig. 3). Moreover, we further investigated the expression patterns in the spine, and some genes showed epigenetic gradients along the anterior–posterior axis (Supplementary Fig. 4).

In addition to the inference of cell-type-specific marker genes, our approach also enabled the identification of cell-type-specific chromatin regulatory elements (Extended Data Fig. 3), providing a resource for defining regulatory elements as cell-type-specific reporters. We examined cell-type-specific transcription factor regulators using deviations of transcription factor motifs (FDR < 0.05, $\log_2[FC] ≥ 0.1$) and found that the most enriched motifs in the peaks that are more accessible in the fetal liver correspond to GATA transcription factors, consistent with their well-studied role in erythroid differentiation[15] (Extended Data Fig. 3c,d). Cluster 5 is enriched for the SOX6 motif, supporting its role in CNS development. *Hoxd11*, which marks the posterior patterning and has a role in limb morphogenesis, was enriched in the limb (Extended Data Fig. 3d). We further conducted enrichment analysis using GREAT, and the pathways matched well with the anatomical annotation (Extended Data Fig. 3e).

We then integrated the spatial-ATAC-seq data with single-cell RNA-sequencing (scRNA-seq) data to assign cell types to each cluster[20] (Fig. 2d–f and Supplementary Fig. 5a). For example, the definitive erythroid cells appeared predominantly in the liver. Intermediate mesoderm was identified in the internal organ region, and radial glia were mainly distributed in the CNS. A refined clustering process also enabled the identification of subpopulations in excitatory neurons with distinct spatial distributions, marker genes and chromatin regulatory elements (Supplementary Fig. 5b–d). During embryonic development, dynamic changes in chromatin accessibility across time and space help to regulate the formation of complex tissue architectures and terminally differentiated cell types. In the embryonic CNS, radial glia function as primary progenitors or neural stem cells, which give rise to various types of

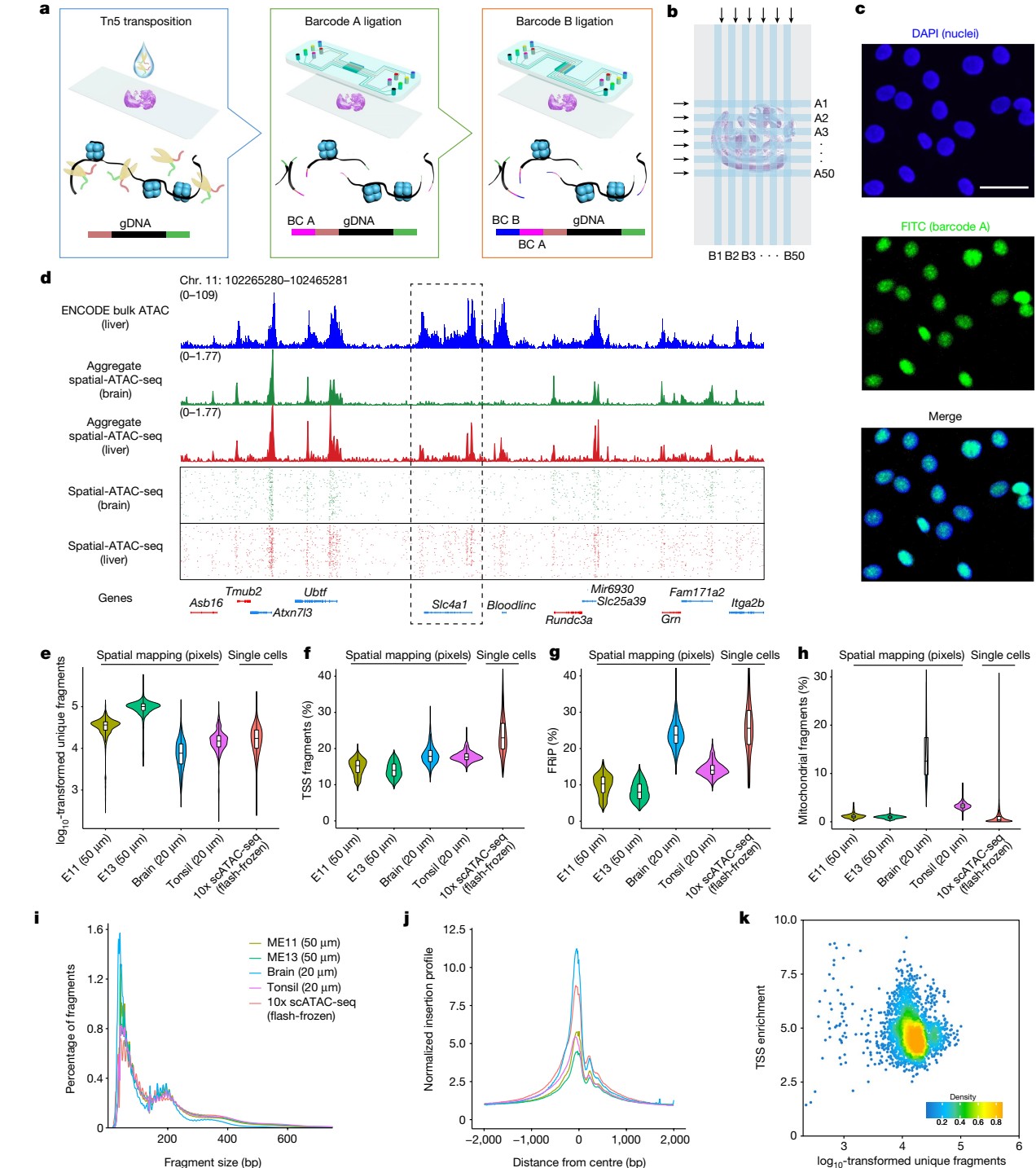

**Fig. 1 | Spatial-ATAC-seq design, workflow and data quality. a**, Schematic workflow. Tn5 transposition was performed in tissue sections, followed by in situ ligation of two sets of DNA barcodes (A1–A50, B1–B50). **b**, Microfluidic crossflow scheme. **c**, Validation of in situ transposition and ligation using fluorescent DNA probes. Tn5 transposition was performed in 3T3 cells on a glass slide stained with DAPI (blue). Next, FITC-labelled barcode A was ligated to the adapters on the transposase-accessible genomic DNA. Scale bar, 50 μm. **d**, Aggregate spatial chromatin accessibility profiles recapitulated published profiles of ATAC-seq in the liver of E13 mouse embryos. **e**, Comparison of the number of unique fragments between spatial-ATAC-seq and 10x scATAC-seq. **f**, Comparison of the fraction of TSS fragments between spatial-ATAC-seq and 10x scATAC-seq. **g**, Comparison of the fraction of mitochondrial fragments between spatial-ATAC-seq and 10x scATAC-seq. **h**, Comparison of the fraction of reads in peaks (FRiP) between spatial-ATAC-seq and 10x scATAC-seq. The number of pixels/cells in E11: 2,162; E13: 2,275; brain: 2,500; tonsil: 2,488; scATAC-seq: 3,789. The box plots show the median (centre line), the first and third quartiles (box limits), and 1.5× the interquartile range (whiskers). **i**, Comparison of the insert size distribution of ATAC-seq fragments between spatial-ATAC-seq and 10x scATAC-seq. **j**, Comparison of the enrichment of ATAC-seq reads around TSSs between spatial-ATAC-seq and 10x scATAC-seq. Colouring is consistent with **i**. **k**, The TSS enrichment score versus unique nuclear fragments per cell in human tonsils.

neuron[21]. We therefore sought to recover the developmental trajectory and examine how the developmental processes proceed across the tissue space. We focused on the course of differentiation from radial glia towards excitatory neurons with postmitotic premature neurons as the immediate state. Spatial projection of each pixel's pseudotime revealed the spatially organized developmental trajectory in neurons

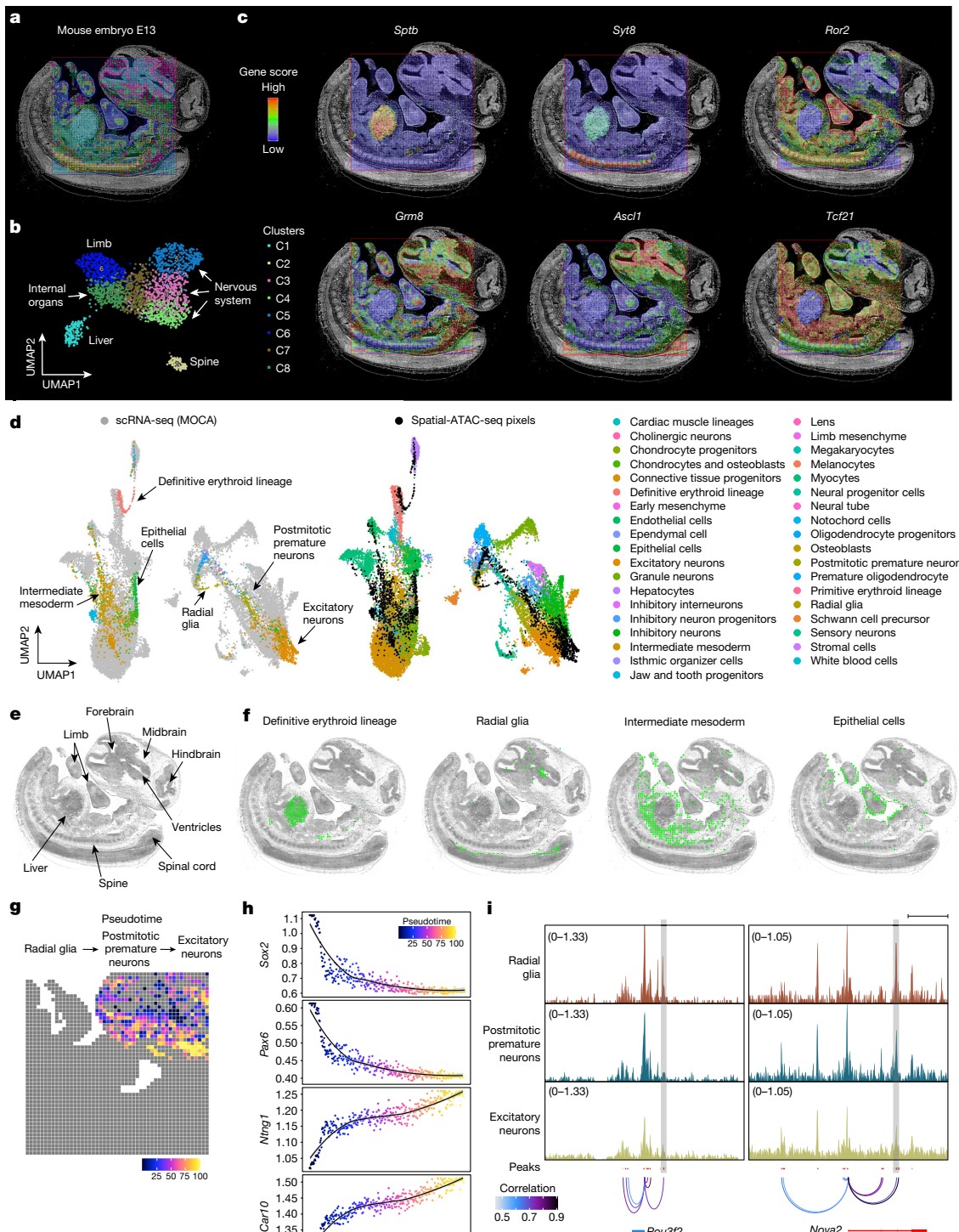

**Fig. 2 | Spatial chromatin accessibility mapping of E13 mouse embryos.**
**a**, An unbiased clustering analysis was performed on the basis of the chromatin accessibility of all tissue pixels (50 μm pixel size). An overlay of clusters with the tissue image reveals that the spatial chromatin accessibility clusters precisely match the anatomical regions. For better visualization, we scaled the size of the pixels. **b**, UMAP embedding of the unsupervised clustering analysis for chromatin accessibility. Cluster identities and colouring of clusters are consistent with **a**. **c**, The spatial mapping of gene scores for selected marker genes in different clusters and the chromatin accessibility at selected genes are highly tissue specific. **d**, Integration of scRNA-seq from E13.5 mouse embryos[20] and

spatial-ATAC-seq data. Unsupervised clustering of the combined data was coloured by different cell types. MOCA, Mouse Organogenesis Cell Atlas. **e**, Anatomical annotation of major tissue regions based on the haematoxylin and eosin (H&E)-stained image. **f**, Spatial mapping of selected cell types identified by label transferring from scRNA-seq to spatial-ATAC-seq data. **g**, Pseudotemporal reconstruction from the developmental process from radial glia, postmitotic premature neurons, to excitatory neurons plotted in space. **h**, The dynamics of the gene scores of selected genes along the pseudotime shown in **g**. **i**, The dynamics of the chromatin accessibility of individual regulatory elements at *Pou3f2* and *Nova2* (highlighted in grey boxes). Scale bar, 20 kb.

(Fig. 2g). We observed that cells early in differentiation clustered around the ventricles in the developing brainstem, whereas those farther away exhibited a more differentiated phenotype. We then identified changes in gene score across this developmental process, and observed high chromatin accessibility in radial glia at *Sox2* and *Pax6* loci, genes encoding transcription factors that are necessary for progenitor self-renewal identity[22]. As expected, there was a clear reduction in accessibility in the transition to postmitotic premature neurons and excitatory neurons, which instead presented chromatin opening at genes expressed in mature neurons, such as *Ntng1* and *Car10* (ref. [23]) (Fig. 2h). We also used the correlation of peak accessibility to predict the interactions between regulatory regions and found dynamically regulated promoter interactions with specific enhancers, such as *Pou3f2* and *Nova2* (Fig. 2i). *Pou3f2* (also known as *Brn2*), encodes a transcription factor that is expressed in mice in late progenitors and postmitotic neurons[24] and that has been shown to be involved in neural development for the production of specific neuronal populations[25]. Our analysis shows a reduction of chromatin accessibility at a specific *Brn2* enhancer during the transition from radial glial to postmitotic premature neurons, but not at other *cis*-regulatory regions, suggesting a role of this region in *Brn2* transcription. A similar decrease in chromatin accessibility was observed in excitatory neurons for a specific intronic enhancer of *Nova2*, which encodes an RNA-binding protein that is expressed in neurons[23]. Thus, our data indicated that spatial-ATAC-seq enables mapping at the spatial level of the chromatin accessibility dynamics at important regulatory regions during neural lineage commitment.

## Spatial mapping of the E11 mouse embryo

To further map chromatin accessibility during mouse fetal development, we profiled mouse embryos at an earlier stage (E11) and identified four clusters with distinct spatial patterns, which showed good agreement with the anatomy (Extended Data Figs. 4a,b and 5a,b). Cluster 1 is located in the fetal liver and aorta–gonad–mesonephros (AGM), which are related to embryonic haematopoiesis. Cluster 2 and cluster 3 consist of tissues associated with neuronal development, such as the mouse brain and neural tube. Cluster 4 includes the embryonic facial prominence, internal organs and limb. Furthermore, cluster identification matched the ENCODE organ-specific bulk ATAC-seq projection onto the UMAP (Extended Data Fig. 5d–g).

We further surveyed the chromatin accessibility that distinguished each cluster (FDR < 0.05, $\log_2[FC] \geq 0.25$) (Extended Data Fig. 4c and Extended Data Fig. 5c). For example, *Slc4a1*, which is required for normal flexibility and the stability of the erythrocyte membrane and for normal erythrocyte shape[15], was highly accessible in the liver and AGM. *Nova2*, which is involved in RNA splicing or metabolism regulation in a specific subset of developing neurons[26], was highly enriched in the brain and neural tube. *Rarg*, which has an essential role in limb bud development, skeletal growth and matrix homeostasis, was activated extensively in the embryonic facial prominence and limb[15]. Moreover, a Gene Ontology (GO) enrichment analysis identified the development processes consistent with the anatomical annotation (Supplementary Fig. 6). To gain deeper insights into the regulatory factors, we clustered chromatin regulatory elements and examined the enrichment for transcription-factor-binding motifs (Extended Data Fig. 6). We observed a strong enrichment of the motifs for *Gata2* and *Ascl2* (Extended Data Fig. 6c,d) in the clusters associated with embryonic haematopoiesis and neuronal development[15], respectively (FDR < 0.05, $\log_2[FC] \geq 0.1$). These master regulators further validated the unique identity of each cluster. Furthermore, the pathways matched well with the anatomical annotation in the enrichment analysis using GREAT (Extended Data Fig. 6e).

To assign cell types to each cluster, we integrated the spatial-ATAC-seq data with the scRNA-seq[20] (Extended Data Fig. 4d–f and Supplementary Fig. 7). The primitive erythroid cells, which are crucial for early embryonic erythroid development, were strongly enriched in the liver and AGM. Radial glia, postmitotic premature neurons and inhibitory neuron progenitors were found in the brain and neural tube. We observed abundant chondrocytes and osteoblasts in the embryonic facial prominence, and the limb mesenchyme was highly enriched in the limb region. Furthermore, compared with E13, hepatocytes and white blood cells could not be identified in the E11 liver region, suggesting that these cell types emerged at the later developmental time points. We also reconstructed the developmental trajectory from radial glia to excitatory neurons (Extended Data Fig. 4g–j) and identified the changes in neuron-development-related genes, which recapitulated transcription factor deviations across this developmental process, including *Notch1*, which is highly expressed in the radial glia and regulates neural stem cell number and function during development[27] (Extended Data Fig. 4h). Moreover, with increased spatial resolution (20 μm pixel size), spatial-ATAC-seq was able to resolve more cell types, such as a thin layer of notochord cells identified in cluster 3 (Extended Data Fig. 7).

To assess the temporal dynamics of chromatin accessibility more directly during development, we identified dynamic peaks that exhibit a significant change in accessibility from the E11 to E13 mouse embryo within fetal liver and excitatory neurons. We observed significant differences in the chromatin accessibility of fetal liver and excitatory neurons between different developmental stages (Supplementary Fig. 8). In particular, chromatin accessibility profiles of the fetal liver at E13 were enriched for GATA motif sequences (Supplementary Fig. 8d,f)—transcription factors that are known to be important in erythroid differentiation[14]. Moreover, the EGR1 motif was enriched in the excitatory neurons at E13 (Supplementary Fig. 8g), which has functional implications during brain development, particularly for the specification of excitatory neurons[28].

## Spatial mapping of the mouse and human brain

To further benchmark and integrate the spatial-ATAC-seq data with available scATAC-seq and scRNA-seq datasets, we applied spatial-ATAC-seq profiling to the P21 mouse brain coronal section at bregma 1 (Fig. 3a). Although nuclear staining with 7-aminoactinomycin D (7-AAD) resolved only the outline of the lateral ventricle (Fig. 3b), unsupervised clustering identified seven clusters with a unique spatial distribution, revealing the intricate arealization of this brain region (Fig. 3c,d). These clusters showed unique accessibility within marker gene regions (FDR < 0.05, $\log_2[FC] \geq 0.1$) for excitatory neurons (*Khdrbs3*, cluster 1), medium spiny neurons (*Pde10a*, cluster 2), lateral septal nucleus (*Dgkg*, cluster 4), corpus-callosum-enriched oligodendrocytes (*Mobp*, cluster 6) and ventricular-zone-enriched astroependymal cells (*Fgfr3*, cluster 7) (Fig. 3e). Integration and co-embedding the spatial-ATAC-seq data with the scATAC-seq atlas[29] validated the identity of the clusters (Fig. 3f,g and Supplementary Table 2). Label transfer from scRNA-seq[23] to spatial-ATAC-seq further confirmed the population assignments and revealed precise spatial localizations of the inferred cell identities (Fig. 3h and Extended Data Fig. 8). Integration with scATAC-seq and scRNA-seq data also helped to deconvolve additional cell types; for example, astrocytes, ependymal cells and subventricular zone neuroblasts within the C7 astroependymal cluster. A subpopulation of cluster 5, a cluster containing vascular cells, was assigned as arterial vascular smooth muscle cells and, accordingly, might line a thin layer of blood vessel, which could not be resolved at a lower spatial resolution (Fig. 3h, Extended Data Fig. 8b and Supplementary Fig. 9). Moreover, cells within the oligodendrocyte lineage could be assigned into distinct populations within a continuum from oligodendrocyte progenitors to mature, differentiated oligodendrocytes (Fig. 3f,g and Extended Data Fig. 8a,b). Although immature oligodendrocyte populations, such as differentiation-committed oligodendrocyte precursors, did not show clear spatial preferences for any of the analysed regions, we observed an enrichment of myelin-forming oligodendrocytes and

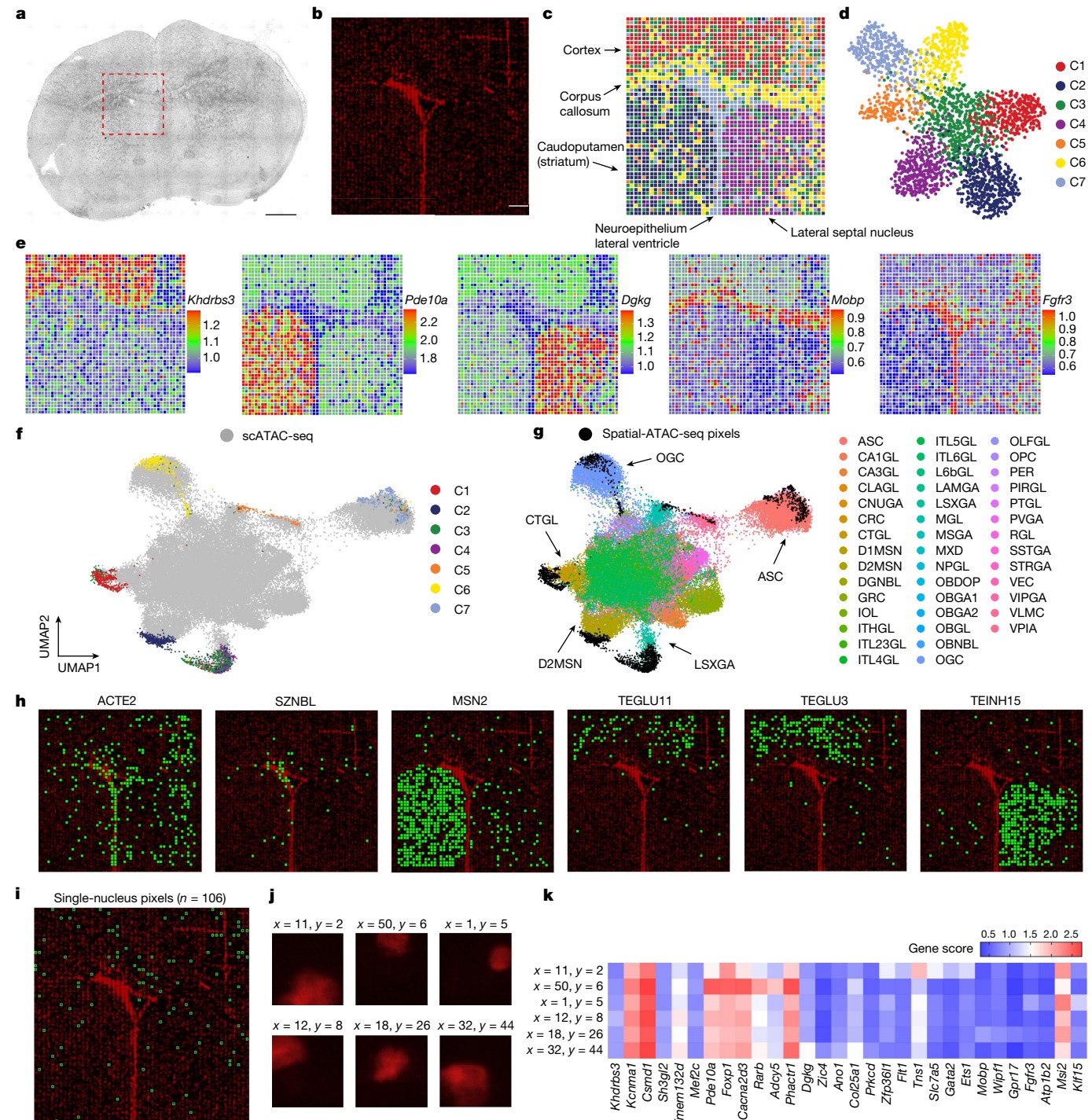

**Fig. 3 | Spatial chromatin accessibility mapping and integrative analysis of P21 mouse brain with a 20 μm pixel size. a**, Bright-field image of a mouse brain tissue section and the region of interest for mapping (red dashed box). Scale bar, 1 mm. **b**, Fluorescence image of nuclear staining with 7-AAD in the region of interest for spatial-ATAC-seq mapping. Scale bar, 200 μm. **c,d**, Unsupervised clustering analysis (**c**) and the spatial distribution (**d**) of each cluster in the mouse brain. For better visualization, we scaled the size of the pixels. **e**, Spatial mapping of gene scores for selected marker genes in different clusters. **f,g**, Integration of scATAC-seq from mouse brains[29] (**f**) and spatial-ATAC-seq (**g**). **h**, Spatial mapping of selected cell types identified by label transfer from scRNA-seq to spatial-ATAC-seq. **i**, The spatial location of pixels containing a single nucleus. **j**, Fluorescence images of selected pixels containing a single nucleus. **k**, Heat map of the gene scores of selected pixels containing a single nucleus. A list of abbreviation definitions can be found in Supplementary Table 2.

mature oligodendrocytes within the white-matter region of the corpus callosum (Extended Data Fig. 8b). Interestingly, integration of our previously generated spatial H3K4me3 and H3K27me3 CUT&Tag data-sets with single-cell CUT&Tag could not fully deconvolute the spatial progenitor and mature oligodendrocyte identities[30], underscoring the power of spatial-ATAC-seq to deconvolve cell types in the brain. Finally, we used the spatial-ATAC-seq data to identify pixels with a sin-gle nucleus, which would be equivalent to scATAC-seq data (Fig. 3i,j). We next visualized the gene scores in a heat map, demonstrating the ability of spatial-ATAC-seq to generate scATAC-seq profiles (Fig. 3k).

To further investigate the performance of our technology across species, we next used spatial-ATAC-seq on adult archival human coronal brain section, including the hippocampus and choroid plexus (Extended Data Fig. 9a). We were able to identify six clusters that were differentially distributed across the tissue section (Extended Data Fig. 9b). To refine and validate their identities, we performed further integration with a scATAC-seq dataset[31], which revealed enriched chromatin accessibility within specific gene regions for neurons (*vGLUT1*, *VGAT*, cluster 1), oligodendrocytes (*MAG*, cluster 2), astrocytes (*GFAP*, cluster 4) and microglia (*IBA1*, cluster 6) (Extended Data Fig. 9c,d). Spatial mapping of the identified clusters correlated well with their expected tissue localization, with the granule cell layer mainly consisting of neurons (Extended Data Fig. 9b,c). These results together confirmed the ability of spatial-ATAC-seq to spatially resolve different cellular populations within human brain tissue on the basis of their chromatin accessibility patterns.

## Spatial mapping of human tonsils

To further demonstrate the ability to profile spatial chromatin accessibility in different species and tissue types, we applied spatial-ATAC-seq to human tonsil tissue. Unsupervised clustering revealed distinct spatial features with the germinal centres (GC) identified mainly in cluster 1 (Fig. 4a–c and Extended Data Fig. 10a). We next examined the spatial patterns of specific marker genes to distinguish cell types (FDR < 0.05, log2[FC] ≥ 0.1) (Fig. 4d and Supplementary Fig. 10) and compared these data with the distribution of protein expression in tonsils (Supplementary Fig. 11). For B-cell-related genes, the accessibility of *CD10*, a marker for mature GC B cells[32], was enriched in the GC regions. *CD27*, a marker for memory B cells[33], was active in the GC and the extrafollicular regions. *CD38*, which marks activated B cells[34], was found to be enriched in the GC. *CXCR4*, which is expressed in the centroblasts in the GC dark zone[35,36], unexpectedly showed high accessibility in only non-GC cells. This discordance between epigenetic state and protein expression may suggest epigenetic priming of pre-GC B cells before entering the GC. It could also be due to the presence of CXCR4+ T cells supporting extrafollicular B cell responses in the setting of inflammation[37]. *PAX5*, a transcription factor for follicular and memory B cells[38], was enriched in the GC but was also observed in the extrafollicular zones in which the memory B cells migrated to. *BHLHE40*, a transcription factor that can bind to the major regulatory regions of the IgH locus, was observed to be highly enriched in the extrafollicular region but completely depleted in the GC, suggesting a potential role in the regulation of class-switch recombination in the pre-GC state[39]. This supports a model of epigenetic control for class-switch recombination that occurs before the formation of the GC response. For T cell-related genes, *CD3* corresponded to T cell zones[40] and was also found to be active in the GC. It is known that trafficking of follicular helper T cells (T$_{FH}$) into the GC requires downregulation of CCR7 and upregulation of CXCR5 (ref. [41]). We observed significantly reduced *CCR7* accessibility in the GC and strong enrichment outside the GC, indicating that this T$_{FH}$ function is indeed epigenetically regulated. *CXCR5* accessibility was extensively detected in the GC but was also observed outside the GC, indicating a possible early epigenetic priming of T$_{FH}$ cells before GC entry for B cell help. The accessibility of *BCL6*, a T$_{FH}$ master transcription factor[42], was strongly enriched in the GC as expected. *FOXP3*, a master transcription factor for follicular regulatory T cells[43], is mainly in the extrafollicular zone but at low frequency according to human protein atlas data (Supplementary Fig. 11). Interestingly, it showed extensive open locus accessibility, suggesting extensive epigenetic priming of pre-GC T cells to potentially develop follicular regulatory T cell function as needed to balance GC activity. *CD25*, a surface marker for regulatory T cells[44], was active in both the GC and the extrafollicular zone. For non-lymphoid cells, *CD11B*, a macrophage marker[45], was inactive in the GC, in contrast to *CD11A*, which was more active in GC lymphocytes.

*CD103* was enriched in GC follicular dendritic cells. *CD144*, which encodes vascular endothelial cadherin (VE-cadherin)[15], corresponded to endothelial microvasculature near to the crypt or between follicles. *CD32*, a surface receptor that is involved in phagocytosis and clearing of immune complexes[46], and *CD55*, a complement decay-accelerating factor, were both active in the same region such that the cells that are not supposed to be cleared can be protected against phagocytosis by blocking the formation of the membrane attack complex[47]. We also examined cell-type-specific transcription factor regulators within each cluster and our data revealed that KLF-family transcription factors were highly enriched in non-GC cells, consistent with a previous study[48] (FDR < 0.05, log$_2$[FC] ≥ 0.1) (Supplementary Fig. 12).

To map cell types onto each cluster, we integrated spatial-ATAC-seq data with scRNA-seq and scATAC-seq datasets[48] (Fig. 4e and Extended Data Fig. 10b). After label transfer from scRNA-seq, we found that cells from cluster 0 were widely distributed in the non-GC region, whereas cells from cluster 4 were enriched in GC (Fig. 4e,f and Extended Data Fig. 10c). We also identified a small region with cells enriched from cluster 13 (Fig. 4f and Extended Data Fig. 10c). To define the cell identities for scRNA-seq clusters, we examined the marker genes for each cluster and found that cluster 0 comprised naive B cells, cluster 4 corresponded to GC B cells and cluster 13 was macrophages (Extended Data Fig. 10d), in agreement with the tissue histology (Fig. 4f).

Lymphocyte activation, maturation and differentiation are regulated by the gene networks under the control of transcription factors[48]. To understand the dynamic regulation process, we implemented a pseudotemporal reconstruction of B cell activation to the GC reaction (Fig. 4g–i). Meanwhile, the projection of each pixel's pseudotime onto spatial coordinates revealed spatially distinct regions in this dynamic process. Interestingly, we found that the enriched macrophage population colocalized with inactivated B cells, consistent with the fact that B cells are activated through acquiring antigens from antigen-presenting macrophages before GC entry or formation[49] (Fig. 4g). Moreover, pseudotemporal ordering of B cell activation revealed dynamic expression and chromatin activity before commitment to the GC state (Fig. 4h), including an early activity of *BCL2* and reduced accessibility within GC B cells as compared to naive populations, suggesting that this antiapoptotic molecule may be actively repressed to ensure that GC B cells are eliminated by apoptosis if they are not selected and rescued by survival signals. By contrast, *LMO2* exhibited increased accessibility at the target sites within GC B cells (Extended Data Fig. 10e), consistent with the previous finding that *LMO2* is specifically upregulated in the GC[50]. We also identified putative target genes of fine-mapped autoimmune genome-wide association study (GWAS) genetic variants, and revealed GC-specific regulatory potential, including at loci of major GC regulators such as *BCL6* (Fig. 4i).

## Discussion

We developed spatial-ATAC-seq for spatially resolved profiling of chromatin accessibility in intact tissue sections with spatial information retained at the cellular level (20 μm pixel size). Single-cell chromatin accessibility can also be derived without tissue dissociation by identifying pixels containing only one nucleus using immunofluorescence imaging. Spatial-ATAC-seq was applied to mouse embryos (E11 and E13) to delineate the epigenetic landscape of organogenesis; we identified all major tissue types with a distinct chromatin accessibility state, and revealed the spatiotemporal changes in development. Mapping the accessible genome in the mouse and human brain revealed the intricate arealization of brain region. We also used spatial-ATAC-seq to map the epigenetic state of different immune cells in human tonsils and revealed the dynamics of B cell activation to GC reaction and putative target genes of fine-mapped autoimmune GWAS genetic variants. Compared with spatial-CUT&Tag for the targeted profiling of histone modifications[30], spatial-ATAC-seq provided a genome-wide

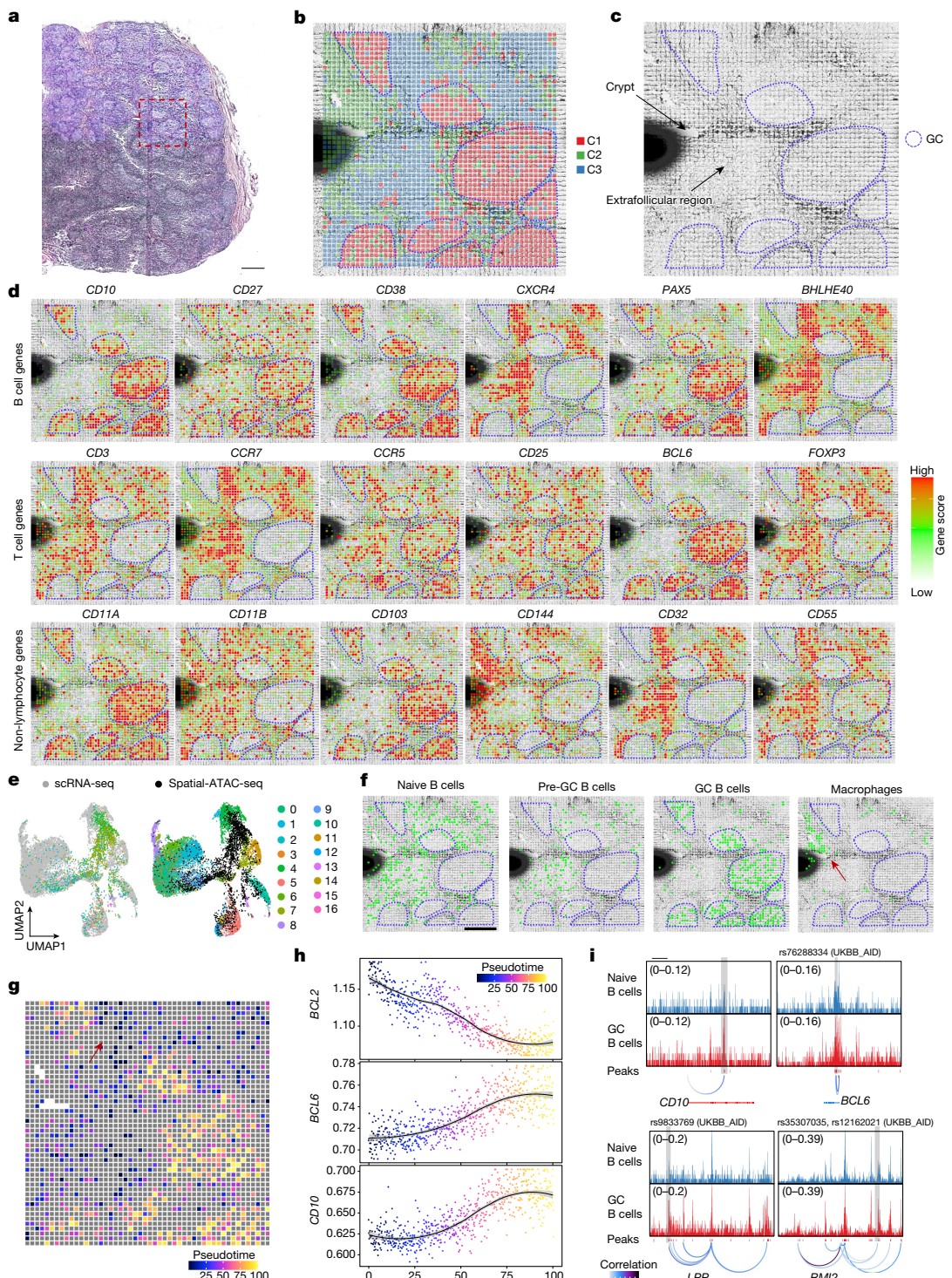

**Fig. 4 | Spatial chromatin accessibility mapping of a human tonsil with a 20 μm pixel size. a**, H&E image of a human tonsil from an adjacent tissue section and a region of interest for spatial chromatin accessibility mapping. Scale bar, 1 mm. **b**, Unsupervised clustering analysis and spatial distribution of each cluster. For better visualization, we scaled the size of the pixels. **c**, Anatomical annotation of major tonsillar regions. **d**, Spatial mapping of the gene scores for selected genes. **e**, Integration of scRNA-seq data[48] and spatial-ATAC-seq data. Unsupervised clustering of the combined data was coloured by different cell types. **f**, Spatial mapping of selected cell types identified by label transferring from scRNA-seq to spatial-ATAC-seq data. Scale bar, 500 μm. **g**, Pseudotemporal reconstruction from the developmental process from naive B cells to GC B cells plotted in space. **h**, Dynamics of the gene scores of selected genes along the pseudotime shown in **g**. **i**, Dynamics of the chromatin accessibility of individual regulatory elements along pseudotime (highlighted in grey boxes). Fine-mapped autoimmune-associated GWAS variants and high-resolution individual single-nucleotide polymorphism loci localizing to accessible chromatin are shown. Scale bar, 25 kb.

chromatin accessibility landscape, for which it is challenging to obtain a high signal-to-noise ratio, especially in fresh frozen tissue sections.

The areas for further development include the following. First, the number of mapping pixels could be further increased by increasing the number of barcodes (for example, 100 × 100) or using serpentine

microfluidic channels for tissue array. Second, different from single-cell technologies, the pixels in spatial-ATAC-seq may contain partial nuclei or multiple nuclei and could lead to cell mixtures that may comprise multiple cell types, complicating data interpretation. This challenge could be addressed by using cell-type deconvolution approaches or seamless integration with high-resolution tissue images, that is, multi-colour immunofluorescence images, to identify the cells in each pixel. We observed that a significant number of pixels (20 μm) contained single nuclei, which can give rise to spatially defined scATAC-seq data. Third, integration with other spatial omics measurements such as tran-scriptomics and proteomics, can provide a comprehensive picture of cell types and cell states. We may combine reagents for DBiT-seq[5] and spatial-ATAC-seq in the same microfluidic barcoding step to achieve spatial multi-omics profiling, which should work in theory but does require further optimization for tissue fixation and reaction condi-tions to make these assays compatible. Finally, spatial-ATAC-seq is yet to be further extended to tissue samples from human patients with a disease to realize its full potential in clinical research. We anticipate that spatial-ATAC-seq will add a new dimension to spatial biology, enabling the profiling of regulatory elements in a spatially resolved manner, which cannot be achieved by spatial transcriptomics or proteomics. Spatial-ATAC-seq may transform multiple biomedical research fields including developmental biology, neuroscience, immunology, oncol-ogy and clinical pathology, therefore empowering scientific discovery and translational medicine in human health and disease.

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

## Methods

### Juvenile mouse brain and sample preparation

Experimental procedures on juvenile (P21) mice were conducted in accordance with the European directive 2010/63/EU, local Swedish directive L150/SJVFS/2019:9, Saknr L150 and the Karolinska Institutet complementary guidelines for procurement and use of laboratory animals, Dnr 1937/03-640. The procedures described here were approved by Stockholms Norra Djurförsöksetiska nämnd, the local committee for ethical experiments on laboratory animals in Sweden, 1995/2019 and 7029/2020.

The mouse line *Sox10:cre-RCE:LoxP* (eGFP), on a C57BL/6xCD1 mixed genetic background, was used for experiments on P21 mice. It was generated by crossing *Sox10:cre* animals[51] (Jackson Laboratory, 025807) on a C57BL/6j genetic background with *RCE:loxP* (enhanced green fluorescent protein (eGFP)) animals[52] (Jackson Laboratory, 32037-JAX) on a C57BL/6xCD1 mixed genetic background. Breeding of female mice with a hemizygous *cre* allele with male mice lacking the *cre* allele (while the reporter allele was kept in hemizygosity or homozygosity in both female and male mice) resulted in labelling of the oligodendrocyte lineage with eGFP. Mice, free of common viral pathogens, ectoparasites, endoparasites and mouse bacterial pathogens, were housed to a maximum of five mice per cage in individually ventilated cages (IVC Sealsafe GM500, Tecniplast). The cages were equipped with hardwood bedding (TAPVEI), nesting material, shredded paper, gnawing sticks and a cardboard box shelter (Scanbur). Mice received regular chew diet and water using a water bottle that was changed weekly. Cages were changed every other week in a laminar air-flow cabinet. General housing parameters, such as relative humidity, temperature and ventilation, were used according to the European convention for the protection of vertebrate animals used for experimental and other scientific purposes treaty ETS 123. Specifically, consistent relative air humidity and temperature were set to 50% and 22 °C, and the air quality was controlled with the use of stand-alone air handling units supplemented with a HEPA filter. Husbandry parameters were monitored using the ScanClime (Scanbur) units. The following light–dark cycle was used: dawn, 6:00–7:00; daylight, 7:00–18:00; dusk, 18:00–19:00; night, 19:00–6:00.

### Post-mortem human brain and sample preparation

One human hippocampus sample was obtained from the Brain Collection of the New York State Psychiatric Institute (NYSPI) and Columbia University[53]. This brain sample was a fresh frozen unfixed specimen. All procedures of brain collection and autopsy were conducted with Institutional Review Board approval and informed consent from the next of kin. The participant selected was free of neuropsychiatric illness on the basis of our validated psychological autopsy interview of the next of kin[54], died of sudden death (industrial accident) with short agonal state (that can affect brain oxygenation if prolonged), had short post-mortem interval (6.5 h), clear neuropathological exam, negative brain toxicology for psychoactive drugs, medication and alcohol, and good RNA quality (RNA integrity number 8.50).

The anterior hippocampus was dissected from a 2-cm-thick coronal block of the right hemisphere, and sectioned at a thickness of 10 μm using a cryostat (Leica 3050S). Each of the serial sections of the dentate gyrus region (around 1 cm × 1 cm) were placed onto an ultraclean glass slide (Electron Microscopy Sciences, 63478-AS). We took one slide for sequencing and performed H&E and Nissl staining on two adjacent slides. All of the samples were stored at −80 °C before use.

### Fabrication and assembly of the microfluidic device

The moulds for microfluidic devices were fabricated in the cleanroom using standard photolithography. We followed the manufacturer's guidelines to spin-coat SU-8-negative photoresist (SU-2010, SU-2025, Microchem) onto a silicon wafer (C04004, WaferPro). The feature heights of the 50-μm-wide and 20-μm-wide microfluidic channel device were about 50 μm and 23 μm, respectively. During exposure to ultraviolet light, chrome photomasks (Front Range Photomasks) were used. Soft lithography was used for the fabrication of polydimethylsiloxane (PDMS) microfluidic devices. We mixed base and curing agent at a 10:1 ratio and added it over the SU-8 masters. The PDMS was cured (at 65 °C for 2 h) after degassing in a vacuum (30 min). After solidification, the PDMS slab was cut out. The outlet and inlet holes were punched for further use. We have published a protocol in terms of device fabrication and operation[55].

### Preparation of tissue slides

Mouse C57 Embryo Sagittal Frozen Sections (MF-104-11-C57) and Human Tonsil Frozen Sections (HF-707) were purchased from Zyagen. Tissues were snap-frozen in optimal cutting temperature compounds, sectioned (thickness of 7–10 μm) and put at the centre of poly-L-lysine-covered glass slides (63478-AS, Electron Microscopy Sciences).

### H&E staining

The frozen slide was warmed at room temperature for 10 min and fixed with 1 ml 4% formaldehyde (10 min). After being washed once with 1× DPBS, the slide was quickly dipped in water and dried with air. Isopropanol (500 μl) was then added to the slide and incubated for 1 min before being removed. After completely dry in the air, the tissue section was stained with 1 ml haematoxylin (Sigma-Aldrich) for 7 min and cleaned in deionized water. The slide was then incubated in 1 ml bluing reagent (0.3% acid alcohol, Sigma-Aldrich) for 2 min and rinsed in deionized water. Finally, the tissue slide was stained with 1 ml eosin (Sigma-Aldrich) for 2 min and cleaned in deionized water.

### Preparation of the transposome

Unloaded Tn5 transposase (C01070010) was purchased from Diagenode, and the transposome was assembled according to the manufacturer's guidelines. The oligos used for transposome assembly were as follows: Tn5MErev, 5′-/5Phos/CTGTCTCTTATACACATCT-3′; Tn5ME-A, 5′-/5Phos/CATCGGCGTACGACTAGATGTGTATAAGAGACAG-3′; Tn5ME-B, 5′-GTCTCGTGGGCTCGGAGATGTGTATAAGAGACAG-3′.

### DNA oligos, DNA barcode sequences and other key reagents

Lists of the DNA oligos that were used for sequencing library construction and PCR (Supplementary Table 3), DNA barcode sequences (Supplementary Table 3) and all other key reagents (Supplementary Table 4) are provided.

### Spatial-ATAC-seq profiling

As we proceeded to develop spatial-ATAC-seq, we went through several versions of chemistry to optimize the protocol to achieve a high yield and a high signal-to-noise ratio for the mapping of tissue sections (Supplementary Fig. 2a). First, a set of 50 DNA oligomers containing both barcode A and adapter were introduced in microchannels to a tissue section for in situ transposition, but the efficiency was low due in part to limited amounts of Tn5 DNA in the microchannels. To address this issue, we conducted bulk transposition followed by two ligation steps to introduce spatial barcodes A and B. We also optimized the fixation condition by reducing the formaldehyde concentration from 4% in chemistry V1 to 0.2%. Furthermore, we tested the sensitivity of different Tn5 transposase enzymes (Diagenode (C01070010) versus Lucigen (TNP92110)).

In the optimized spatial-ATAC-seq protocol, the frozen slide was warmed at room temperature for 10 min. The tissue was then fixed with formaldehyde (0.2% for 5 min) and quenched with glycine (1.25 M for 5 min) at room temperature. After fixation, the tissue was washed twice with 1 ml 1× DPBS and cleaned in deionized water. The tissue

section was then permeabilized with 500 µl lysis buffer (10 mM Tris-HCl, pH 7.4, 10 mM NaCl, 3 mM MgCl$_2$, 0.01% Tween-20, 0.01% NP-40, 0.001% digitonin, 1% BSA) for 15 min and was washed with 500 µl wash buffer (10 mM Tris-HCl pH 7.4, 10 mM NaCl, 3 mM MgCl$_2$, 1% BSA, 0.1% Tween-20) for 5 min. Then, 100 µl transposition mix (50 µl 2× tagmentation buffer, 33 µl 1× DPBS, 1 µl 10% Tween-20, 1 µl 1% digitonin, 5 µl transposome, 10 µl nuclease-free H$_2$O) was added followed by incubation at 37 °C for 30 min. After removing the transposition mix, 500 µl 40 mM EDTA was added for incubation at room temperature for 5 min to stop transposition. Finally, the EDTA was removed, and the tissue section was washed with 500 µl 1× NEBuffer 3.1 for 5 min.

For barcode A in situ ligation, the first PDMS slab was used to cover the region of interest, the bright-field image was taken using a ×10 objective (Thermo Fisher Scientific, EVOS FL Auto microscope (AMAFD1000), EVOS FL Auto Software (REV 32044)) for further alignment. The tissue slide and PDMS device were then clamped with an acrylic clamp. First, DNA barcode A was annealed with ligation linker 1: 10 µl of each DNA barcode A (100 µM), 10 µl of ligation linker (100 µM) and 20 µl of 2× annealing buffer (20 mM Tris, pH 7.5–8.0, 100 mM NaCl, 2 mM EDTA) were added together and mixed well. Then, 5 µl ligation reaction solution (50 tubes) was prepared by adding 2 µl of ligation mix (72.4 µl of RNase-free water, 27 µl of T4 DNA ligase buffer, 11 µl T4 DNA ligase, 5.4 µl of 5% Triton X-100), 2 µl of 1× NEBuffer 3.1 and 1 µl of each annealed DNA barcode A (A1–A50, 25 µM) and loaded into each of the 50 channels under a vacuum. The chip was kept in a wet box for incubation (37 °C, 30 min). After flowing through 1× NEBuffer 3.1 for washing (5 min), the clamp and PDMS were removed. The slide was quickly dipped in water and dried with air.

For barcode B in situ ligation, the second PDMS slab with channels perpendicular to the first PDMS was attached to the dried slide carefully. A bright-field image was taken and the acrylic clamp was used to press the PDMS against the tissue. The annealing of DNA barcode B with ligation linker 2 was performed the same as described above for the annealing of DNA barcode A and ligation linker 1. The preparation and addition of the ligation reaction solution for DNA barcode B (B1–B50, 25 µM) were also the same as described for DNA barcode A (A1–A50, 25 µM). The chip was kept in a wet box for incubation (37 °C for 30 min). After flowing through 1× DPBS for washing (5 min), the clamp and PDMS were removed, the tissue section was dipped in water and dried with air. The final bright-field image of the tissue was taken.

For tissue digestion, the region of interest of the tissue was covered with a square PDMS well gasket, and 100 µl reverse cross-linking solution (50 mM Tris-HCl, pH 8.0, 1 mM EDTA, 1% SDS, 200 mM NaCl, 0.4 mg ml$^{-1}$ proteinase K) was loaded into it. The lysis was conducted in a wet box (58 °C, 2 h). The final tissue lysate was collected into a 200 µl PCR tube for incubation with rotation (65 °C, overnight).

For library construction, the lysate was first purified using the Zymo DNA Clean & Concentrator-5 kit and eluted into 20 µl of DNA elution buffer, followed by mixing with the PCR solution (2.5 µl 25 µM new P5 PCR primer, 2.5 µl 25 µM Ad2 primer, 25 µl 2× NEBNext Master Mix). Then, PCR was performed using the following program: 72 °C for 5 min; 98 °C for 30 s; and then cycled 5 times at 98 °C for 10 s, 63 °C for 10 s and 72 °C for 1 min. To determine additional cycles, 5 µl of the pre-amplified mixture was first mixed with the qPCR solution (0.5 µl 25 µM new P5 PCR primer, 0.5 µl 25 µM Ad2 primer, 0.24 µl 25× SYBR Green, 5 µl 2× NEBNext Master Mix, 3.76 µl nuclease-free H$_2$O). Then, the qPCR reaction was performed under the following conditions: 98 °C for 30 s; and then 20 cycles of 98 °C for 10 s, 63 °C for 10 s and 72 °C for 1 min. Finally, the remaining 45 µl of the pre-amplified DNA was amplified by running the required number of additional cycles of PCR (the cycles needed to reach 1/3 of the saturated signal in qPCR).

To remove PCR primer residues, the final PCR product was purified using 1× Ampure XP beads (45 µl) according to the standard protocol and eluted into 20 µl nuclease-free H$_2$O. Before sequencing, an Agilent Bioanalyzer High Sensitivity Chip was used to quantify the concentration and size distribution of the library. Next-generation sequencing was performed using the Illumina HiSeq 4000 or NovaSeq 6000 sequencer (paired-end 150 bp mode with custom read 1 primer).

## Data preprocessing

Two constant linker sequences (linker 1 and linker 2) were used to filter read 1, and the filtered sequences were transformed to Cell Ranger ATAC format (10x Genomics). The genome sequences were included in the new read 1, barcodes A and barcodes B were included in the new read 2. The resulting fastq files were aligned to the mouse reference (mm10) or human reference (GRCh38) genome, filtered to remove duplicates and counted using Cell Ranger ATAC v.1.2. The BED-like fragments files were generated for downstream analysis. The fragments file contains fragments of information on the genome and tissue location (barcode A × barcode B). A preprocessing pipeline we developed using Snakemake workflow management system (v5.28.0) is available at GitHub (https://github.com/dyxmvp/Spatial_ATAC-seq).

## Data visualization

We first identified pixels on tissue samples by manual selection from microscopy images using Adobe Illustrator (v.25.4.3) (https://github.com/rongfan8/DBiT-seq), and a custom Python script was used to generate metadata files that were compatible with the Seurat workflow for spatial datasets.

The fragment file was read into ArchR as a tile matrix with a genome binning size of 5 kb, and pixels that were not on the tissue were removed on the basis of the metadata file generated in the previous step. Data normalization and dimensionality reduction was conducted using iterative latent semantic indexing (iterations = 2, resolution = 0.2, varFeatures = 25000, dimsToUse = 1:30, n.start = 10), followed by graph clustering and UMAP embedding (nNeighbors = 30, metric = cosine, minDist = 0.5)[14].

The Gene Score model in ArchR was used to generate the gene accessibility score. A gene score matrix was generated for downstream analysis. The getMarkerFeatures and getMarkers function in ArchR (testMethod = "wilcoxon", cutOff = "FDR < = 0.05") was used to identify the marker regions/genes for each cluster, and the marker genes were discussed in the manuscript because they were identified as one of the top differential genes between clusters, and they were also known in the literature. Gene-score imputation was implemented with addImputeWeights for data visualization. The enrichGO function in the clusterProfiler package was used for GO enrichment analysis (qvalueCutoff = 0.05)[56]. For spatial data visualization, results obtained in ArchR were loaded into Seurat v.3.2.3 to map the data back to the tissue section[57,58]. For better visualization, we scaled the size of the pixels using the 'pt.size.factor' parameter in the Seurat package (Extended Data Fig. 8c).

Genome browser tracks were plotted using the plotBrowserTrack function in ArchR. Spatial-ATAC-seq data were normalized to the recommended and default value (normMethod = "ReadsInTSS"), which simultaneously normalizes tracks based on sequencing depth and sample data quality. Blue-coloured genes are on the minus strand and red-coloured genes are on the plus strand. The loops are the links between a peak and a gene, and the colour shows the Pearson correlation between peak accessibility and gene expression. Peaks were called with macs2 using addReproduciblePeakSet function in ArchR.

To project bulk ATAC-seq data, we downloaded raw sequencing data aligned to the mm10 genome (BAM files) from ENCODE. After counting the reads in 5 kb tiled genomes using the getCounts function in chromVAR[59], the ENCODE ATAC-seq data were subsampled in pseudo single cells (*n* = 250) and were projected onto spatial-ATAC UMAPs using the projectBulkATAC function in ArchR.

Cell type identification and pseudo-scRNA-seq profiles were added through integration with scRNA-seq reference data[20]. The

FindTransferAnchors function (Seurat v.3.2 package) was used to align pixels from spatial-ATAC-seq with cells from scRNA-seq by comparing the spatial-ATAC-seq gene score matrix with the scRNA-seq gene expression matrix. The GeneIntegrationMatrix function in ArchR was used to add cell identities and pseudo-scRNA-seq profiles.

Pseudobulk group coverages based on cluster identities were generated using the addGroupCoverages function and used for peak calling with macs2 using the addReproduciblePeakSet function in ArchR. To compute per-cell motif activity, chromVAR[59] was run with addDeviationsMatrix using the cisbp motif set after a background peak set was generated using addBgdPeaks. Cell-type-specific marker peaks were identified using the getMarkerFeatures (bias = c("TSSEnrichment", "log10(nFrags)", testMethod = "wilcoxon") and getMarkers (cutOff = "FDR < = 0.05 & Log2FC > = 0.1") functions.

Pseudotemporal reconstruction was implemented by trajectory analysis using ArchR. We used the cell-type definitions from label transfer as described above. A trajectory backbone was first created in the form of an ordered vector of cell group labels. We then used the addTrajectory function to create a trajectory, and added the pseudotime to Seurat spatial object to map the data back to the tissue section. Dynamics for selected gene score along the pseudotime were plotted with the plotTrajectory function using the default values.

Correlation analysis was conducted by calculating the Pearson correlation coefficient in R. P values were calculated using the cor. test function and were adjusted for multiple comparisons using the Benjamini–Hochberg method[60].

### Reporting summary

Further information on research design is available in the Nature Research Reporting Summary linked to this article.

## Data availability

Raw and processed data reported in this paper are deposited in the Gene Expression Omnibus (GEO) with accession code GSE171943. The resulting fastq files were aligned to the mouse reference genome (mm10) or human reference genome (GRCh38). Published data for data quality comparison and integrative data analysis are available online: flash frozen cortex, hippocampus and ventricular zone from embryonic mouse brain (E18) (https://www.10xgenomics.com/resources/datasets/flash-frozen-cortex-hippocampus-and-ventricular-zone-from-embryonic-mouse-brain-e-18-1-standard-1-2-0), ENCODE mouse embryo ATAC-seq (11.5 days) (https://www.encodeproject.org/search/?type=Experiment&status=released&related_series.@type=OrganismDevelopmentSeries&replicates.library.biosample.organism.scientific_name=Mus+musculus&assay_title=ATAC-seq&life_stage_age=embryonic%2011.5%20days), ENCODE mouse embryo ATAC-seq (13.5 days) (https://www.encodeproject.org/search/?type=Experiment&status=released&related_series.@type=OrganismDevelopmentSeries&replicates.library.biosample.organism.scientific_name=Mus+musculus&assay_title=ATAC-seq&life_stage_age=embryonic%2013.5%20days), Mouse Organogenesis Cell Atlas (MOCA) (https://oncoscape.v3.sttrcancer.org/atlas.gs.washington.edu.mouse.rna/downloads), Atlas of Gene Regulatory Elements in Adult mouse Cerebrum (http://catlas.org/mousebrain/#!/downloads), Atlas of the Adolescent Mouse Brain (http://mousebrain.org/adolescent/downloads.html), human hippocampus scATAC-seq data (GSE147672), human tonsil scATAC-seq data (GSE165860), human tonsil scRNA-seq data (GSE165860), and the Allen Developing Mouse Brain Atlas (https://developingmouse.brain-map.org/). A list of published data for data quality comparison and integrative data analysis is provided in in Supplementary Table 5.

## Code availability

Code for sequencing data analysis is available at GitHub (https://github.com/dyxmvp/Spatial_ATAC-seq) and archived at Zenodo (https://doi.org/10.5281/zenodo.6565118).

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

**Acknowledgements** We thank the staff at the Yale Center for Research Computing for guidance and use of the research computing infrastructure; T. Jimenez-Beristain in the G.C.-B laboratory for writing laboratory animal ethics permits and for assistance with animal experiments; T. Wu, T. Li and R. Hen at Columbia University for scientific discussions and suggestions; and C. Sissoko and A. N. Santiago at Columbia University for help with anatomical annotation of the human hippocampus slices. The moulds for microfluidic devices were fabricated at the Yale SEAS cleanroom and the Yale West Campus Cleanroom. Next-generation sequencing was conducted at the Yale Stem Cell Center Genomics Core Facility, which was supported by the Connecticut Regenerative Medicine Research Fund and the Li Ka Shing Foundation. Services provided by the Genomics Core of Yale Cooperative Center of Excellence in Hematology (U54DK106857) were used. This research was supported by Packard Fellowship for Science and Engineering (to R.F.), Stand-Up-to-Cancer (SU2C) Convergence 2.0 Award (to R.F.) and Yale Stem Cell Center Chen Innovation Award (to R.F.). It was also supported in part by grants from the US National Institutes of Health (NIH) (U54AG076043 to R.F., S.H., J.E.C. and M.L.X.; UG3CA257393, R01CA245313 and RF1MH128876, U01CA260507 to R.F.; UH3TR002151 to K.W.L.). Y.L. was supported by the Society for ImmunoTherapy of Cancer (SITC) Fellowship. Work in G.C.-B.'s research group was supported by the Swedish Research Council (grant 2019-01360), the European Union (Horizon 2020 Research and Innovation Programme/European Research Council Consolidator Grant EPIScOPE, agreement number 681893), the Swedish Cancer Society (Cancerfonden; 190394 Pj), the Knut and Alice Wallenberg Foundation (grants 2019-0107 and 2019-0089), the Swedish Society for Medical Research (SSMF, grant JUB2019), the Göran Gustafsson Foundation for Research in Natural Sciences and Medicine, the Ming Wai Lau Centre for Reparative Medicine and the Karolinska Institutet.

**Author contributions** investigation: Y.D., D.Z., P.K. and Y.X.; Data analysis: Y.D., M. Bartosovic, D.Z., P.K., G.C.-B. and R.F.; Resources: G.S., X.Q., K.W.L., G.B.R., A.J.D., J.J.M. and M. Boldrini; S.M., M.L.X., S.H. and J.E.C. provided advice and input; Original draft: Y.D., D.Z. and R.F. All of the authors reviewed, edited and approved the manuscript.

**Competing interests** R.F. and Y.D. are inventors of a patent application related to this work (PCT Patent Application No. PCT/US2021/065669). R.F. is scientific founder and advisor of IsoPlexis, Singleron Biotechnologies and AtlasXomics. The interests of R.F. were reviewed and managed by Yale University Provost's Office in accordance with the University's conflict of interest policies. The other authors declare no competing interests.

**Additional information**
**Correspondence and requests for materials** should be addressed to Gonçalo Castelo-Branco or Rong Fan.

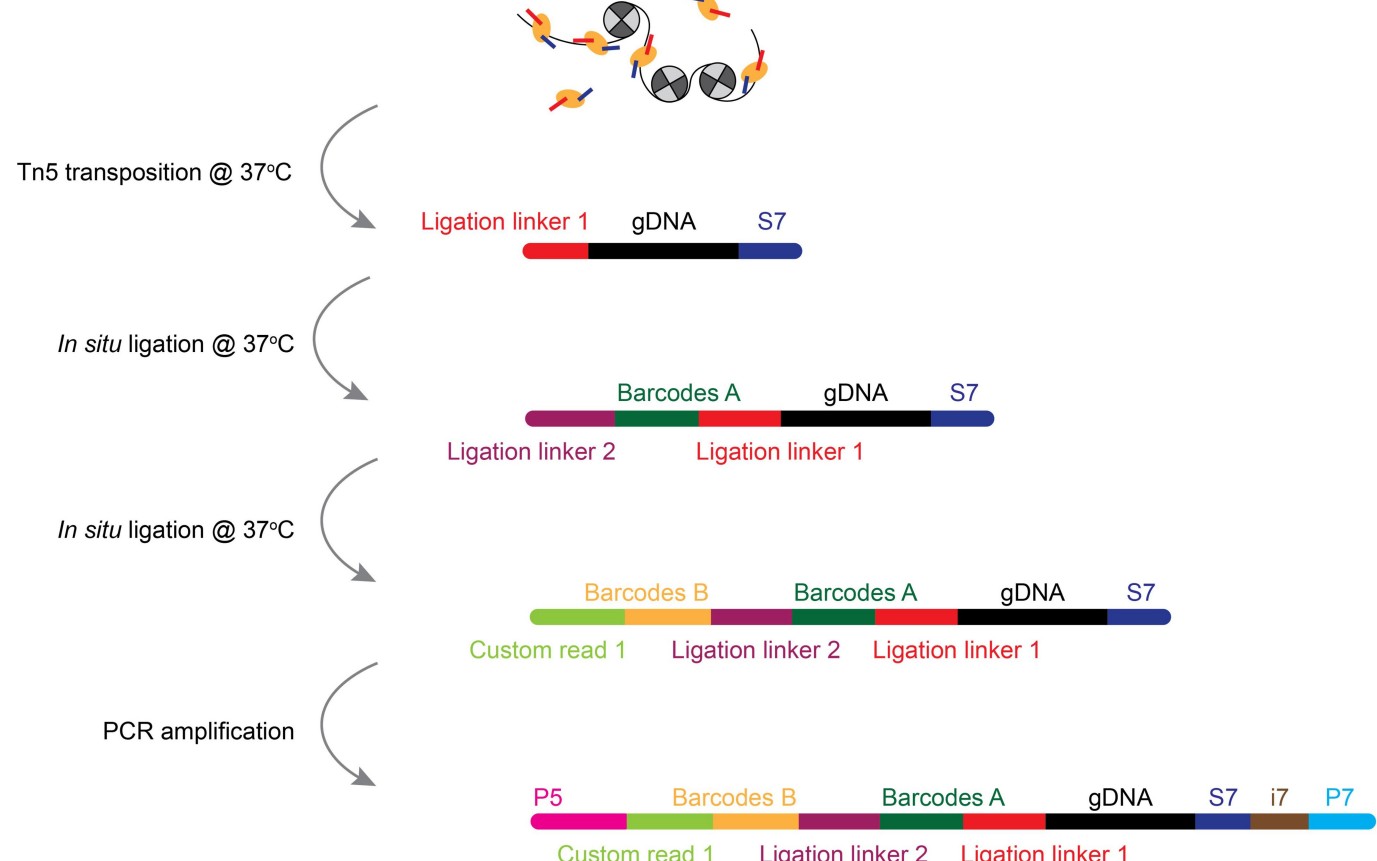

Tn5 transposition @ 37°C

Ligation linker 1    gDNA    S7

*In situ* ligation @ 37°C

Barcodes A    gDNA    S7
Ligation linker 2    Ligation linker 1

*In situ* ligation @ 37°C

Barcodes B    Barcodes A    gDNA    S7
Custom read 1    Ligation linker 2    Ligation linker 1

PCR amplification

P5    Barcodes B    Barcodes A    gDNA    S7    i7    P7
Custom read 1    Ligation linker 2    Ligation linker 1

**Extended Data Fig. 1 | Chemistry workflow of spatial-ATAC-seq.** A tissue
section on a standard aminated glass slide was lightly fixed with formaldehyde.
Then, Tn5 transposition was performed at 37 °C, and the adapters containing
ligation linker 1 were inserted to the cleaved genomic DNA at transposase
accessible sites. Afterwards, a set of DNA barcode A solutions were introduced
by microchannel-guided flow delivery to perform in situ ligation reaction for
appending a distinct spatial barcode Ai (i = 1–50) and ligation linker 2. Then,
a second set of barcodes Bj (j = 1-50) were introduced using another set of
microfluidic channels perpendicularly to those in the first flow barcoding step,
which were subsequently ligated at the intersections, resulting in a mosaic of
tissue pixels, each containing a distinct combination of barcodes Ai and Bj
(i = 1–50, j = 1–50). After DNA fragments were collected by reversing cross-
linking, the library construction was completed during PCR amplification.

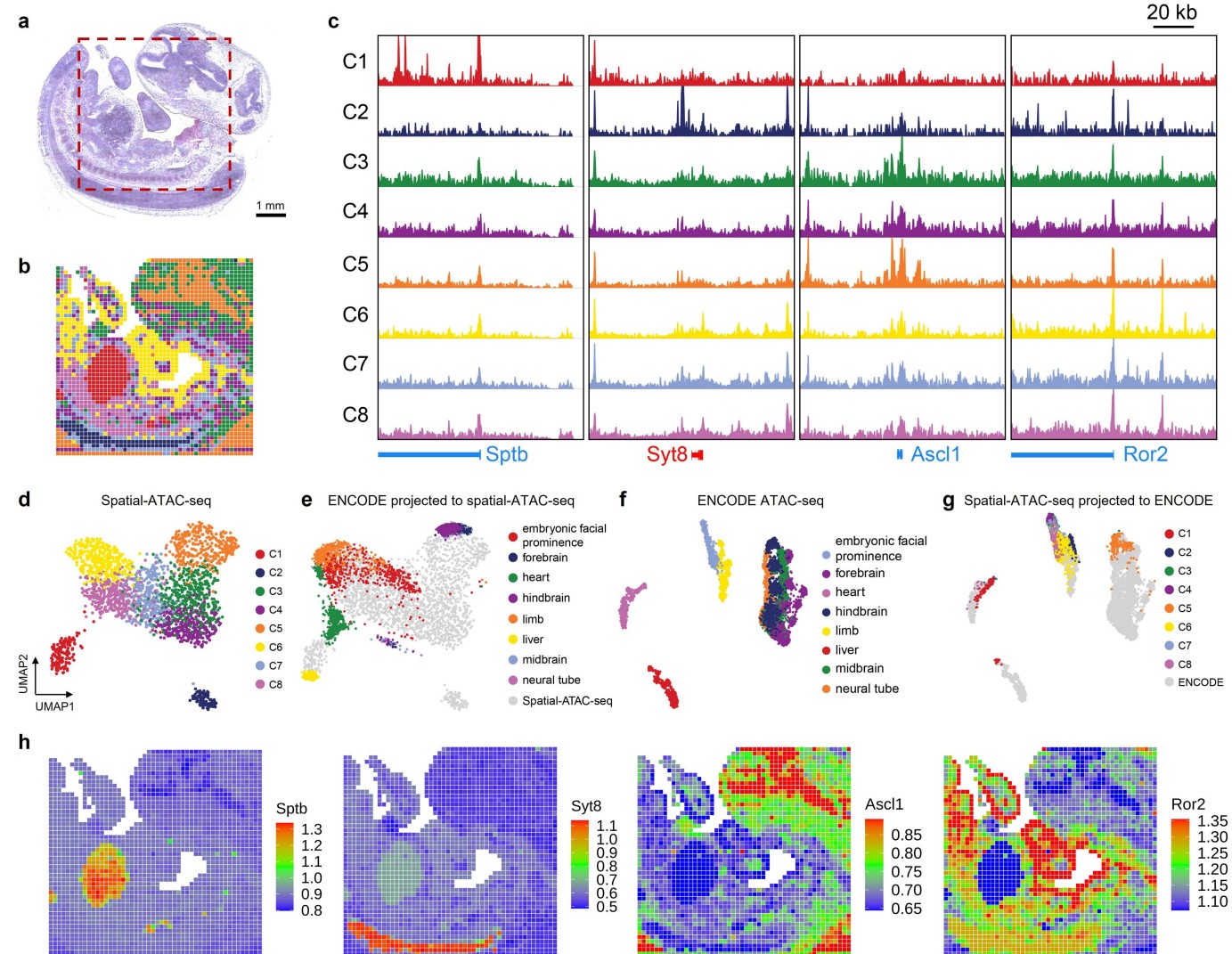

**Extended Data Fig. 2 | Further analysis of spatial chromatin accessibility mapping of E13 mouse embryo and validation with ENCODE reference data.** **a**, H&E image from an adjacent tissue section and a region of interest for spatial chromatin accessibility mapping (50 μm pixel size). **b**, Unsupervised clustering analysis and spatial distribution of each cluster. For better visualization, we scaled the size of the pixels. **c**, Genome browser tracks of selected marker genes in different clusters. **d**, UMAP embedding of unsupervised clustering analysis for spatial-ATAC-seq. Cluster identities and colouring of clusters are consistent with (**b**). **e**, LSI projection of ENCODE bulk ATAC-seq data from diverse cell types of the E13.5 mouse embryo dataset onto the spatial ATAC-seq embedding. **f**, UMAP embedding of unsupervised clustering analysis for ENCODE bulk ATAC-seq data from diverse cell types of the E13.5 mouse embryo dataset. **g**, LSI projection of spatial-ATAC-seq data onto ENCODE bulk ATAC-seq embedding. **h**, Spatial mapping of gene scores for selected marker genes in different clusters.

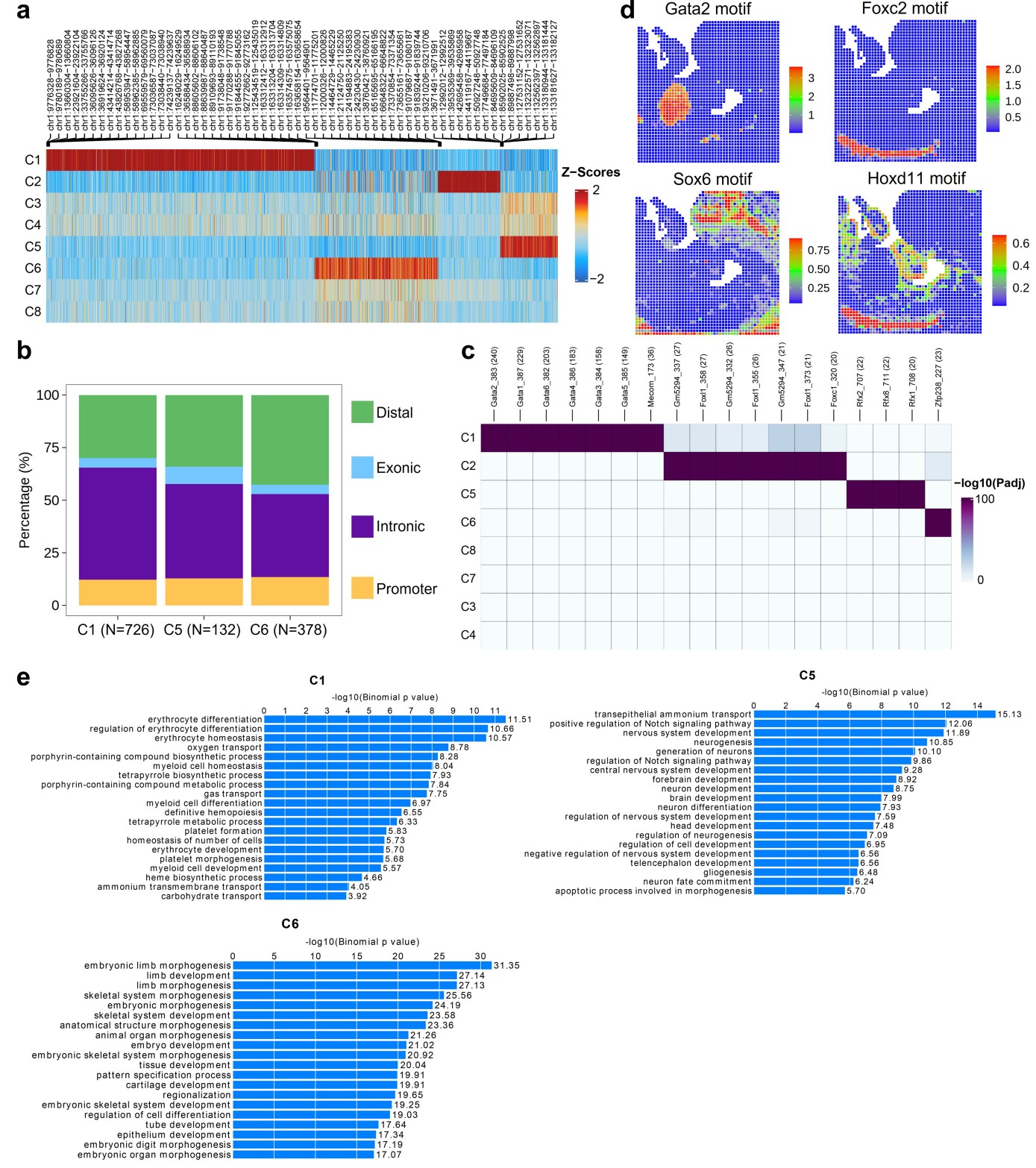

**Extended Data Fig. 3 | Motif enrichment analysis of the E13 mouse embryo data. a**, Heatmap of spatial-ATAC-seq marker peaks across all clusters identified with bias-matched differential testing. **b**, Annotation of marker peaks across clusters. **c**, Heatmap of motif hypergeometric enrichment-adjusted P values within the marker peaks of each cluster. **d**, Spatial mapping of selected TF motif deviation scores. **e**, GREAT enrichment analysis of marker peaks across clusters.

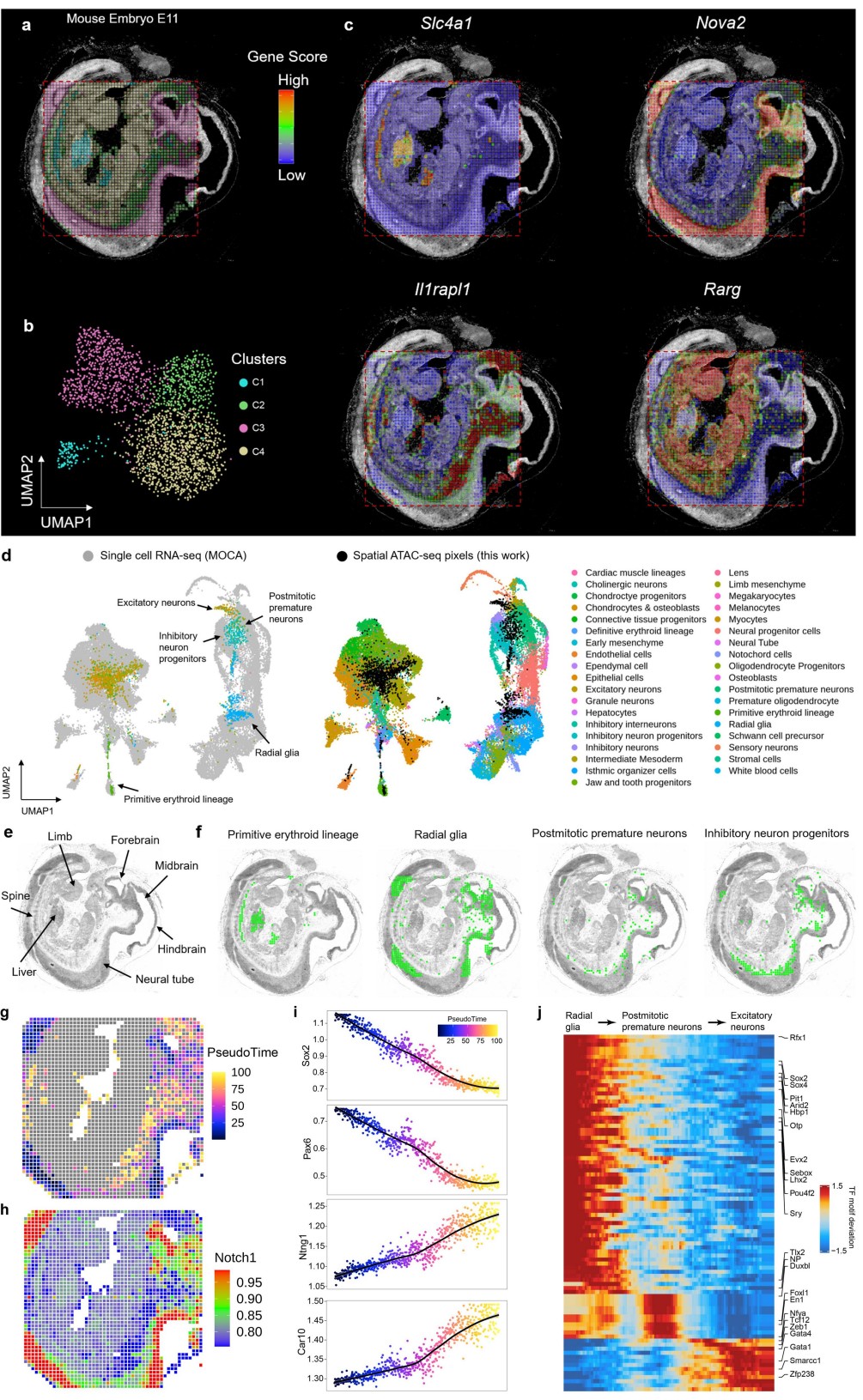

**Extended Data Fig. 4** | See next page for caption.

**Extended Data Fig. 4 | Spatial chromatin accessibility mapping of E11 mouse embryo and spatiotemporal analysis (50 μm pixel size). a**, Unsupervised clustering analysis and spatial distribution of each cluster. Overlay with the tissue image reveals that the spatial chromatin accessibility clusters precisely match the anatomic regions. For better visualization, we scaled the size of the pixels. **b**, UMAP embedding of unsupervised clustering analysis for chromatin accessibility. Cluster identities and colouring of clusters are consistent with (**a**). **c**, Spatial mapping of gene scores for selected marker genes in different clusters and the chromatin accessibility at select genes are highly tissue specific.

**d**, Integration of scRNA-seq from E11.5 mouse embryos[20] and spatial ATAC-seq data. Unsupervised clustering of the combined data was coloured by different cell types. **e**, Anatomic annotation of major tissue regions based on the H&E image. **f**, Spatial mapping of selected cell types identified by label transferring from scRNA-seq to spatial ATAC-seq data. **g**, Pseudotemporal reconstruction from the developmental process from radial glia to excitatory neurons plotted in space. **h**, Spatial mapping of gene scores for *Notch1*. **i**, dynamics for selected gene score along the pseudo-time shown in (**g**). **j**, Pseudo-time heatmap of TF motifs changes from radial glia to excitatory neurons.

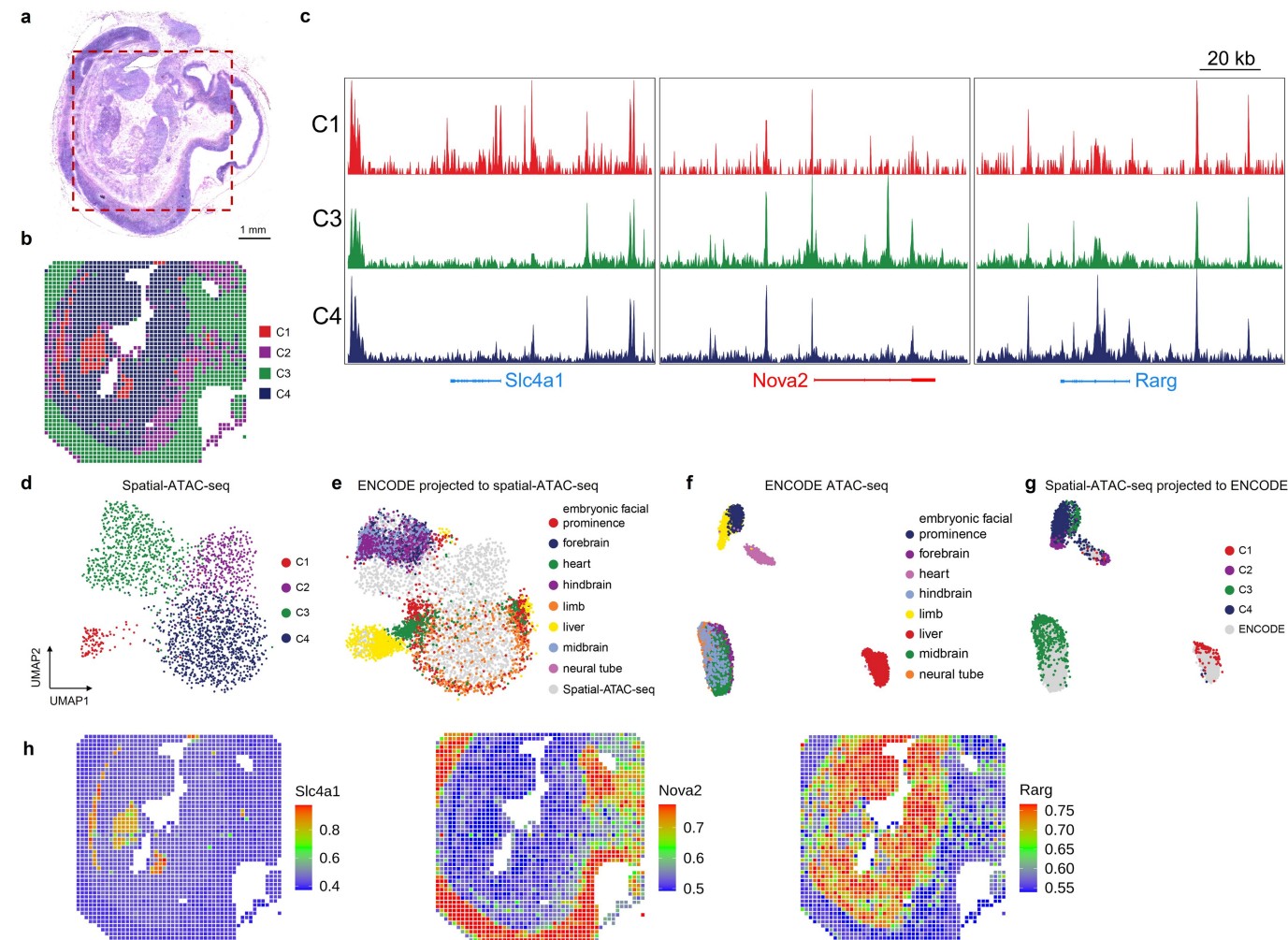

**Extended Data Fig. 5 | Further analysis of spatial chromatin accessibility mapping of E11 mouse embryo (50 μm pixel size) and validation with the ENCODE reference data. a**, H&E image from an adjacent tissue section and a region of interest for spatial chromatin accessibility mapping. **b**, Unsupervised clustering analysis and spatial distribution of each cluster. For better visualization, we scaled the size of the pixels. **c**, Genome browser tracks of selected marker genes in different clusters. **d**, UMAP embedding of unsupervised clustering analysis for spatial ATAC-seq. Cluster identities and colouring of clusters are consistent with (**b**). **e**, LSI projection of ENCODE bulk ATAC-seq data from diverse cell types of the E11.5 mouse embryo dataset onto the spatial-ATAC-seq embedding. **f**, UMAP embedding of unsupervised clustering analysis for ENCODE bulk ATAC-seq data from diverse cell types of the E11.5 mouse embryo dataset. **g**, LSI projection of spatial ATAC-seq data onto ENCODE bulk ATAC-seq embedding. **h**, Spatial mapping of gene scores for selected marker genes in different clusters.

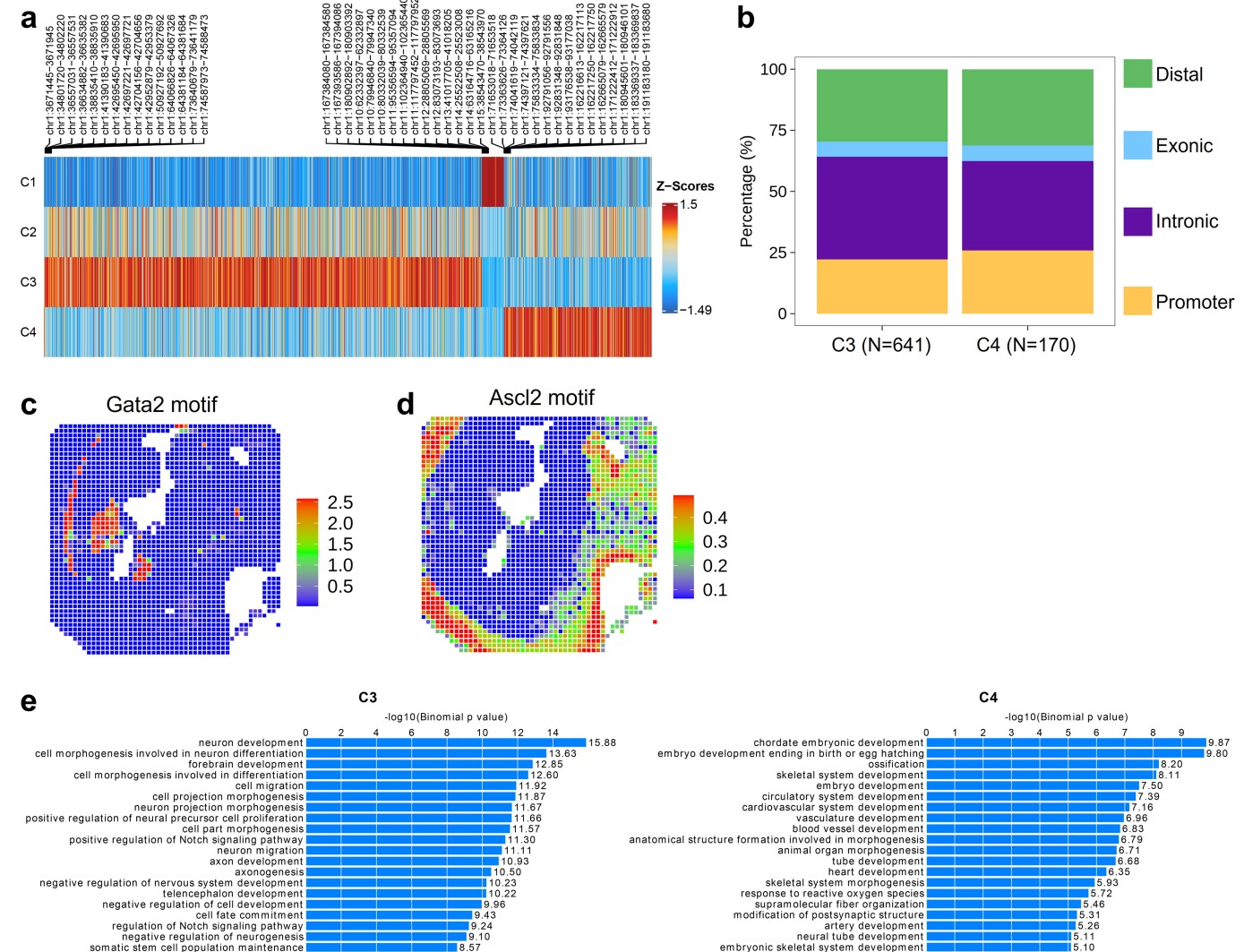

**Extended Data Fig. 6 | Motif enrichment analysis in E11 mouse embryo.**
**a**, Heatmap of spatial ATAC-seq marker peaks across all clusters identified with bias-matched differential testing. **b**, Annotation of marker peaks across clusters. **c, d**, Spatial mapping of selected TF motif deviation scores. **e**, GREAT enrichment analysis of marker peaks across clusters.

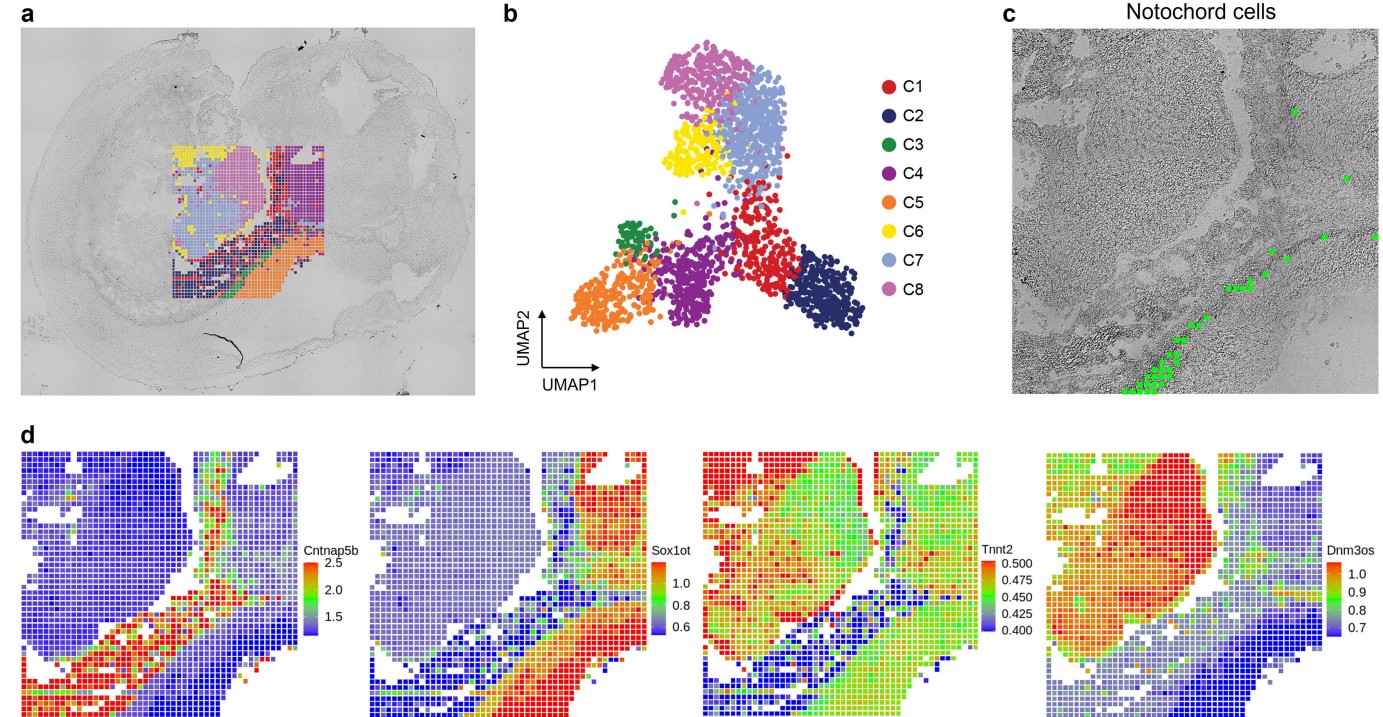

**Extended Data Fig. 7 | Spatial chromatin accessibility mapping of E11 mouse embryo and spatiotemporal analysis (20 μm pixel size). a**, Unsupervised clustering analysis and spatial distribution of each cluster. Overlay with the tissue image reveals that the spatial chromatin accessibility clusters precisely match the anatomic regions. For better visualization, we scaled the size of the pixels. **b**, UMAP embedding of unsupervised clustering analysis for chromatin accessibility. Cluster identities and colouring of clusters are consistent with (**a**). **c**, Spatial mapping of notochord cells identified by label transferring from scRNA-seq (E11.5 mouse embryos[20]) to spatial ATAC-seq data. **d**, Spatial mapping of gene scores for selected marker genes in different clusters and the chromatin accessibility at select genes are highly tissue specific.

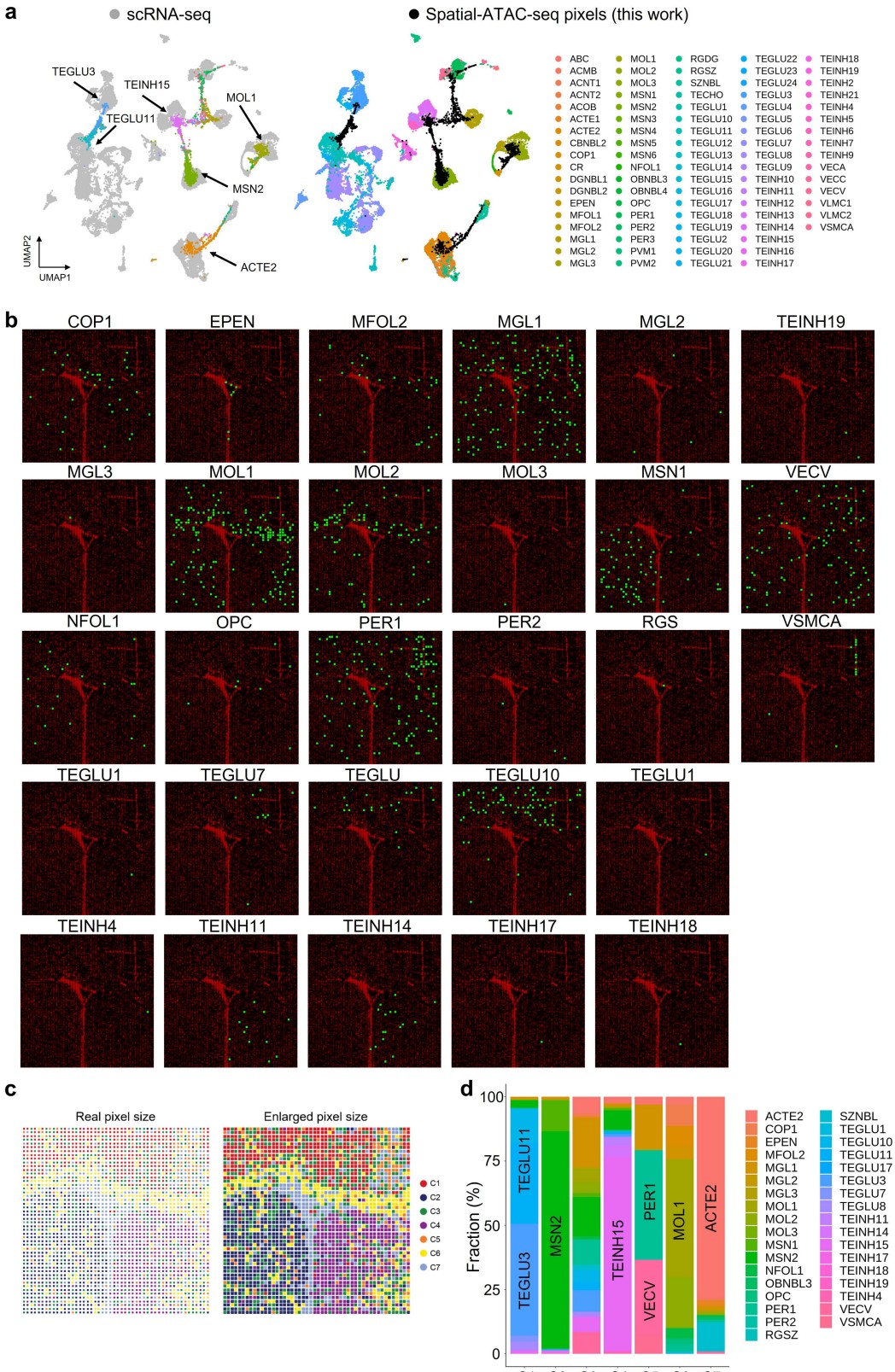

**Extended Data Fig. 8 | Integrative analysis of spatial-ATAC-seq and scRNA-seq for P21 mouse brain. a**, Integration of scRNA-seq from mouse brains[23] and spatial-ATAC-seq data. **b**, Spatial mapping of cell types identified by label transfer from scRNA-seq[23] to spatial-ATAC-seq. **c**, Spatial distribution of each cluster in the mouse brain with real (left) and enlarged (right) pixel size. **d**, Fraction of cell types in each spatial-ATAC-seq cluster. A list of abbreviation definitions can be found in Supplementary Table 2.

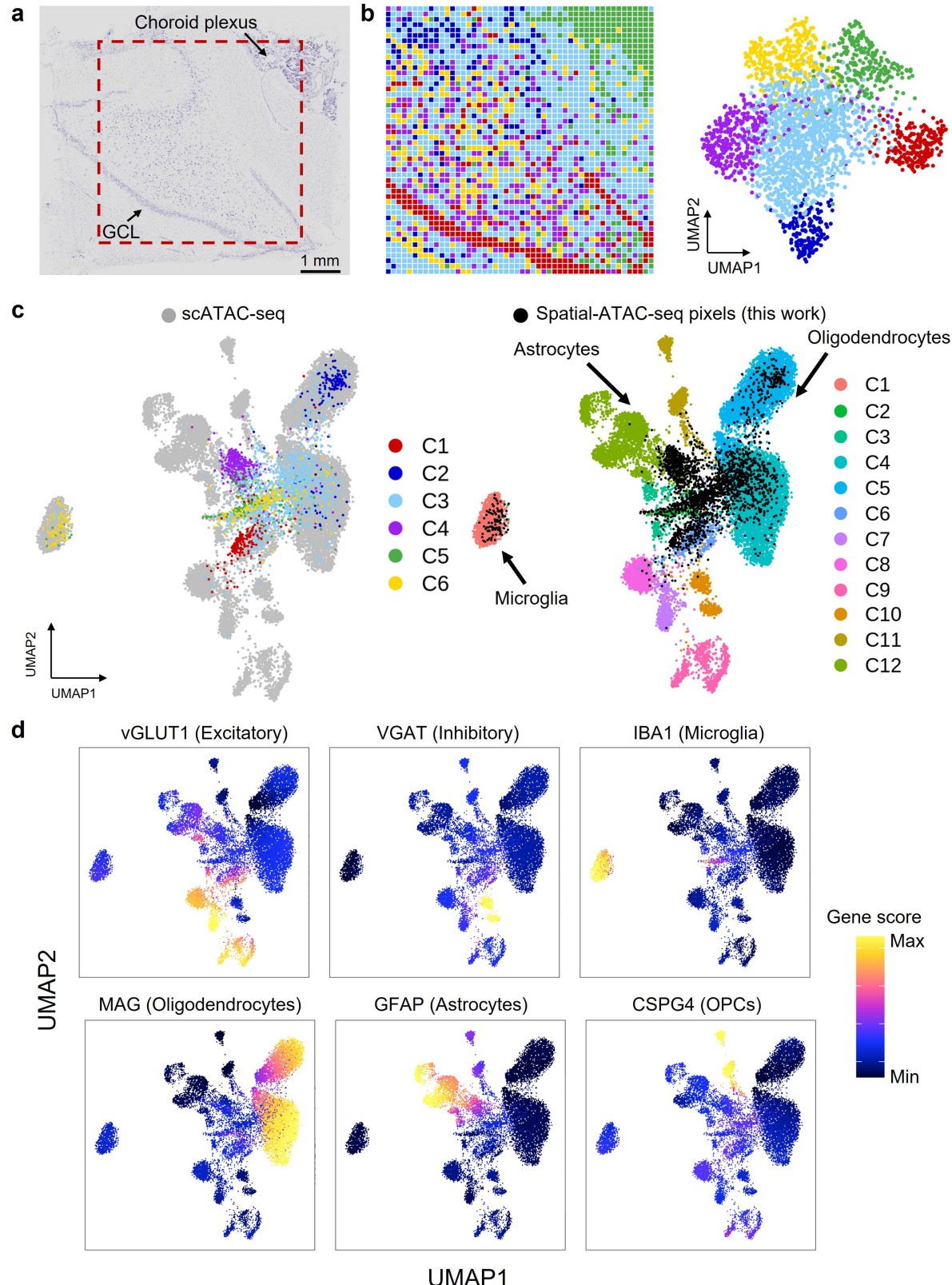

**Extended Data Fig. 9 | Spatial chromatin accessibility mapping of human hippocampus. a**, Nissl-stained tissue section adjacent to the one used for spatial chromatin accessibility mapping and region of interest for spatial-ATAC-seq (50 μm pixel size). **b**, Unsupervised clustering analysis and spatial distribution of each cluster. For better visualization, we scaled the size of the pixels. **c**, Integration of scATAC-seq[31] from human hippocampus and spatial-ATAC-seq. Colouring of spatial-ATAC-seq is consistent with (**b**). **d**, Co-embedding spatial-ATAC-seq and scATAC-seq datasets, coloured by gene score for the annotated lineage-defining gene[31].

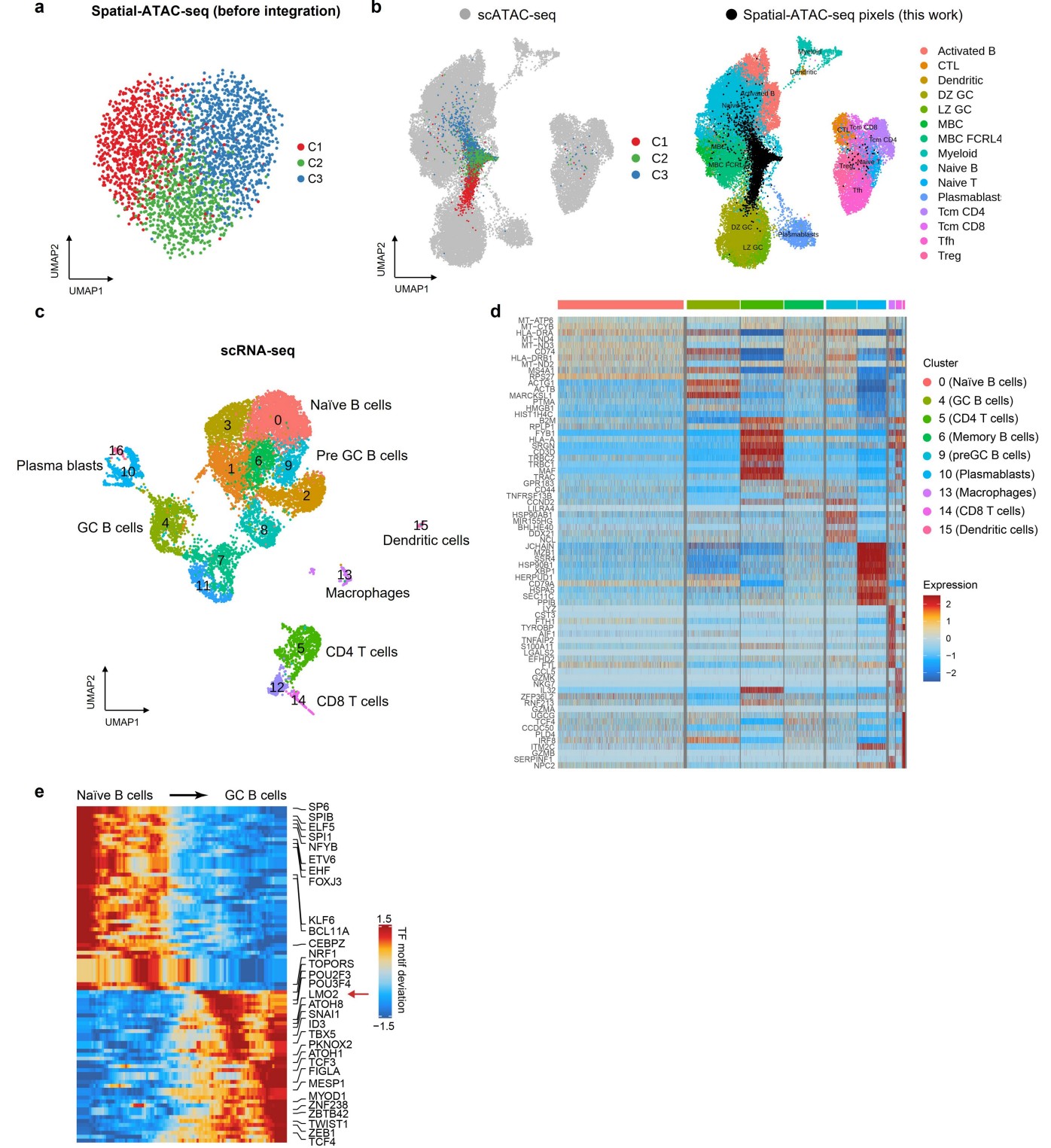

**Extended Data Fig. 10 | Further analysis of spatial chromatin accessibility mapping of human tonsil. a**, UMAP of tonsillar spatial-ATAC-seq data. **b**, Integration of scATAC-seq[48] from human tonsils and spatial-ATAC-seq.

**c**, UMAP of tonsillar immune scRNA-seq reference data[48]. **d**, Heatmap comparing key marker gene expression across selected immune cell types. **e**, Pseudo-time heatmap of TF motifs changes from Naïve B cells to GC B cells.

# Reporting Summary

Nature Research wishes to improve the reproducibility of the work that we publish. This form provides structure for consistency and transparency in reporting. For further information on Nature Research policies, see our Editorial Policies and the Editorial Policy Checklist.

## Statistics

For all statistical analyses, confirm that the following items are present in the figure legend, table legend, main text, or Methods section.

| n/a | Confirmed | |
|---|---|---|
| ☐ | ☒ | The exact sample size (*n*) for each experimental group/condition, given as a discrete number and unit of measurement |
| ☐ | ☒ | A statement on whether measurements were taken from distinct samples or whether the same sample was measured repeatedly |
| ☐ | ☒ | The statistical test(s) used AND whether they are one- or two-sided *Only common tests should be described solely by name; describe more complex techniques in the Methods section.* |
| ☐ | ☒ | A description of all covariates tested |
| ☐ | ☒ | A description of any assumptions or corrections, such as tests of normality and adjustment for multiple comparisons |
| ☐ | ☒ | A full description of the statistical parameters including central tendency (e.g. means) or other basic estimates (e.g. regression coefficient) AND variation (e.g. standard deviation) or associated estimates of uncertainty (e.g. confidence intervals) |
| ☐ | ☒ | For null hypothesis testing, the test statistic (e.g. *F*, *t*, *r*) with confidence intervals, effect sizes, degrees of freedom and *P* value noted *Give P values as exact values whenever suitable.* |
| ☒ | ☐ | For Bayesian analysis, information on the choice of priors and Markov chain Monte Carlo settings |
| ☒ | ☐ | For hierarchical and complex designs, identification of the appropriate level for tests and full reporting of outcomes |
| ☐ | ☒ | Estimates of effect sizes (e.g. Cohen's *d*, Pearson's *r*), indicating how they were calculated |

*Our web collection on statistics for biologists contains articles on many of the points above.*

## Software and code

Policy information about availability of computer code

| Data collection | EVOS FL Auto Software (REV 32044), Illumina HiSeq 4000 System, Illumina NovaSeq 6000 System. |
|---|---|
| Data analysis | R 3.6.1, python 3.7, ArchR 1.0.1, Seurat 3.2.3, clusterProfiler 3.16.1, GREAT 4.0.4, Cell Ranger ATAC v1.2, Snakemake v5.28.0, Adobe Illustrator v25.4.3, Fiji ImageJ 1.53q |
| | Scripts for data analysis were written in R and python with code available at https://github.com/dyxmvp/Spatial_ATAC-seq and https://github.com/rongfan8/DBiT-seq |

For manuscripts utilizing custom algorithms or software that are central to the research but not yet described in published literature, software must be made available to editors and reviewers. We strongly encourage code deposition in a community repository (e.g. GitHub). See the Nature Research guidelines for submitting code & software for further information.

## Data

Policy information about availability of data

All manuscripts must include a data availability statement. This statement should provide the following information, where applicable:
- Accession codes, unique identifiers, or web links for publicly available datasets
- A list of figures that have associated raw data
- A description of any restrictions on data availability

Raw and processed data reported in this paper are deposited in the Gene Expression Omnibus (GEO) with accession code GSE171943. Resulting fastq files were aligned to the mouse reference genome (mm10) or human reference genome(GRCh38). Published data for data quality comparison and integrative data analysis include Flash frozen cortex, hippocampus, and ventricular zone from embryonic mouse brain (E18) (https://www.10xgenomics.com/resources/datasets/flash-frozen-cortex-hippocampus-and-ventricular-zone-from-embryonic-mouse-brain-e-18-1-standard-1-2-0 ), ENCODE mouse embryo ATAC-seq (11.5 days) (https://

# Field-specific reporting

Please select the one below that is the best fit for your research. If you are not sure, read the appropriate sections before making your selection.

☒ Life sciences ☐ Behavioural & social sciences ☐ Ecological, evolutionary & environmental sciences

For a reference copy of the document with all sections, see nature.com/documents/nr-reporting-summary-flat.pdf

# Life sciences study design

All studies must disclose on these points even when the disclosure is negative.

| | |
|---|---|
| Sample size | No directly relevant. No sample size calculation was performed. Samples sizes were chosen primarily based on experiment length, sample availability, and sequencing costs. The current manuscript mainly described a new method for profiling spatially resolved chromatin accessibility and the sample sizes are sufficient because each sample serves as a proof-of-concept for the new technology. |
| Data exclusions | No data were excluded from the study. |
| Replication | All attempts at replication was successful. For E13 mouse embryo, two replicates have been done on adjacent tissue sections to test the reproducibility of the new technology. Other experiments were performed once to serve as a proof-of-concept for the new technology. |
| Randomization | Randomization was not applicable because the focus of this paper is the development of a new method for profiling spatially resolved chromatin accessibility and did not involve allocating samples/organisms/participants into experimental groups. |
| Blinding | Blinding was not applicable because the focus of this paper is the development of a new method for profiling spatially resolved chromatin accessibility and did not involve group allocation, and by extension, blinding. |

# Reporting for specific materials, systems and methods

We require information from authors about some types of materials, experimental systems and methods used in many studies. Here, indicate whether each material, system or method listed is relevant to your study. If you are not sure if a list item applies to your research, read the appropriate section before selecting a response.

## Materials & experimental systems

| n/a | Involved in the study |
|---|---|
| ☒ | ☐ Antibodies |
| ☐ | ☒ Eukaryotic cell lines |
| ☒ | ☐ Palaeontology and archaeology |
| ☐ | ☒ Animals and other organisms |
| ☐ | ☒ Human research participants |
| ☒ | ☐ Clinical data |
| ☒ | ☐ Dual use research of concern |

## Methods

| n/a | Involved in the study |
|---|---|
| ☒ | ☐ ChIP-seq |
| ☒ | ☐ Flow cytometry |
| ☒ | ☐ MRI-based neuroimaging |

# Eukaryotic cell lines

Policy information about cell lines

| | |
|---|---|
| Cell line source(s) | NIH/3T3 cells were from ATCC. |
| Authentication | None of the cell lines were authenticated. |
| Mycoplasma contamination | Cell lines were not tested for Mycoplasma contamination. |
| Commonly misidentified lines (See ICLAC register) | No commonly misidentified cell lines were used. |

# Animals and other organisms

Policy information about studies involving animals; ARRIVE guidelines recommended for reporting animal research

Laboratory animals

The mouse line Sox10:Cre-RCE:LoxP (EGFP), on a C57BL/6xCD1 mixed genetic background, female was used for experiments on P21 mice.

Mice were kept in individually ventilated cages (IVC sealsafe GM500, Tecniplast) with hardwood bedding, nesting material, shredded paper, gnawing sticks and card box shelter. Cages were changed every other week. Mice received regular chew diet (either R70 diet or R34, Lantmännen Lantbruk, Sweden) and water using a water bottle that was changed weekly. Housing parameters including relative humidity, temperature, and ventilation were established following the European convention for the protection of vertebrate animals used for experimental and other scientific purposes treaty ETS 123. Specifically, consistent relative air humidity and temperature were set to 50% and 22°C, and the air quality was controlled with the use of stand-alone air handling units supplemented with HEPA filter. Monitoring of husbandry parameters is done using ScanClime (Scanbur) units. The following light/dark cycle was used: dawn 6:00–7:00, daylight 7:00–18:00, dusk 18:00–19:00, night 19:00–6:00.

Wild animals

No wild animals were used in the study.

Field-collected samples

No field collected samples were used in the study.

Ethics oversight

Experimental procedures on juvenile (P21) mice were conducted in accordance with the European directive 2010/63/EU, local Swedish directive L150/SJVFS/2019:9, Saknr L150 and Karolinska Institutet complementary guidelines for procurement and use of laboratory animals, Dnr 1937/03-640. The procedures described here were approved by Stockholms Norra Djurförsöksetiska nämnd, the local committee for ethical experiments on laboratory animals in Sweden, lic.nr. 1995/2019 and 7029/2020.

Note that full information on the approval of the study protocol must also be provided in the manuscript.

# Human research participants

Policy information about studies involving human research participants

Population characteristics

Age: 31, Gender: Male. The subject was free of neuropsychiatric illness, had clear neuropathological exam, negative brain toxicology for psychoactive drugs, medication and alcohol.

Recruitment

The subject selected was free of neuropsychiatric illness based on our validated psychological autopsy interview of the next of kin, died of sudden death (industrial accident) with short agonal state (that can affect brain oxygenation if prolonged), had short post-mortem interval (6.5 Hour), clear neuropathological exam, negative brain toxicology for psychoactive drugs, medication and alcohol, and good RNA quality (RNA integrity number 8.50). Biases were not applicable because the focus of this paper is the development of a new method for profiling spatially resolved chromatin accessibility and each sample serves as a proof-of-concept for the new technology.

Ethics oversight

All procedures of brain collection and autopsy were conducted with Institutional Review Board approvals and informed consent from the next of kin.

Note that full information on the approval of the study protocol must also be provided in the manuscript.

