## [Peer Review File · Nature]

Manuscript Title: Spatial profiling of chromatin accessibility in mouse and human tissues

Redactions – Third Party Material

Redactions – unpublished data

Reviewer Comments & Author Rebuttals

Reviewer Reports on the Initial Version:

Referees' comments:

Referee #1 (Remarks to the Author):

In the manuscript by Deng et al., the authors present a method for measuring spatially resolved chromatin accessibility in tissue sections on slides. They use an elegant method of combinatorially indexing pixels of the section after tagmentation using fabricated microfluidic devices to deliver barcoded oligos to specific locations on the tissue section. After establishing the technique, it is used to profile two mouse embryo sections and one human tonsil section. The authors also integrate their data with existing bulk ATAC-seq data and single cell RNA-seq (scRNA-seq) data and show good alignment with existing data sets. While the method presents a major additional resolution across which chromatin accessibility can be explored, there are some aspects of the manuscript that I think are in need of improvement. My detailed comments are below (broken into Major and Minor sections). My expertise is in single cell genomics, not developmental biology or immunology, so my comments are largely focused on the technique and the genomic aspects of the manuscript.

Major comments

-Lack of full exploration of chromatin data. The authors show that gene activity scores can be explored, there are some aspects of the manuscript that I think are in need of improvement. My detailed comments are below (broken into Major and Minor as rough proxies of gene expression, and evaluate motif enrichments, but many of the sections). My expertise is in single cell genomics, not developmental biology or immunology, features that are unique to chromatin accessibility data are not really emphasized. For example, my expertise is in single cell genomics, not developmental biology or immunology, features that are largely focused on the technique and the genomic aspects of the manuscript. Do not attempt to link these elements with the genes they might regulate. In either the embryo or tonsil data, I would love to see if some regulatory elements can be found that are only accessible at a certain distance to some fixed reference. I know that the radial glia and anterior/posterior pseudotime analyses sort of get at this, but I wanted to see how this plays out at the level of individual regulatory elements and potentially inferred interactions defined with a program like Cicero.

- No whole embryo single-cell ATAC-seq data to compare to. The authors do a nice job comparing their spatial-ATAC data to existing bulk ATAC-seq and scRNA-seq, but the most natural comparison would be to scATAC-seq collected on the same sample types. This would involve a fair amount of further experimentation, but the authors should either identify a published scATAC-seq on mouse embryos or generate some data to compare to directly.

- Lack of discussion about numbers of sections that can be profiled. scATAC-seq can profile

10,000+ cells in one experiment. Spatial-ATAC can handle 2500 pixels. This results in very different resolutions and throughput. Can multiple sections be done simultaneously? How do we get to the N that would be necessary to have fully fleshed out chromatin profiles of specific features? Or could one envision a situation where 10X ATAC is projected onto spatial-ATAC? I would like the authors to be more explicit about the use cases they envision with this technique in the future.

- Given the reliance on marker genes, it makes it unclear whether the same sorts of applications could be accomplished with scATAC-seq data projected onto ISH maps. In principle, one could theoretically use scATAC-seq to generate gene activity scores and then project those onto the same ISH maps shown in the manuscript. The manuscript would benefit from an explicit description of how spatial-ATAC is superior.

- Data quality somewhat overstated. Generating a new resolution for measuring chromatin accessibility creates an issue in terms of what to benchmark against. For the most part, the manuscript does a fine job integrating with some scATAC data, some bulk ATAC data and some scRNA data. However, the conclusion from those analyses for me is that spatial-ATAC does suffer some degradation of data quality and appears to miss some tissue types (e.g. heart). Given this, I don't think it's appropriate to characterize the data quality as "excellent" or "equivalent" in the manuscript.

Minor (by section)

- As a general note, it would be a great help to have line numbers and page numbers in the future.

"Main Text"

- I get the point the authors are trying to make in using the term "unbiased". But it's not really accurate to say that single cell sequencing defines states and types in an unbiased way. There's quite a bit of bias in all of it – the marker genes used are all from the literature. Dimensionality reduction can introduce biases. Transcripts can be difficult to capture for some genes. The method of isolation of single cells/nuclei (Fluidigm vs 10X vs SeqWell vs sciRNA-seq vs Smart-seq2, etc.) introduces a bias. Relevant to the second sentence too, tissue dissociation introduces many biases. It would be nice if more of this subtlety could make it into the introduction.

"Spatial chromatin accessibility sequencing design and workflow"

- Middle of first paragraph: The statement "ligated at the intersections" was a little vague. I assume that the authors mean that the A and B barcodes are hybridized with linkers and then ligated to the 5' end of the Tn5 oligo through successive rounds of ligation, but I'm not 100% sure that's what they mean.

- Figure 1a: The figures are small and hard to see the meaningful details. Printed out, it was hard to see the barcodes appended to the Tn5 oligos. And the orthogonal microfluidics

creating a 50x50 grid is impossible to see visually. Would be nice to demonstrate this pictorially in some way.

- Second paragraph of the same section: I agree that 10x is the best reference to compare to for QC. But obviously a pixel is a very different thing than a nucleus and so the authors should be more explicit about the limitations of these comparisons. On average, how many nuclei are there per 20um pixel in their tissue sections? Per 50um pixel? How many pixels covering tissue are lacking nuclei altogether? Can the authors estimate whether barcodes diffuse across pixel boundaries? Does this affect data quality/analysis?

- Similarly, it's not appropriate to say that the data are "equivalent" between spatial-ATAC-seq and scATAC-seq. There is a 2.5- to 7.5-fold reduction in mitochondrial reads for scATAC-seq relative to spatial-ATAC-seq. And there's a 30-60% improvement in TSS reads for scATAC-seq. There are clear advantages for spatial-ATAC, but I think the fair interpretation is that it comes at some cost to data quality.

- In the Methods, the authors mention that pixels not on tissue are removed. I would like to see some quality metrics for the "blank" pixels added to the supplement. Do you get reads back from these regions? Are there fewer? Do they come from peaks? Etc.?

- The authors mention performing correlation analysis of replicates, but how that is conducted is not defined. Please be explicit about how measures are quantitatively compared and what correlation metric is used.

- Figure 1c does not appear to be mentioned in the text until after Figure 1i. I would recommend mentioning it earlier or reorganizing Figure 1.

- Figure 1d-f: As above, I can't think of a better reference than scATAC-seq, but it's not totally fair to compare pixels to nuclei for complexity, signal-to-noise, etc. I would like the figures to specify that 10X represents cells while "Spatial mapping" represents pixels.

- Figure 1g-h: The lines are too thick and the colors are too hard to distinguish here. It makes it very difficult to assess the relative quality of data. Perhaps fewer lines could be shown in the main figure? I think V1 or V2 metrics could be moved to the Supplement. It looks like the TSS enrichment is significantly better for 10X here than spatial-ATAC. Also, two spatial-ATAC conditions have very low TSS enrichments, but I can't tell which ones based on the plot.

"Spatial chromatin accessibility mapping of E13 mouse embryo"

- Figure S3d – what is hsrATAC-seq?

- Figure S3e (and Figure S9e) – 20kb seems too big to see meaningful data given the size of the plots. Consider making the distance covered smaller or stretching the plots out horizontally.

- For projection of ENCODE ATAC-seq onto spatial-ATAC UMAPs, please be explicit that the data is being subsampled in "pseudo single cells". How many? There seem to be some lack of overlap between bulk data and single-cell data. The authors should comment on this in the manuscript. Also, could the authors please include a supplementary table that details the exact data sets downloaded?

- For the UMAP, it is clear that the resolution isn't great even for quite diverse cell types.

Having 10X scATAC data on embryos would really help to identify if this is a limitation of N or a limitation of pixel size or a limitation of developmental stage.

- As the concept of marker genes are introduced in the main text, the statistical significance is not discussed. Are the genes called out in the text listed because they were known in the literature first? Or were they identified as specifically active in individual clusters? It seems like genes are mentioned for both reasons at various points. The authors could be more explicit about whether a marker gene is being discussed because it was identified as differential between clusters or because it was pulled from the literature. I would add what significance threshold is used to define something as a marker to the main text. How many genes are identified as markers for each cluster? How many go up/down?

- Instead of using GO on marker genes from activity scores, what happens if the authors take marker peaks and look for enrichments with a program like GREAT?

- The liver subclusters are not visually compelling in Figure S3g and it is unclear to me how the marker genes associate with the subclusters. How were the 2 subclusters defined? The text does not specify. Similarly, for the spinal A/P data, the correlations seem muted. Would any of these survive multiple testing corrections?

- For the second paragraph of the section, one of the truly unique things that ATAC-seq can provide are the locations of distal regulatory elements. Unfortunately, Figure S7a doesn't even get specifically mentioned in the main text. How many of the elements that define spatial clusters are proximal to TSSs? Distal? What other features are they associated with? Could the authors build maps linking distal elements to the genes they regulate with a program like Cicero? Or are the data too sparse? Can the authors see A/B compartment switching between clusters? Are there any other features of the chromatin that wouldn't be directly inferable from an analogous RNA map?

- Also, in identifying enriched motifs in this section (and elsewhere), there is no discussion of statistical rigor. Did the authors just look at the most enriched motif? Was it significant after multiple testing correction? Were there other motifs for each cluster enriched/depleted?

- In the third paragraph, the statement on erythroid cells in the liver seems based on qualitative assignments of anatomical features and cell type projections, so I would recommend softening the language from "were exclusively enriched" to "appeared predominantly clustered" or something along those lines.

- Also, the authors mention cells not identifiable in E11, but the E11 data has not been introduced yet.

- For the pseudotime analysis of neuron development, it would be worthwhile to include individual peaks that significantly change accessibility over pseudotime in addition to marker genes and TF motifs. Or are the data too sparse for this analysis as well?

"Spatial chromatin accessibility mapping of E11..."

- What happens when visualizing E11 and E13 spatial-ATAC data together with UMAP?

- It's true that the AGM does pop out, but many other features of "fine structure" do not. It would be worth exploring why or softening the language about how fine a feature that can

be identified.

- Second paragraph: again, some statistical rigor is needed here. How many sites are differential between clusters? What pathways come out with a region-based enrichment analysis like GREAT instead of GO?

- The authors state that a higher proportion of radial glia are identified in E11, but not E13. But what are the proportions. Without replicates to identify if this is significantly different, it's a hard statement to evaluate. It might be worth removing from the text in that case.

- For the fourth paragraph of this section, the authors identify differentially accessible elements between E11 and E13 in the liver. In addition to motifs, are these elements distal/proximal/etc.? Figure 3l and O provide sites opening and closing, but the volcano plots look strange. 3o in particular looks not properly centered for Fold-change. The authors might want to explore why the nadir of the volcano is not at Log2FC 0. In 3m and 3p, I don't see lines connecting individual motifs to individual points. Do all the other genes need to be listed on those plots? The authors should highlight the individual point that represents the highlighted motifs in both.

“Spatial chromatin accessibility mapping of human tonsil and immune cell states”

- I see clusters defined, but I don't see a UMAP for tonsil pixels. Did I miss it?

- When discussing enrichments of accessibility for certain genes in GC regions, please quantify and give a sense of the statistical significance again.

- Also, as with the embryos, it would be helpful to know if markers are defined as significant markers for UMAP clusters, or were pulled from the literature, or identified some other way. This is not always clear in the text.

- Last paragraph on B cell pseudotime maturation in GCs. This strikes me as potentially the strongest example in favor of spatial-ATAC-seq. This is the model that might allow for dynamic Cicero models of distinct modules looping in to regulate target genes in order and would be something that you really could not get easily with another technology. Have the authors tried something like that? Even just showing distinct elements around several genes of interest that open or close in pseudotime would be a very compelling case for spatial-ATAC. Also, might be worth considering analogously if distance from GC edge (inside or outside of it), leads to different models of gene regulation from the perspective of individual regulatory elements.

Discussion

- I think it's inappropriate to call the data quality “excellent”. As I've said above, I think the data quality are good with this assay, but it seems very clear to me that they are not quite as good as single cell or bulk.

- I'd like the Discussion to include consideration of other limitations of the technique (such as throughput, the difference between a pixel and a cell, the potential of partial nuclei in pixels, the consequences of multiple nuclei in pixels).

- I'd also like the Discussion to include unique future advantages of having chromatin

accessibility (instead of RNA or protein) at pixel resolution (instead of single cells).

Methods

- I noticed that LSI was run with “sampleCells” set to 10,000. But I assume that only 2500 pixels were analyzed for each experiment. Do I have that right?

Referee #2 (Remarks to the Author):

In their manuscript, Deng et al present a technique for spatially resolved sequencing of accessible chromatin, by combining a microfluidic device the authors previously invented and used in RNA sequencing, with Tn5 transposase technology. It's exciting that the technique is able to deliver spatially resolved open chromatin profiles (though the exact spatial resolution remains unclear), and the number of unique fragments per pixel is substantial, making the method of potential use in a variety of biological settings. While an intriguing method, its characterization seems rather cursory. The authors should have gone into significantly more depth to characterize their method, and clearly present its strengths, weaknesses, and known sources of technical noise.

1) Although clearly spatial information is retained in this technique (it is visible by eye), there is no experimental or computational analysis that definitively proves the spatial resolutions claimed by the channel sizes used (visual inspection of expression is not quantitative). A quantitative assessment of this spatial resolution is important, since there are clearly histological features that are lost when comparing the images in, for example, Fig S4. Can the authors look at the features of the spinal cord, compute the thickness of those features as represented by ATAC signal of, say, Sox10 and/or Nkx6-1, and provide a quantitative assessment of resolution? Can this be done with both the 20 micron and 50 micron devices, on serial sections, so that the reader can clearly compare and contrast the feature thicknesses for these two resolutions, and determine whether a smaller channel size really does improve spatial resolution?

2) Relatedly, I am confused about the separation between pixels—I apologize if this was explained somewhere and I missed it, but shouldn't the pixels be separated by a certain distance, namely the channel thickness? As such, wouldn't the expectation be that these pixels are separated by some amount of dead space? Has this separation been included for the visualizations shown in these figures? What are the absolute size dimensions, in microns, of the pixels shown in the ATAC plots?

3) As far as I can tell, the manuscript contains data from six datasets in total—four E11 experiments, one E13, and one Tonsil (looking at Figs 1g and S2). Can the authors provide

some information about throughput of the method? Is it conceivable that one could generate a full 3-D volume of the embryo, or is one limited in some way experimentally?

4) Figures 2 and 3 seem rather repetitive. What is the point of presenting, for example, the single cell mapping and pseudotime analyses for both E11 and E13? Are we learning anything new about the method by looking at two different time points? I found the pseudotime shown in Fig 2g especially confusing—why are the limb regions displaying pseudotime loading for a trajectory that is labeled as “Radial glia -> postmitotic premature neurons -> excitatory neurons”? In displaying these pseudotime analyses, can the authors clearly label the expected spatial developmental trajectory, so that those who are unfamiliar with neural development can understand whether the method appropriately recapitulates what is biologically known?

5) I found the inclusion of “V1” data confusing. As far as I can tell, V1 is simply a lot worse in sensitivity than “V2.” Since this method has never been published before, why even mention “V1” or include this data? The number of unique fragments per pixel is impressive, and it seems to me that simply presenting the V2 data as the “spatial-ATAC-seq” method would be far clearer to readers.

Small questions/points:

- For the integrative analyses in Figs 2d and 3d, the authors should show all cell type mappings in extended data, rather than just the selected annotations seen in the 2f/3f panels.
- There is a lot of variability in the % mitochondrial fragments across replicates. Can the authors provide some explanation for why this might be?
- The authors manually remove pixels that are not overlying tissue. When assessing diffusion/resolution, it might be useful to retain those pixels to understand if and how signal is leaking across adjacent pixels.

Referee #3 (Remarks to the Author):

Deng et al., introduce ‘spatial-ATAC-seq’ a technology that profiles the chromatin accessibility of spot regions while simultaneously retaining the spatial location of the cell in the tissue. The design of this method cleverly combines the chemistry of ATAC-seq and microfluidic deterministic barcoding of the tissue that has been previously developed by the Fan lab; where a section of tissue is subjected to Tn5 transposition to profile the chromatin

accessibility followed by in-situ ligation of DNA barcodes that denote spatial location within the tissue. The authors demonstrate the applicability of spatial-ATAC-seq by applying it to 50 μm sections of E11 and E13 mouse embryos as well as to 20 μm sections of human tonsils. These datasets largely recapitulate known molecular features for each individual tissue types.

The novelty of this manuscript lies in the technological advancement to the field of spatial omics. Spatial-ATAC-seq, would be one of the first methodologies that adds an epigenetic layer to spatial technologies - and the first to do this systematically over a tissue section. The authors successfully demonstrate with proof-of-principle experiments that this methodology is powered to identify known biological features within each dataset. However, the authors do not show how this methodology can be leveraged to discover new biological phenomena. Overall, while the manuscript presents several pieces of interesting data, the authors do not make significant biological claims. Without the spatial component to the analysis, the findings and novelty of the manuscript is limited. The major concerns are:

1. In Figures 2 and 3 the authors identify a relatively low number of clusters given the complexity of the mouse fetus at the stage under study. The authors should characterize the resolution of the method relative to that of single-cell ATAC-Seq, for example relative to the ones detected in this atlas:

<https://science.sciencemag.org/content/370/6518/eaba7612.long>

2. In the discussion the authors state “We observed that a significant number of pixels (20 μm) contained single nuclei and the extraction of sequencing reads from these pixels can give rise to spatially-defined single-cell ATAC-seq data.” However, the data to support this claim about resolution is missing from the manuscript.

3. In Figure 1c, the authors compare the spatial-ATAC data with bulk ATAC data from the liver, could the authors show that the genomic region shown is specific to the liver and different from other tissue types?

4. From the pseudotime analysis in Figures 2g-h and 3g-h, the authors do not make a claim for any insight generated from the spatial component. Also, the methods section on pseudotime for figure is unclear. It seems likely that real insight could stem from this analysis, so we would encourage the authors to explore further what the data may be suggesting.

5. In Figures S3 and S9, the authors project the ENCODE ATAC-Seq onto the latent space defined by their spatial ATAC-Seq. However, since the resolution appears to be low for spatial ATAC-Seq data, it would be informative to also examine the opposite: projection of

the spatial ATAC-Seq data onto the ENCODE-defined latent space.

6. The authors conclude that there is a discordance between gene expression and chromatin accessibility: “CXCR4, which is expressed in the centroblasts in the GC dark zone [...]”. To make this claim the authors should explicitly show the evidence for a lack of gene expression as opposed to only showing the chromatin accessibility.

7. The authors do not discuss the applicability of the method. As part of the publication of the method, the authors should include a detailed protocol for how a lab with reasonable resources may be able to set it up.

Minor Critiques

1. In figure 1, in addition to the QC metrics shown in Figure 1, the authors should include a FRiP score (fraction of reads in peaks) as an indicator of ATAC-seq quality.

2. Citations of known biological phenomena are missing in a significant proportion of the manuscript. A few such places in the manuscript are cited below.

“Sptb, which plays a role in stability of erythrocyte membranes, was activated extensively in the liver. Syt8, which is important in neurotransmission, had a high level of gene activity in the spine. Ascl1 showed strong enrichment in the mouse brain, which is known to be involved in the commitment and differentiation of neuron and oligodendrocyte (Fig. 2c, Fig. S4e, f). Sox10 marks oligodendrocyte progenitor cells (OPCs). It was expressed at a high level in the dorsal root ganglia (DRGs), which are adjacent to the spinal cord (Fig. S4a, b). Olig2 is a marker of neural progenitors, pre-OPCs and OPCs. Olig2 is expressed in a small domain of the spinal cord, in the ventral domains of the forebrain, and in some posterior regions (brain stem, midbrain and hindbrain), which is consistent with the high gene score in the spatial ATAC-seq data (Fig. S4c, d).”

“Sox2, which is required for stem cell maintenance in the central nervous system, and Ntng1, which is involved in controlling patterning and neuronal circuit formation (Fig. 2h, i)”

“For example, Slc4a1, which are required for normal flexibility and stability of the erythrocyte membrane and for normal erythrocyte shape, were highly active in liver and AGM. Nova2, which is involved in RNA splicing or metabolism regulation in a specific subset of developing neurons, was highly enriched in the brain and neural tube.”

“Interestingly, it showed extensive open locus accessibility, suggesting extensive epigenetic priming of pre-GC T cells to potentially develop TFR function as needed to balance GC activity. CD25, a surface marker for regulatory T cells, was active in both GC and the extrafollicular zone. For non-lymphoid cells, CD11B, a macrophage marker, was inactive in GC, on contrast to CD11A, which was more active in GC lymphocytes. CD103 was enriched in GC follicular dendritic cells. CD144, which encodes vascular endothelial cadherin (VE-cadherin), corresponded to endothelial microvasculature near the crypt or between follicles. CD32, a surface receptor involved in phagocytosis and clearing of immune complexes, and

CD55, a complement decay- accelerating factor, were both active in the same region such that the cells not supposed to be cleared can be protected against phagocytosis by blocking the formation of the membrane attack complex.”

Author Rebuttals to Initial Comments:

We are very appreciative of all the constructive review comments, which resulted in a much improved manuscript. All major concerns from three reviewers have been fully addressed. Please find below, our point-by-point responses to all review comments. Thank you!

Referee #1 (Remarks to the Author):

In the manuscript by Deng et al., the authors present a method for measuring spatially resolved chromatin accessibility in tissue sections on slides. They use an elegant method of combinatorially indexing pixels of the section after tagmentation using fabricated microfluidic devices to deliver barcoded oligos to specific locations on the tissue section. After establishing the technique, it is used to profile two mouse embryo sections and one human tonsil section. The authors also integrate their data with existing bulk ATAC-seq data and single cell RNA-seq (scRNA-seq) data and show good alignment with existing data sets. While the method presents a major additional resolution across which chromatin accessibility can be explored, there are some aspects of the manuscript that I think are in need of improvement. My detailed comments are below (broken into Major and Minor sections). My expertise is in single cell genomics, not developmental biology or immunology, so my comments are largely focused on the technique and the genomic aspects of the manuscript.

Response: We would like to thank the reviewer for the positive feedback regarding our manuscript and work!

Major comments

- Lack of full exploration of chromatin data. The authors show that gene activity scores can serve as rough proxies of gene expression, and evaluate motif enrichments, but many of the features that are unique to chromatin accessibility data are not really emphasized. For example, there are at best cursory mentions of distal regulatory elements, and the authors do not attempt to link these elements with the genes they might regulate. In either the embryo or tonsil data, I would love to see if some regulatory elements can be found that are only accessible at a certain distance to some fixed reference. I know that the radial glia and anterior/posterior pseudotime analyses sort of get at this, but I wanted to see how this plays out at the level of individual regulatory elements and potentially inferred interactions defined with a program like Cicero.

Response: Thanks for the comments! According to this suggestion, we further explored the existing data for regulatory elements and linked these elements with the genes they might regulate using the addPeak2GeneLinks function in ArchR. We studied both embryo and tonsil data, and we found some distinct elements around several genes of interest that open or close in pseudotime or distinct spatial locations.

In the revision, we sought to exploit our spatial-ATAC-seq data to recover the spatially organized developmental trajectory and examine how developmental processes proceed across the tissue space. We studied the course of a developmental process from radial glia to excitatory neurons with postmitotic premature neurons as the immediate state after the radial glial differentiation and ordered these cells

in pseudo-time using ArchR. Spatial projection of each pixel's pseudo-time value revealed the spatially organized developmental trajectory in neurons (Please see below and also Fig. 2 in the revised manuscript). Interestingly, we observed that cells early in differentiation clustered around the ventricles in the developing brainstem, whereas those farther away exhibited a more differentiated phenotype (Please see below and also Fig. 2g). We also used the correlation of peak accessibility to predict the interactions between regulatory regions and found dynamically regulated promoter interactions with specific enhancers in genes, such as *Pou3f2* and *Nova2* (Please see below and also Fig. 2i). *Pou3f2*, also known as *Brn2*, encodes a transcription factor expressed in mouse in late progenitors and postmitotic neurons³⁹ and that has been shown to be important during neural development for the production of specific neuronal populations⁴⁰. Our analysis shows a reduction of chromatin accessibility at a specific *Brn2* enhancer during the transition from radial glial to postmitotic premature neurons, but not at other cis-regulatory regions, suggesting a role of this region in *Brn2* transcription. A similar decrease chromatin accessibility was observed in excitatory neurons for a specific intronic enhancer of *Nova2*, encoding an RNA binding protein expressed in neurons³⁸. Thus, our data indicated that spatial-ATAC-seq allows to map at a spatial level the chromatin accessibility dynamics at important regulatory regions during neural lineage commitment.

In addition, we implemented a pseudotemporal reconstruction of B cell activation to the GC reaction, where GC B cells were enriched inside GC and Naïve B cells are outside of it (Fig. 4f, g). The dynamic chromatin accessibility of some individual regulatory elements around several genes of interest were investigated. We identified putative target genes of fine-mapped autoimmune GWAS genetic variants, and revealed GC-specific regulatory potential, including at loci of major GC regulators such as *BCL6* (Please see below and Fig. 4i).

- No whole embryo single-cell ATAC-seq data to compare to. The authors do a nice job comparing their spatial-ATAC data to existing bulk ATAC-seq and scRNA-seq, but the most natural comparison would be to scATAC-seq collected on the same sample types. This would involve a fair amount of further experimentation, but the authors should either identify a published scATAC-seq on mouse embryos or generate some data to compare to directly.

Response: Thanks for the comments. Currently, there is no published scATAC-seq on mouse embryos. Therefore, we generated new spatial-ATAC-seq data from P21 mouse brain and human hippocampus, which have existing scATAC-seq data. We did data integration with scATAC-seq in mouse brain, human tonsil, and human hippocampus. Spatial tissue pixels were found to conform well into the clusters of scATAC-seq data (Please see below and also Fig. 3f, Fig. S18c, and Fig. S19b).

(1) Data integration with scATAC-seq in mouse brain

(2) Data integration with scATAC-seq in human tonsil

(3) Data integration with scATAC-seq in human hippocampus

- Lack of discussion about numbers of sections that can be profiled. scATAC-seq can profile 10,000+ cells in one experiment. Spatial-ATAC can handle 2500 pixels. This results in very different resolutions and throughput. Can multiple sections be done simultaneously? How do we get to the N that would be necessary to have fully fleshed out chromatin profiles of specific features? Or could one envision a situation where 10X ATAC is projected onto spatial-ATAC? I would like the authors to be more explicit about the use cases they envision with this technique in the future.

Response: Thanks for the comments. For current spatial-ATAC-seq protocol, each person can process two sections simultaneously in Day 1 experiment (i.e. 5000 pixels). However, in a week (5 days), we can run four Day 1 experiments and store the lysates in -20 °C, and then perform Day 2 experiment for these 8 lysates simultaneously.

To further increase the throughput, we can increase the number of barcodes (e.g. 100x100), such that 10,000 pixels can be handled. Additionally, a serpentine microfluidic channel design will enable spatial mapping of tissue array (please see the channel design below). We added “First, the number of mapping pixels of spatial-ATAC-seq could be further increased by increasing the number of barcodes (e.g. 100 × 100) or using a serpentine microfluidic channel design for spatial mapping of tissue array.” in the discussion of the revised manuscript.

(a) Serpentine microfluidic channel design for barcodes A1-A100

Image redacted

Image redacted

Image redacted

Indeed, we can integrate scATAC-seq data with spatial-ATAC-seq. We performed the data integration in mouse brain, human tonsil, and human hippocampus. Spatial tissue pixels were found to conform well into the clusters of scATAC-seq data (Please see above and also Fig. 3f, Fig. S18c, and Fig. S19b).

- Given the reliance on marker genes, it makes it unclear whether the same sorts of applications could be accomplished with scATAC-seq data projected onto ISH maps. In principle, one could theoretically use scATAC-seq to generate gene activity scores and then project those onto the same ISH maps shown in the manuscript. The manuscript would benefit from an explicit description of how spatial-ATAC is superior.

Response: Thanks for the comments. The marker genes are being discussed in the manuscript because they were identified as one of the top differential genes between clusters, and they were also known in the literature. It is true that scATAC-seq could be projected to ISH data, however, only the gene activity can be transferred, but we cannot project the scATAC-seq to the exact spatial location in the ISH maps. For spatial-ATAC-seq, we were able to obtain the exact spatial location of each pixel,

and could be correlated to cell morphology by using immunostaining (Please see Fig. 3 in the revised manuscript). Additionally, as the reviewer suggested, spatial-ATAC-seq could identify regulatory elements and linked these elements with the genes they might regulate, which cannot be achieved in ISH. Furthermore, tissue dissociation in single-cell technologies may preferentially select certain cell types or perturb cellular states as a result of the dissociation or other environmental stresses. For clarification, we have added “We developed spatial-ATAC-seq for spatially resolved and genome-wide profiling of chromatin accessibility in intact tissue sections with spatial information retained at cellular level (20 µm pixel size). Single-cell chromatin accessibility can also be derived without tissue dissociation by identifying pixels containing only one nucleus using immunofluorescence imaging.” in the discussion of the revised manuscript.

- Data quality somewhat overstated. Generating a new resolution for measuring chromatin accessibility creates an issue in terms of what to benchmark against. For the most part, the manuscript does a fine job integrating with some scATAC data, some bulk ATAC data and some scRNA data. However, the conclusion from those analyses for me is that spatial-ATAC does suffer some degradation of data quality and appears to miss some tissue types (e.g. heart). Given this, I don't think it's appropriate to characterize the data quality as “excellent” or “equivalent” in the manuscript.

Response: Thanks for the comments. In the revised manuscript, we removed these words, such as “excellent” or “equivalent”.

Minor (by section)

- As a general note, it would be a great help to have line numbers and page numbers in the future.

Response: Thanks for the suggestion. We have added line numbers and page numbers in the revised manuscript.

“Main Text”

- I get the point the authors are trying to make in using the term “unbiased”. But it's not really accurate to say that single cell sequencing defines states and types in an unbiased way. There's quite a bit of bias in all of it – the marker genes used are all from the literature. Dimensionality reduction can introduce biases. Transcripts can be difficult to capture for some genes. The method of isolation of single cells/nuclei (Fluidigm vs 10X vs SeqWell vs sciRNA-seq vs Smart-seq2, etc.) introduces a bias. Relevant to the second sentence too, tissue dissociation introduces many biases. It would be nice if more of this subtlety could make it into the introduction.

Response: Thanks for the comments. In the revised manuscript, we removed the term “unbiased”. In addition, we added “Furthermore, the method of isolation in single-cell technologies may preferentially select certain cell types or perturb cellular states as a result of the dissociation or other environmental stresses¹⁻³.”

“Spatial chromatin accessibility sequencing design and workflow”

- Middle of first paragraph: The statement “ligated at the intersections” was a little vague. I assume that the authors mean that the A and B barcodes are hybridized with linkers and then ligated to the 5' end of the Tn5 oligo through successive rounds of ligation, but I'm not 100% sure that's what they mean.

Response: Thanks for pointing out this confusion. We revised “ligated at the intersections” to “Barcodes A and B with linkers were ligated to the 5’ end of the Tn5 oligo through successive rounds of ligation.”

- Figure 1a: The figures are small and hard to see the meaningful details. Printed out, it was hard to see the barcodes appended to the Tn5 oligos. And the orthogonal microfluidics creating a 50x50 grid is impossible to see visually. Would be nice to demonstrate this pictorially in some way.

Response: Thanks for the suggestion. We have increased the text font in the revised Figure 1a, and we also added new Figure 1b to demonstrate the orthogonal microfluidics creating a 50x50 grid (Please see below and also Fig. 1a, b).

- Second paragraph of the same section: I agree that 10x is the best reference to compare to for QC. But obviously a pixel is a very different thing than a nucleus and so the authors should be more explicit about the limitations of these comparisons. On average, how many nuclei are there per 20um pixel in their tissue sections? Per 50um pixel? How many pixels covering tissue are lacking nuclei altogether? Can the authors estimate whether barcodes diffuse across pixel boundaries? Does this affect data quality/analysis?

Response: Thanks for the comments. According to this suggestion, we added “(spatial-ATAC-seq pixels may contain more than one nucleus, which is based on the tissue type and cell size)” in the revised manuscript. For E10 mouse embryo, in our previous paper⁴, on average there were 1-2 cells per 10µm pixel and 25 cells per 50µm pixel (Please see below, right panel). In addition, we have evaluated the possibility of DNA diffusion by analyzing a 3D fluorescence confocal image, which confirmed negligible leakage signal throughout the tissue section thickness⁴. It was found to be $0.9 \pm 0.2 \mu\text{m}$ for 10 µm channels and $4.5 \pm 1 \mu\text{m}$ for 50 µm channels, which validated spatially confined delivery and binding of DNA barcodes in tissue using microfluidics (Please see below, left panel).

Image redacted

- Similarly, it's not appropriate to say that the data are "equivalent" between spatial-ATAC-seq and scATAC-seq. There is a 2.5- to 7.5-fold reduction in mitochondrial reads for scATAC-seq relative to spatial-ATAC-seq. And there's a 30-60% improvement in TSS reads for scATAC-seq. There are clear advantages for spatial-ATAC, but I think the fair interpretation is that it comes at some cost to data quality.

Response: Thanks for the comments. In the revised manuscript, we removed these words, such as "excellent" or "equivalent".

- In the Methods, the authors mention that pixels not on tissue are removed. I would like to see some quality metrics for the "blank" pixels added to the supplement. Do you get reads back from these regions? Are there fewer? Do they come from peaks? Etc.?

Response: Thanks for the suggestion. We compared the number of unique fragments in pixels in tissue (E13 mouse embryo) and those not in tissue, and we found that pixels not in tissue had significantly fewer unique fragments (median is 8069) than pixels in tissue (median is 100786). In addition, ~ 7% of reads from the "blank" pixels were in peaks (Please see below and Fig. S2c, d in the revised manuscript).

- The authors mention performing correlation analysis of replicates, but how that is conducted is not defined. Please be explicit about how measures are quantitatively compared and what correlation metric is used.

Response: Thanks for the comments. Correlation analysis of replicates was conducted by calculating the Pearson correlation coefficient of chromatin accessibility profiles between pseudo-bulk spatial-ATAC-seq data, which was constructed from the tile matrix in ArchR. For clarification, we added "Correlation analysis of replicates was conducted by calculating the Pearson correlation coefficient of chromatin accessibility profiles between pseudo-bulk spatial-ATAC-seq data, which was constructed from the tile matrix in ArchR." in the Fig.S2 figure caption of the revised manuscript.

- Figure 1c does not appear to be mentioned in the text until after Figure 1i. I would recommend mentioning it earlier or reorganizing Figure 1.

Response: Thanks for the suggestion. We mentioned Figure 1c (revised Fig. 1d) earlier in the text in the revised manuscript.

- Figure 1d-f: As above, I can't think of a better reference than scATAC-seq, but it's not totally fair to compare pixels to nuclei for complexity, signal-to-noise, etc. I would like the figures to specify that 10X represents cells while "Spatial mapping" represents pixels.

Response: Thanks for the comments. According to this suggestion, we specified that 10X represents cells while “Spatial mapping” represents pixels in the revised Fig. 1e-h (Please see below and Fig. 1e-h in the revised manuscript).

- Figure 1g-h: The lines are too thick and the colors are too hard to distinguish here. It makes it very difficult to assess the relative quality of data. Perhaps fewer lines could be shown in the main figure? I think V1 or V2 metrics could be moved to the Supplement. It looks like the TSS enrichment is significantly better for 10X here than spatial-ATAC. Also, two spatial-ATAC conditions have very low TSS enrichments, but I can't tell which ones based on the plot.

Response: Thanks for the comments. According to this suggestion, we decreased the line thickness and removed the “V1” data and only presented the V2.1 data in Fig. 1i, j (please see below and Fig. 1i, j in the revised manuscript).

“Spatial chromatin accessibility mapping of E13 mouse embryo”

- Figure S3d – what is hsrATAC-seq?

Response: Thanks for pointing out this confusion. We have changed it to “Spatial-ATAC-seq” (Please see below and also Fig. S3e).

- Figure S3e (and Figure S9e) – 20kb seems too big to see meaningful data given the size of the plots. Consider making the distance covered smaller or stretching the plots out horizontally.

Response: Thanks for the comments. Following the reviewer’s suggestion, we stretched the plots out horizontally (Please see below and also Fig. S3c and Fig. S10c in the revised manuscript).

- For projection of ENCODE ATAC-seq onto spatial-ATAC UMAPs, please be explicit that the data is being subsampled in “pseudo single cells”. How many? There seem to be some lack of overlap between bulk data and single-cell data. The authors should comment on this in the manuscript. Also, could the authors please include a supplementary table that details the exact data sets downloaded?

Response: Thanks for the comments. We added “..., the ENCODE ATAC-seq data was subsampled in pseudo single cells (n = 250) and was projected onto spatial-ATAC UMAPs using the projectBulkATAC function in ArchR” in the revised methods section. In addition, we added “Some lack of overlap between bulk data and spatial-ATAC-seq data may be due to each pixel contains multiple cell types.” Moreover, we included a supplementary table that details the exact data sets downloaded in the revised methods section “Published data for data quality comparison and integrative data analysis”.

- For the UMAP, it is clear that the resolution isn’t great even for quite diverse cell types. Having 10X scATAC data on embryos would really help to identify if this is a limitation of N or a limitation of pixel size or a limitation of developmental stage.

Response: Thanks for the comments. Currently, there is no published scATAC-seq on mouse embryos. Therefore, we generated new spatial-ATAC-seq data from P21 mouse brain and human hippocampus, which have existing scATAC-seq data. We did data integration with scATAC-seq in mouse brain, human tonsil, and human hippocampus. Spatial tissue pixels were found to conform well into the clusters of scATAC-seq data (Please see below and also Fig. 3f, Fig. S18c, and Fig. S19b).

(1) Data integration with scATAC-seq in mouse brain

(2) Data integration with scATAC-seq in human tonsil

(3) Data integration with scATAC-seq in human hippocampus

- As the concept of marker genes are introduced in the main text, the statistical significance is not discussed. Are the genes called out in the text listed because they were known in the literature first? Or were they identified as specifically active in individual clusters? It seems like genes are mentioned for both reasons at various points. The authors could be more explicit about whether a marker gene is being discussed because it was identified as differential between clusters or because it was pulled

from the literature. I would add what significance threshold is used to define something as a marker to the main text. How many genes are identified as markers for each cluster? How many go up/down?

Response: Thanks for the comments. The marker genes are being discussed in the manuscript because they were identified as one of the top differential genes between clusters, and they were also known in the literature. In the revised manuscript, we specified that “marker genes were identified as differential between clusters”, we also added what significance threshold is used to define marker gene. For example, “We further examined cell type-specific marker genes, identified as differential between clusters (FDR < 0.05, Log2FC >= 0.25)”. For clarification, We also added the ArchR function used to calculate marker genes in the methods section, “The getMarkerFeatures and getMarkers function in ArchR (testMethod = “wilcoxon”, cutOff = “FDR <= 0.05”) was used to identify the marker regions/genes for each cluster, and the marker genes are being discussed in the manuscript because they were identified as one of the top differential genes between clusters, and they were also known in the literature.” in the revised methods section.” Finally, we added a supplemental file that included all marker genes in different samples.

- Instead of using GO on marker genes from activity scores, what happens if the authors take marker peaks and look for enrichments with a program like GREAT?

Response: Thanks for the comments. We conducted GREAT enrichment analysis for each cluster, and the GREAT pathways identified the development processes consistent with the anatomical annotation (Please see below, Fig. S7e in the revised manuscript).

- The liver subclusters are not visually compelling in Figure S3g and it is unclear to me how the marker genes associate with the subclusters. How were the 2 subclusters defined? The text does not specify. Similarly, for the spinal A/P data, the correlations seem muted. Would any of these survive multiple testing corrections?

Response: Thanks for the comments. We removed Figure S3g since the marker genes associate with the subclusters were not statistically significant. For the spinal A/P data, the p-value was calculated using cor.test function in R. For clarification, we added “Correlation analysis of was conducted by calculating the Pearson correlation coefficient in R, and p-value was calculated using cor.test function.” In the revised methods section.

- For the second paragraph of the section, one of the truly unique things that ATAC-seq can provide are the locations of distal regulatory elements. Unfortunately, Figure S7a doesn't even get specifically mentioned in the main text. How many of the elements that define spatial clusters are proximal to TSSs? Distal? What other features are they associated with? Could the authors build maps linking distal elements to the genes they regulate with a program like Cicero? Or are the data too sparse? Can the authors see A/B compartment switching between clusters? Are there any other features of the chromatin that wouldn't be directly inferable from an analogous RNA map?

Response: Thanks for the comments. In the revised manuscript, we annotated the peaks (Please see below and Fig. S7b in the revised manuscript). In addition, we further explored the existing data for regulatory elements and linked these elements with the genes they might regulate using the addPeak2GeneLinks function in ArchR. We studied both embryo and tonsil data, and we found some distinct elements around several genes of interest that open or close in pseudotime or distinct spatial locations (Please see above, Fig. 2i and Fig. 4i in the revised manuscript).

- Also, in identifying enriched motifs in this section (and elsewhere), there is no discussion of statistical rigor. Did the authors just look at the most enriched motif? Was it significant after multiple testing correction? Were there other motifs for each cluster enriched/depleted?

Response: Thanks for the comments. In the manuscript, we only looked at the most enriched motifs. In the revised manuscript, we added what significance threshold is used to define enriched motifs. For example, "To further utilize the underlying chromatin accessibility data, we sought to examine cell type-specific transcription factor (TF) regulators within each cluster using deviations of TF motifs (FDR < 0.05, Log2FC >= 0.1)."

- In the third paragraph, the statement on erythroid cells in the liver seems based on qualitative assignments of anatomical features and cell type projections, so I would recommend softening the language from "were exclusively enriched" to "appeared predominantly clustered" or something along those lines.

Response: Thanks for the comments. In the revised manuscript, we changed the text from "were exclusively enriched" to "appeared predominantly clustered".

- Also, the authors mention cells not identifiable in E11, but the E11 data has not been introduced yet.

Response: Thanks for the comments. We discussed this later in the E11 data in the revised manuscript, "Additionally, compared to E13, hepatocytes and white blood cells could not be identified in the E11 liver region, suggesting that these cell types emerged at the later developmental time points."

- For the pseudotime analysis of neuron development, it would be worthwhile to include individual peaks that significantly change accessibility over pseudotime in addition to marker genes and TF motifs. Or are the data too sparse for this analysis as well?

Response: Thanks for the comments. According to this suggestion, we further explored the existing data for regulatory elements and linked these elements with the genes they might regulate using the `addPeak2GeneLinks` function in ArchR. We studied both embryo and tonsil data, and we found some distinct elements around several genes of interest that open or close in pseudotime or distinct spatial locations.

In the revision, we sought to exploit our spatial-ATAC-seq data to recover the spatially organized developmental trajectory and examine how developmental processes proceed across the tissue space. We studied the course of a developmental process from radial glia to excitatory neurons with postmitotic premature neurons as the immediate state after the radial glial differentiation and ordered these cells in pseudo-time using ArchR. Spatial projection of each pixel's pseudo-time value revealed the spatially organized developmental trajectory in neurons (Please see below and also Fig. 2 in the revised manuscript). Interestingly, we observed that cells early in differentiation clustered around the ventricles in the developing brainstem, whereas those farther away exhibited a more differentiated phenotype (Please see below and also Fig. 2g). We also used the correlation of peak accessibility to predict the interactions between regulatory regions and found dynamically regulated promoter interactions with specific enhancers in genes, such as *Pou3f2* and *Nova2* (Please see below and also Fig. 2i). *Pou3f2*, also known as *Brn2*, encodes a transcription factor expressed in mouse in late progenitors and postmitotic neurons³⁹ and that has been shown to be important during neural development for the production of specific neuronal populations⁴⁰. Our analysis shows a reduction of chromatin accessibility at a specific *Brn2* enhancer during the transition from radial glial to postmitotic premature neurons, but not at other cis-regulatory regions, suggesting a role of this region in *Brn2* transcription. A similar decrease chromatin accessibility was observed in excitatory neurons for a specific intronic enhancer of *Nova2*, encoding an RNA binding protein expressed in neurons³⁸. Thus, our data indicated that spatial-ATAC-seq allows to map at a spatial level the chromatin accessibility dynamics at important regulatory regions during neural lineage commitment.

“Spatial chromatin accessibility mapping of E11...”

- What happens when visualizing E11 and E13 spatial-ATAC data together with UMAP?

Response: Thanks for the comments. We visualized E11 and E13 spatial-ATAC data together with UMAP. Spatial tissue pixels from E11 and E13 were found to conform well (Please see below and also Fig. S15a, b).

- It's true that the AGM does pop out, but many other features of "fine structure" do not. It would be worth exploring why or softening the language about how fine a feature that can be identified.

Response: Thanks for the comments. In the revised manuscript, we removed the text about how fine a feature that can be identified.

- Second paragraph: again, some statistical rigor is needed here. How many sites are differential between clusters? What pathways come out with a region-based enrichment analysis like GREAT instead of GO?

Response: Thanks for the comments. In the revised manuscript, we annotated the peaks (Please see below and Fig. S12b in the revised manuscript). In addition, we conducted GREAT enrichment analysis for each cluster, and the GREAT pathways identified the development processes consistent with the anatomical annotation (Please see below and Fig. S12e in the revised manuscript).

(1) Peak annotation:

(2) GREAT enrichment analysis:

- The authors state that a higher proportion of radial glia are identified in E11, but not E13. But what are the proportions. Without replicates to identify if this is significantly different, it's a hard statement to evaluate. It might be worth removing from the text in that case.

Response: Thanks for the comments. In the revised manuscript, we removed this statement from the text.

- For the fourth paragraph of this section, the authors identify differentially accessible elements between E11 and E13 in the liver. In addition to motifs, are these elements distal/proximal/etc.? Figure 3l and O provide sites opening and closing, but the volcano plots look strange. 3o in particular looks not properly centered for Fold-change. The authors might want to explore why the nadir of the volcano is not at Log₂FC 0. In 3m and 3p, I don't see lines connecting individual motifs to individual points. Do all the other genes need to be listed on those plots? The authors should highlight the individual point that represents the highlighted motifs in both.

Response: Thanks for the comments. The volcano plots look not properly centered for Fold-change, since there are a lot of points have high Fold-change but low p-value. In the revised manuscript, we replaced the volcano plots with a bar plot showing sites opening and closing (Please see below and Fig. S15c in the revised manuscript).

In addition, we added lines connecting individual motifs to individual points. We also reduced the number of genes listed on the plots and highlighted the individual points and motifs (Please see below and Fig. S15d-g in the revised manuscript).

“Spatial chromatin accessibility mapping of human tonsil and immune cell states”

- I see clusters defined, but I don't see a UMAP for tonsil pixels. Did I miss it?

Response: Thanks for the comments. We added the UMAP for tonsil pixels in the revised manuscript. (Please see below and also Fig. S19a)

- When discussing enrichments of accessibility for certain genes in GC regions, please quantify and give a sense of the statistical significance again.

Response: Thanks for the comments. In the revised manuscript, we added what significance threshold is used to define marker gene, “We set out to explore the spatial patterns of specific marker genes to distinguish cell types (FDR < 0.05, Log2FC >= 0.1)”.

- Also, as with the embryos, it would be helpful to know if markers are defined as significant markers for UMAP clusters, or were pulled from the literature, or identified some other way. This is not always clear in the text.

Response: Thanks for the comments. The marker genes are being discussed in the manuscript because they were identified as one of the top differential genes between clusters, and they were also known in the literature. For clarification, we added “..., and the marker genes are being discussed in

the manuscript because they were identified as one of the top differential genes between clusters, and they were also known in the literature.” in the revised methods section.

- Last paragraph on B cell pseudotime maturation in GCs. This strikes me as potentially the strongest example in favor of spatial-ATAC-seq. This is the model that might allow for dynamic Cicero models of distinct modules looping in to regulate target genes in order and would be something that you really could not get easily with another technology. Have the authors tried something like that? Even just showing distinct elements around several genes of interest that open or close in pseudotime would be a very compelling case for spatial-ATAC. Also, might be worth considering analogously if distance from GC edge (inside or outside of it), leads to different models of gene regulation from the perspective of individual regulatory elements.

Response: Thanks for the comments! According to this suggestion, we further explored the existing data for regulatory elements and linked these elements with the genes they might regulate using the `addPeak2GeneLinks` function in ArchR. We studied both embryo and tonsil data, and we found some distinct elements around several genes of interest that open or close in pseudotime or distinct spatial locations.

In the revision, we implemented a pseudotemporal reconstruction of B cell activation to the GC reaction, where GC B cells were enriched inside GC and Naïve B cells are outside of it (Fig. 4f, g). The dynamic chromatin accessibility of some individual regulatory elements around several genes of interest were investigated. We identified putative target genes of fine-mapped autoimmune GWAS genetic variants, and revealed GC-specific regulatory potential, including at loci of major GC regulators such as *BCL6* (Please see below and Fig. 4i).

Discussion

- I think it's inappropriate to call the data quality "excellent". As I've said above, I think the data quality are good with this assay, but it seems very clear to me that they are not quite as good as single cell or bulk.

Response: Thanks for the comments. In the revised manuscript, we removed these words, such as "excellent" or "equivalent".

- I'd like the Discussion to include consideration of other limitations of the technique (such as throughput, the difference between a pixel and a cell, the potential of partial nuclei in pixels, the consequences of multiple nuclei in pixels).

Response: Thanks for the comments. In the revised Discussion, we added "The limitations or the areas for further development include the following. First, the number of mapping pixels of spatial-ATAC-seq could be further increased by increasing the number of barcodes (e.g. 100 × 100) or using a serpentine microfluidic channel design for spatial mapping of tissue array. Second, different from single cell technologies, the pixels in spatial-ATAC-seq may contain partial nuclei or multiple nuclei and could lead to cell mixtures that may comprise multiple cell-types, complicating data interpretation. This challenge could be addressed by using cell-type deconvolution approaches or seamless integration with high-resolution tissue images, i.e., multicolor immunofluorescence image, to identify the cells in each pixel."

- I'd also like the Discussion to include unique future advantages of having chromatin accessibility (instead of RNA or protein) at pixel resolution (instead of single cells).

Response: Thanks for the comments. In the revised Discussion, we added "We anticipate that spatial-ATAC-seq will add a new dimension to spatial biology, enabling the profiling of regulatory elements in a spatially resolved manner, which cannot be achieved by spatial transcriptomics or proteomics."

Methods

- I noticed that LSI was run with "sampleCells" set to 10,000. But I assume that only 2500 pixels were analyzed for each experiment. Do I have that right?

Response: Thanks for pointing out this confusion. "sampleCells" is a parameter to specify the number of cells to subsample and perform clustering on. The remaining cells that were not subsampled will be assigned to the cluster of the nearest subsampled cell. This enables a decrease in run time, and the default value is 10,000. However, if cells fewer than 10,000, the function will use all of the cells instead of subsampling. For clarification, we removed "sampleCells = 10000" in the revised methods section.

Referee #2 (Remarks to the Author):

In their manuscript, Deng et al present a technique for spatially resolved sequencing of accessible chromatin, by combining a microfluidic device the authors previously invented and used in RNA sequencing, with Tn5 transposase technology. It's exciting that the technique is able to deliver spatially resolved open chromatin profiles (though the exact spatial resolution remains unclear), and the number of unique fragments per pixel is substantial, making the method of potential use in a variety of biological settings. While an intriguing method, its characterization seems rather cursory. The authors should have gone into significantly more depth to characterize their method, and clearly present its strengths, weaknesses, and known sources of technical noise.

Response: We would like to thank the reviewer for the positive feedback regarding our manuscript and work.

1. Although clearly spatial information is retained in this technique (it is visible by eye), there is no experimental or computational analysis that definitively proves the spatial resolutions claimed by the channel sizes used (visual inspection of expression is not quantitative). A quantitative assessment of this spatial resolution is important, since there are clearly histological features that are lost when comparing the images in, for example, Fig S4. Can the authors look at the features of the spinal cord, compute the thickness of those features as represented by ATAC signal of, say, Sox10 and/or Nkx6-1, and provide a quantitative assessment of resolution? Can this be done with both the 20 micron and 50 micron devices, on serial sections, so that the reader can clearly compare and contrast the feature thicknesses for these two resolutions, and determine whether a smaller channel size really does improve spatial resolution?

Response: Thanks for the comments. In the revised manuscript, we generated more data from mouse brain and E11 mouse embryo with 20 μm pixel size (Please see below, Fig. 3, Fig. S14, S16, and S17). To determine the spatial resolution of this approach, we first performed immunofluorescence staining with 7-AAD (7-Aminoactinomycin D, a nuclear DNA dye), which revealed fine features including a thin layer of vascular cells (width is 14.2 μm) (Please see below, top panel and Fig. S17a in the revised manuscript) and SVZ (width is 16.8 μm) (Please see below, bottom panel and Fig. S17b in the revised manuscript). We then applied spatial-ATAC-seq to the same tissue section, which successfully identified these features by a single layer of pixels (width is 20 μm) (Please see below and Fig. S17a, b in the revised manuscript). It shows that spatial-ATAC-seq is able to resolve the features that close to the pixel size.

In addition, we binned 20 μm spatial-ATAC-seq data (Please see below, left panel and Fig. 3c in the revised manuscript) to the feature size of 60 μm (two adjacent 20 μm spots) (right panel and Fig. S17c in the revised manuscript). The high spatial resolution of spatial-ATAC-seq is critical to mapping more cell types: when data were aggregated into larger feature sizes, cell types in heterogeneous regions of tissue could not be resolved. For example, vascular cells (C5) and astrocytes (C7) in the 20 μm spatial-ATAC-seq data (left panel) cannot be identified in the data with lower resolution (right panel).

Moreover, we conducted spatial-ATAC-seq experiment with E11 mouse embryo tissue sample at resolution of 20 μm pixel size (please see below and Fig. S14 in the revised manuscript). Compared to 50 μm device, 20 μm experiment were able to resolve more cell types. For instance, a thin layer of notochord cells was identified with 20 μm resolution (C3), which could not be resolved in the 50 μm experiment.

2. Relatedly, I am confused about the separation between pixels—I apologize if this was explained somewhere and I missed it, but shouldn't the pixels be separated by a certain distance, namely the channel thickness? As such, wouldn't the expectation be that these pixels are separated by some amount of dead space? Has this separation been included for the visualizations shown in these figures? What are the absolute size dimensions, in microns, of the pixels shown in the ATAC plots?

Response: Thanks for pointing out this confusion. Indeed, pixels should be separated by the distance of channel thickness (please see below for 20 µm spatial-ATAC-seq data for mouse brain), and the absolute size dimensions of the pixels are the same as channel size (20 µm). However, for better visualization, we increased the pixel size, which is a common feature in the Seurat package. For clarification, we added "For better visualization, we scaled the size of the pixels using the "pt.size.factor" parameter in the Seurat package." in the methods of the revised manuscript.

3. As far as I can tell, the manuscript contains data from six datasets in total—four E11 experiments, one E13, and one Tonsil (looking at Figs 1g and S2). Can the authors provide some information about throughput of the method? Is it conceivable that one could generate a full 3-D volume of the embryo, or is one limited in some way experimentally?

Response: Thanks for the comments. For current spatial-ATAC-seq protocol, each person can process two sections simultaneously in Day 1 experiment (i.e. 5000 pixels). However, in a week (5 days), we can run four Day 1 experiments and store the lysates in -20 °C, and then perform Day 2 experiment for these 8 lysates simultaneously. Therefore, it is conceivable that one could generate a full 3-D volume of the embryo.

To further increase the throughput, we can increase the number of barcodes (e.g. 100x100), such that 10,000 pixels can be handled. Additionally, a serpentine microfluidic channel design will enable spatial mapping of tissue array (please see the channel design below). We added "First, the number of mapping pixels of spatial-ATAC-seq could be further increased by increasing the number of barcodes

(e.g. 100 × 100) or using a serpentine microfluidic channel design for spatial mapping of tissue array.”
in the discussion of the revised manuscript.

(a) Serpentine microfluidic channel design for barcodes A1-A100

Image redacted

Image redacted

Image redacted

4. Figures 2 and 3 seem rather repetitive. What is the point of presenting, for example, the single cell mapping and pseudotime analyses for both E11 and E13? Are we learning anything new about the method by looking at two different time points? I found the pseudotime shown in Fig 2g especially confusing—why are the limb regions displaying pseudotime loading for a trajectory that is labeled as “Radial glia -> postmitotic premature neurons -> excitatory neurons”? In displaying these pseudotime analyses, can the authors clearly label the expected spatial developmental trajectory, so that those who are unfamiliar with neural development can understand whether the method appropriately recapitulates what is biologically known?

Response: Thanks for the comments. We moved the E11 data to the supplemental figure and added the new data from mouse brain with 20 μm pixel size (please see Fig. 3 in the revised manuscript). In addition, we selected pixels in the central nervous system and performed pseudotime analyses in this region (please see below and Fig. 2g in the revised manuscript). Moreover, as the reviewer suggested, to clarify the expected spatial developmental trajectory, we added “We then identified changes in gene score across this developmental process, and observed high chromatin accessibility in radial glia at *Sox2* and *Pax6* loci, genes encoding for transcription factors necessary for progenitor self-renewal identity⁵. As expected, there was a clear reduction of accessibility in the transition to postmitotic premature neurons and excitatory neurons, which instead presented chromatin opening at genes expressed in mature neurons as *Ntng1*^{5,6} and *Car10*⁷ (Fig. 2h). We also used the correlation of peak accessibility to predict the interactions between regulatory regions and found dynamically regulated promoter interactions with specific enhancers in genes, such as *Pou3f2* and *Nova2* (Fig. 2i). *Pou3f2*, also known as *Brn2*, encodes a transcription factor expressed in mouse in late progenitors and postmitotic neurons⁸ and that has been shown to be important during neural development for the production of specific neuronal populations⁹. Our analysis shows a reduction of chromatin accessibility at a specific *Brn2* enhancer during the transition from radial glial to postmitotic premature neurons, but not at other cis-regulatory regions, suggesting a role of this region in *Brn2* transcription. A similar decrease chromatin accessibility was observed in excitatory neurons for a specific intronic enhancer of *Nova2*, encoding an RNA binding protein expressed in neurons⁷. Thus, our data indicated that spatial-ATAC-seq allows to map at a spatial level the chromatin accessibility dynamics at important regulatory regions during neural lineage commitment.”

5. I found the inclusion of “V1” data confusing. As far as I can tell, V1 is simply a lot worse in sensitivity than “V2.” Since this method has never been published before, why even mention “V1” or include this data? The number of unique fragments per pixel is impressive, and it seems to me that simply presenting the V2 data as the “spatial-ATAC-seq” method would be far clearer to readers.

Response: Thanks for the comments. According to this suggestion, we removed the “V1” data and only presented the V2.1 data in Fig. 1 (please see below and Fig. 1e-k in the revised manuscript).

Small questions/points:

- For the integrative analyses in Figs 2d and 3d, the authors should show all cell type mappings in extended data, rather than just the selected annotations seen in the 2f/3f panels.

Response: Thanks for the comments. We added all cell type mappings in extended data for the integrative analyses in Figs 2 and 3 (Please see Fig. S8a and Fig. S16a, in the revised manuscript).

- There is a lot of variability in the % mitochondrial fragments across replicates. Can the authors provide some explanation for why this might be?

Response: Thanks for the comments. The variability in the percentage of mitochondrial fragments may come from different type of tissues, since we observed similar percentage of mitochondrial fragments in the mouse embryo data, whereas the differences were found in the mouse brain and human tonsil. For clarification, we added “The variability in the percentage of mitochondrial fragments may come from different type of tissues.” in the revised manuscript.

- The authors manually remove pixels that are not overlying tissue. When assessing diffusion/resolution, it might be useful to retain those pixels to understand if and how signal is leaking across adjacent pixels.

Response: Thanks for the comments. We compared the number of unique fragments in pixels in tissue (E13 mouse embryo) and those not in tissue, and we found that pixels not in tissue had significantly fewer unique fragments (median is 8069) than pixels in tissue (median is 100786) (Please see below and Fig. S2c in the revised manuscript).

In addition, in our previous paper⁴, we have evaluated the possibility of DNA diffusion by analyzing a 3D fluorescence confocal image, which confirmed negligible leakage signal throughout the tissue section thickness (Please see below). It was found to be $0.9 \pm 0.2 \mu\text{m}$ for $10 \mu\text{m}$ channels and $4.5 \pm 1 \mu\text{m}$ for $50 \mu\text{m}$ channels, which validated spatially confined delivery and binding of DNA barcodes in tissue using microfluidics.

Image redacted

Referee #3 (Remarks to the Author):

Deng et al., introduce 'spatial-ATAC-seq' a technology that profiles the chromatin accessibility of spot regions while simultaneously retaining the spatial location of the cell in the tissue. The design of this method cleverly combines the chemistry of ATAC-seq and microfluidic deterministic barcoding of the tissue that has been previously developed by the Fan lab; where a section of tissue is subjected to Tn5 transposition to profile the chromatin accessibility followed by in-situ ligation of DNA barcodes that denote spatial location within the tissue. The authors demonstrate the applicability of spatial-ATAC-seq by applying it to 50 um sections of E11 and E13 mouse embryos as well as to 20 um sections of human tonsils. These datasets largely recapitulate known molecular features for each individual tissue types.

The novelty of this manuscript lies in the technological advancement to the field of spatial omics. Spatial-ATAC-seq, would be one of the first methodologies that adds an epigenetic layer to spatial technologies - and the first to do this systematically over a tissue section. The authors successfully demonstrate with proof-of-principle experiments that this methodology is powered to identify known biological features within each dataset. However, the authors do not show how this methodology can be leveraged to discover new biological phenomena. Overall, while the manuscript presents several pieces of interesting data, the authors do not make significant biological claims. Without the spatial component to the analysis, the findings and novelty of the manuscript is limited. The major concerns are:

Response: We would like to thank the reviewer for the positive feedback regarding our manuscript and work.

1. In Figures 2 and 3 the authors identify a relatively low number of clusters given the complexity of the mouse fetus at the stage under study. The authors should characterize the resolution of the method relative to that of single-cell ATAC-Seq, for example relative to the ones detected in this atlas: <https://science.sciencemag.org/content/370/6518/eaba7612.long>

Response: Thanks for the comments. Currently, there is no published scATAC-seq on mouse embryos. Therefore, we generated new spatial-ATAC-seq data from P21 mouse brain and human hippocampus, which have existing scATAC-seq data. We did data integration with scATAC-seq in mouse brain, human tonsil, and human hippocampus. Spatial tissue pixels were found to conform well into the clusters of scATAC-seq data (Please see below and also Fig. 3f, Fig. S18c, and Fig. S19b).

(1) Data integration with scATAC-seq in mouse brain

(2) Data integration with scATAC-seq in human tonsil

(3) Data integration with scATAC-seq in human hippocampus

2. In the discussion the authors state “We observed that a significant number of pixels (20μm) contained single nuclei and the extraction of sequencing reads from these pixels can give rise to spatially-defined single-cell ATAC-seq data.” However, the data to support this claim about resolution is missing from the manuscript.

Response: Thanks for the comments. In the revised manuscript, we demonstrated the combination of spatial-ATAC-seq with immunofluorescence staining in the same tissue section (Please see below and Fig. 3i, j in the revised manuscript). A mouse brain tissue section was stained with 7-AAD (7-Aminoactinomycin D, a nuclear DNA dye), and then spatial-ATAC-seq was performed with 20 μm pixel size. With 7-AAD staining for nucleus, we could identify the pixels containing only one nucleus (N=106). It should be noted that the number of pixels contained single nuclei is also related to tissue types.

3. In Figure 1c, the authors compare the spatial-ATAC data with bulk ATAC data from the liver, could the authors show that the genomic region shown is specific to the liver and different from other tissue types?

Response: Thanks for the suggestion. In the revised manuscript, we showed the signals around *Slc4a1* gene, which is specifically enriched in the liver region but not in the brain (Please see below and Fig. 1d in the revised manuscript). *Slc4a1* is required for normal flexibility and stability of the erythrocyte membrane and for normal erythrocyte shape¹⁰.

4. From the pseudotime analysis in Figures 2g-h and 3g-h, the authors do not make a claim for any insight generated from the spatial component. Also, the methods section on pseudotime for figure is unclear. It seems likely that real insight could stem from this analysis, so we would encourage the authors to explore further what the data may be suggesting.

Response: Thanks for the comments! According to this suggestion, we further explored the existing data for regulatory elements and linked these elements with the genes they might regulate using the `addPeak2GeneLinks` function in ArchR. We studied both embryo and tonsil data, and we found some distinct elements around several genes of interest that open or close in pseudotime or distinct spatial locations.

In the revision, we sought to exploit our spatial-ATAC-seq data to recover the spatially organized developmental trajectory and examine how developmental processes proceed across the tissue space. We studied the course of a developmental process from radial glia to excitatory neurons with postmitotic premature neurons as the immediate state after the radial glial differentiation and ordered these cells in pseudo-time using ArchR. Spatial projection of each pixel's pseudo-time value revealed the spatially organized developmental trajectory in neurons (Please see below and also Fig. 2 in the revised manuscript). Interestingly, we observed that cells early in differentiation clustered around the ventricles in the developing brainstem, whereas those farther away exhibited a more differentiated phenotype (Please see below and also Fig. 2g). We also used the correlation of peak accessibility to predict the interactions between regulatory regions and found dynamically regulated promoter interactions with specific enhancers in genes, such as *Pou3f2* and *Nova2* (Please see below and also Fig. 2i). *Pou3f2*, also known as *Brn2*, encodes a transcription factor expressed in mouse in late progenitors and postmitotic neurons³⁹ and that has been shown to be important during neural development for the production of specific neuronal populations⁴⁰. Our analysis shows a reduction of chromatin accessibility

at a specific *Brn2* enhancer during the transition from radial glial to postmitotic premature neurons, but not at other cis-regulatory regions, suggesting a role of this region in *Brn2* transcription. A similar decrease chromatin accessibility was observed in excitatory neurons for a specific intronic enhancer of *Nova2*, encoding an RNA binding protein expressed in neurons³⁸. Thus, our data indicated that spatial-ATAC-seq allows to map at a spatial level the chromatin accessibility dynamics at important regulatory regions during neural lineage commitment.

In addition, we implemented a pseudotemporal reconstruction of B cell activation to the GC reaction, where GC B cells were enriched inside GC and Naïve B cells are outside of it (Fig. 4f, g). The dynamic chromatin accessibility of some individual regulatory elements around several genes of interest were investigated. We identified putative target genes of fine-mapped autoimmune GWAS genetic variants, and revealed GC-specific regulatory potential, including at loci of major GC regulators such as *BCL6* (Please see below and Fig. 4i).

Finally, following the reviewer's suggestion, we provided a more detailed method section to describe details of pseudotime analysis and updated the GitHub repository to share the code outlining how the analysis was performed (https://github.com/dyxmvp/Spatial_ATAC-seq).

5. In Figures S3 and S9, the authors project the ENCODE ATAC-Seq onto the latent space defined by their spatial ATAC-Seq. However, since the resolution appears to be low for spatial ATAC-Seq data, it would be informative to also examine the opposite: projection of the spatial ATAC-Seq data onto the ENCODE-defined latent space.

Response: Thanks for the comments. Following the reviewer's suggestion, we projected the spatial-ATAC-Seq data onto the ENCODE-defined latent space. Indeed, the resolution of two-dimensional representation of the data obtained was improved (Please see below and also Fig. S3f, g and Fig.S10f, g in the revised manuscript).

(1) E13 mouse embryo:

(2) E11 mouse embryo:

6. The authors conclude that there is a discordance between gene expression and chromatin accessibility: "CXCR4, which is expressed in the centroblasts in the GC dark zone [...]". To make this claim the authors should explicitly show the evidence for a lack of gene expression as opposed to only showing the chromatin accessibility.

Response: Thanks for the comments. The higher expression of CXCR4 in the centroblasts in the GC dark zone was reported by several studies^{11,12}. For example, Victora et al. (*Blood*, 2012)¹¹ did

histologic examination and confirmed higher expression of CXCR4 in the GC dark zone (Please see below). For clarification, we have added the related references in the revised manuscript.

Image redacted

7. The authors do not discuss the applicability of the method. As part of the publication of the method, the authors should include a detailed protocol for how a lab with reasonable resources may be able to set it up.

Response: Thanks for the comments. We have included a detailed protocol in the Methods section for device fabrication, experimental procedure, and data analysis. Additionally, we have published a protocol in terms of device fabrication and operation, which will help the labs without any experience in microfluidics set up the platform¹³ (added in the revised Methods section). Moreover, we will publish a step-by-step protocol in Protocol Exchange or protocols.io.

Minor Critiques

1. In figure 1, in addition to the QC metrics shown in Figure 1, the authors should include a FRiP score (fraction of reads in peaks) as an indicator of ATAC-seq quality.

Response: Thanks for the comments. We included a figure for FRiP score as an indicator of ATAC-seq quality in the revised manuscript. (Please see below and also Fig. 1g)

2. Citations of known biological phenomena are missing in a significant proportion of the manuscript. A few such places in the manuscript are cited below.

“Sptb, which plays a role in stability of erythrocyte membranes, was activated extensively in the liver. Syt8, which is important in neurotransmission, had a high level of gene activity in the spine. Ascl1 showed strong enrichment in the mouse brain, which is known to be involved in the commitment and differentiation of neuron and oligodendrocyte (Fig. 2c, Fig. S4e, f). Sox10 marks oligodendrocyte progenitor cells (OPCs). It was expressed at a high level in the dorsal root ganglia (DRGs), which are adjacent to the spinal cord (Fig. S4a, b). Olig2 is a marker of neural progenitors, pre-OPCs and OPCs. Olig2 is expressed in a small domain of the spinal cord, in the ventral domains of the forebrain, and in some posterior regions (brain stem, midbrain and hindbrain), which is consistent with the high gene score in the spatial ATAC-seq data (Fig. S4c, d).”

“Sox2, which is required for stem cell maintenance in the central nervous system, and Ntng1, which is involved in controlling patterning and neuronal circuit formation (Fig. 2h, i)”

“For example, Slc4a1, which are required for normal flexibility and stability of the erythrocyte membrane and for normal erythrocyte shape, were highly active in liver and AGM. Nova2, which is involved in RNA splicing or metabolism regulation in a specific subset of developing neurons, was highly enriched in the brain and neural tube.”

“Interestingly, it showed extensive open locus accessibility, suggesting extensive epigenetic priming of pre-GC T cells to potentially develop TFR function as needed to balance GC activity. CD25, a surface marker for regulatory T cells, was active in both GC and the extrafollicular zone. For non-lymphoid cells, CD11B, a macrophage marker, was inactive in GC, on contrast to CD11A, which was more active in GC lymphocytes. CD103 was enriched in GC follicular dendritic cells. CD144, which encodes vascular endothelial cadherin (VE-cadherin), corresponded to endothelial microvasculature near the crypt or between follicles. CD32, a surface receptor involved in phagocytosis and clearing of immune complexes, and CD55, a complement decay- accelerating factor, were both active in the same region such that the cells not supposed to be cleared can be protected against phagocytosis by blocking the formation of the membrane attack complex.”

Response: Thanks for the comments. We added references for known biological phenomena in the revised manuscript, and the total number of references increased from 41 to 79.

References

- 1 Nguyen, Q. H., Pervolarakis, N., Nee, K. & Kessenbrock, K. Experimental Considerations for Single-Cell RNA Sequencing Approaches. *Frontiers in Cell and Developmental Biology* **6**, doi:10.3389/fcell.2018.00108 (2018).
- 2 Denisenko, E. *et al.* Systematic assessment of tissue dissociation and storage biases in single-cell and single-nucleus RNA-seq workflows. *Genome Biology* **21**, 130, doi:10.1186/s13059-020-02048-6 (2020).
- 3 van den Brink, S. C. *et al.* Single-cell sequencing reveals dissociation-induced gene expression in tissue subpopulations. *Nature Methods* **14**, 935-936, doi:10.1038/nmeth.4437 (2017).
- 4 Liu, Y. *et al.* High-Spatial-Resolution Multi-Omics Sequencing via Deterministic Barcoding in Tissue. *Cell* **183**, 1665-1681.e1618, doi:<https://doi.org/10.1016/j.cell.2020.10.026> (2020).
- 5 Gómez-López, S. *et al.* Sox2 and Pax6 maintain the proliferative and developmental potential of gliogenic neural stem cells In vitro. *Glia* **59**, 1588-1599, doi:<https://doi.org/10.1002/glia.21201> (2011).
- 6 Nishimura-Akiyoshi, S., Niimi, K., Nakashiba, T. & Itohara, S. Axonal netrin-Gs transneuronally determine lamina-specific subdendritic segments. *Proceedings of the National Academy of Sciences* **104**, 14801-14806, doi:doi:10.1073/pnas.0706919104 (2007).
- 7 Zeisel, A. *et al.* Molecular Architecture of the Mouse Nervous System. *Cell* **174**, 999-1014.e1022, doi:<https://doi.org/10.1016/j.cell.2018.06.021> (2018).
- 8 Hagino-Yamagishi, K. *et al.* Predominant expression of Brn-2 in the postmitotic neurons of the developing mouse neocortex. *Brain Research* **752**, 261-268, doi:[https://doi.org/10.1016/S0006-8993\(96\)01472-2](https://doi.org/10.1016/S0006-8993(96)01472-2) (1997).
- 9 Sugitani, Y. *et al.* Brn-1 and Brn-2 share crucial roles in the production and positioning of mouse neocortical neurons. *Genes & development* **16**, 1760-1765 (2002).
- 10 Stelzer, G. *et al.* The GeneCards Suite: From Gene Data Mining to Disease Genome Sequence Analyses. *Current Protocols in Bioinformatics* **54**, 1.30.31-31.30.33, doi:<https://doi.org/10.1002/cpbi.5> (2016).
- 11 Victora, G. D. *et al.* Identification of human germinal center light and dark zone cells and their relationship to human B-cell lymphomas. *Blood* **120**, 2240-2248, doi:10.1182/blood-2012-03-415380 (2012).
- 12 Caron, G., Le Gallou, S., Lamy, T., Tarte, K. & Fest, T. CXCR4 Expression Functionally Discriminates Centroblasts versus Centrocytes within Human Germinal Center B Cells. *The Journal of Immunology* **182**, 7595, doi:10.4049/jimmunol.0804272 (2009).
- 13 Su, G. *et al.* Spatial multi-omics sequencing for fixed tissue via DBiT-seq. *STAR Protocols* **2**, 100532, doi:<https://doi.org/10.1016/j.xpro.2021.100532> (2021).

Reviewer Reports on the First Revision:

Referees' comments:

Referee #1 (Remarks to the Author):

Below are my follow-up responses to the rebuttal submitted by Deng et al. The work remains novel and impactful and the manuscript has only been improved by the changes. Deng et al. have made numerous edits to their manuscript in response to my comments. Overall, I found their responses thoughtful and largely sufficient to address the concerns. There were a few points that I felt were still outstanding and so I have provided those here:

- On page 4, line 17: The reference to Fig. 2c, d comes right after a new statement about mitochondrial reads, but these figures don't have anything to do with mitochondrial reads. They are pertinent to the signal from blank pixels and so I would recommend that they are specifically referenced in the text as such. It's somewhat concerning that the FRiP is not all that different between tissue and non-tissue pixels, which might also be worth briefly commenting on in the text.
- On page 4, line 45: I like that the definition of a marker gene is now provided. I would encourage the authors to also include supplemental tables with the genes and peaks that meet their significance/fold-change thresholds for every comparison. It would be of great help to the community to have significant genes and peaks identified from this data set provided in a tangible format along with the actual p-values, adjusted p-values, and fold-changes for each cluster, pseudotime trajectory, etc.
- On page 5, line 21: The authors more clearly defined how the genes shown in Fig S6 were chosen in their response to my comment about the muted correlations of the A/P trends. I take from their response that the correlation p-values were not corrected for multiple testing. If so, this is not statistically valid. The correlation tests should be run for all genes with activity scores across the A/P axis and then the tests should be corrected for multiple testing. At that point it would be appropriate to highlight some genes while reporting the adjusted p-values. Some of the plots in the current figure will likely survive that test, but many will not.
- On page 6, lines 5-7: I like that the relationship to ventricles is mentioned, but I'd like Fig. 2g to annotate the ventricles so that this relationship is obvious to the non-expert. In addition, I noticed in response to one of the other reviewers that the relative sizes of the pixels and void spaces have been modified for visualization for all the pixel plots. I appreciate that the authors added this to the Methods, but I think it should also be added to each legend. And perhaps both views could be provided side-by-side in the supplement for at least some (if not all) of the pixel plots.
- On Fig. 2i: It would be good to indicate the y-axis scale for the browser tracks (including Fig. 1d and Fig. 4i). In addition, it would be helpful if the authors indicated the direction of transcription for the genes. Same goes for Fig. 4i. Further, the legends of Figs 1, 2, 4 don't really properly describe the browser tracks. The legends should indicate that read depth per base pair is plotted (is it normalized?). What the loops represent should be mentioned. Where the correlation color is coming from should be mentioned. What the peaks are should be mentioned. What the gray boxes are highlighting should be defined. Similarly, somewhere the pseudotime plots (e.g. 2h) need to be clearly defined – how are cells binned, how is the gene activity score aggregated, etc.?
- For Figure 3f,g: I am glad that integration with scATAC-seq is now provided. Two things: (1) it is

difficult to see how the cells really sit on top of each other with the UMAPs side-by-side. I would rather the points from the other technology are plotted in gray or black as in 2d. This would also be true for the tonsil and hippocampus data. Also, the RNA integration is now too squished to really discern. I would recommend moving this to the Supplement.

- The authors were kind enough to provide an estimate of the number of nuclei per pixel in their response. I would like to see these numbers make it into the main text (I would like to see the violin plots of diffusion distance and nuclei per channel size put in a Supplementary figure too – with the y-axes properly labeled).

- For Figure S15c: The authors have replaced the volcano plot with a barplot, but this does not really solve the problem. Most of the p-values near 1 (i.e. $-\log_{10} p\text{-value} = 0$) should be at or around a fold-change of 1 (i.e. $\log_2 \text{fold-change} = 0$) by definition. These points would have no change between the two conditions and therefore no significant difference. The fact that most of the points with a p-value of 1 appear to be somewhere around a \log_2 fold-change of -2 would be interpreted as sites with a 4-fold less accessibility in E11 being the sites with the least difference between the groups. Obviously, this is logically nonsensical. Therefore, it must be that the volcano plot is not centered properly. One possibility is that the \log_2 fold-change being plotted is before correcting for sequencing depth or number of cells or something along those lines, while the statistical test is being conducted after correcting for those factors. This might imply that the report of the relative proportion of sites opening and closing is off. It could be even more biased towards opening than the barplot suggests. Whatever the case may be, I would encourage the authors to more deeply investigate the discrepancy regardless of whether they change the plot or not. It may affect some of their interpretations significantly.

Referee #2 (Remarks to the Author):

The authors provide a revised manuscript with more high-resolution data, and additional clarification regarding details of technical validation. I continue to have concerns about the technical characterization of the data they are generating, which I describe below.

In addition, and most importantly, since the original submission of their manuscript, their lab has published spatial cut+tag using the DBit-seq system in Science (Deng et al, 2022). The cut+tag protocol combines antibody binding with targeted Tn5 transposition to reveal regions of the genome containing specific chromatin marks (targeted by the antibody). This Cut+tag protocol is identical to the ATAC-seq protocol in this paper, except the Tn5 is deployed genome-wide, to assess chromatin accessibility, rather than at antibody-targeted locations. As such, the novelty of this submitted manuscript--reporting the feasibility of combining Tn5 transposition with Dbit-seq-mediated ligation--has already been published.

Regarding the technical validation: the authors do a feature profile analysis to compare the thickness of a thin layer of vascular cells in fluorescent images and spatial-ATAC images. What is left now to do is to better characterize the false positive noise. In many of these images, one can see "lines" of signal running vertically or horizontally across the image. The authors should explain the source of this noise, and what computational (or experimental) steps might be employed to mitigate it.

Referee #3 (Remarks to the Author):

The authors have revised their manuscript adding data, analyses, and explanatory notes. They have competently addressed all of my concerns as well as those of the other reviewers. I congratulate the authors on an exciting new method and insightful analyses.

Author Rebuttals to First Revision:

We are very appreciative of all the constructive review comments, which resulted in an improved manuscript. All major concerns from the reviewers have been fully addressed. Please find below, our point-by-point responses to all review comments. Thank you!

Referee #1 (Remarks to the Author):

Below are my follow-up responses to the rebuttal submitted by Deng et al. The work remains novel and impactful and the manuscript has only been improved by the changes. Deng et al. have made numerous edits to their manuscript in response to my comments. Overall, I found their responses thoughtful and largely sufficient to address the concerns.

Response: We would like to thank the reviewer for the positive feedback regarding our manuscript and work!

There were a few points that I felt were still outstanding and so I have provided those here:

- On page 4, line 17: The reference to Fig. 2c, d comes right after a new statement about mitochondrial reads, but these figures don't have anything to do with mitochondrial reads. They are pertinent to the signal from blank pixels and so I would recommend that they are specifically referenced in the text as such. It's somewhat concerning that the FRiP is not all that different between tissue and non-tissue pixels, which might also be worth briefly commenting on in the text.

Response: Thanks for the comments. According to this suggestion, we added "We also found that pixels that are not in tissue had significantly fewer unique fragments than pixels in tissue (Supplementary Fig. 1c, d)." in the revised manuscript.

- On page 4, line 45: I like that the definition of a marker gene is now provided. I would encourage the authors to also include supplemental tables with the genes and peaks that meet their significance/fold-change thresholds for every comparison. It would be of great help to the community to have significant genes and peaks identified from this data set provided in a tangible format along with the actual p-values, adjusted p-values, and fold-changes for each cluster, pseudotime trajectory, etc.

Response: Thanks for the comments. We added a supplemental file that included all marker genes in different samples (Supplementary Table 1).

- On page 5, line 21: The authors more clearly defined how the genes shown in Fig S6 were chosen in their response to my comment about the muted correlations of the A/P trends. I take from their response that the correlation p-values were not corrected for multiple testing. If so, this is not statistically valid. The correlation tests should be run for all genes with activity scores across the A/P axis and then the tests should be corrected for multiple testing. At that point it would be appropriate to highlight some genes while reporting the adjusted p-values. Some of the plots in the current figure will likely survive that test, but many will not.

Response: Thanks for the comments. According to this suggestion, all p-values were adjusted for multiple comparisons by Benjamini & Hochberg method, and we highlighted some genes while reporting the adjusted p-values. (Please see below and Supplementary Fig. 4)

- On page 6, lines 5-7: I like that the relationship to ventricles is mentioned, but I'd like Fig. 2g to annotate the ventricles so that this relationship is obvious to the non-expert. In addition, I noticed in response to one of the other reviewers that the relative sizes of the pixels and void spaces have been modified for visualization for all the pixel plots. I appreciate that the authors added this to the Methods, but I think it should also be added to each legend. And perhaps both views could be provided side-by-side in the supplement for at least some (if not all) of the pixel plots.

Response: Thanks for the comments. According to this suggestion, we annotated the ventricles in Fig. 2e (Please see below).

Additionally, for visualization of pixel sizes, we added “For better visualization, we scaled the size of the pixels.” to each legend and provided side-by-side comparison in the Extended Data Fig.8c (Please see below).

- On Fig. 2i: It would be good to indicate the y-axis scale for the browser tracks (including Fig. 1d and Fig. 4i). In addition, it would be helpful if the authors indicated the direction of transcription for the genes. Same goes for Fig. 4i. Further, the legends of Figs 1, 2, 4 don't really properly describe the browser tracks. The legends should indicate that read depth per base pair is plotted (is it normalized?). What the loops represent should be mentioned. Where the correlation color is coming from should be mentioned. What the peaks are should be mentioned. What the gray boxes are highlighting should be defined. Similarly, somewhere the pseudotime plots (e.g. 2h) need to be clearly defined – how are cells binned, how is the gene activity score aggregated, etc.?

Response: Thanks for the suggestion. In the revised manuscript, we added the y-axis scale in the browser tracks and defined what gray boxes are highlighting in the legends of Fig. 2i and Fig. 4i. In addition, we indicated direction of transcription for the genes, what the loops represent, where the correlation color is coming, what the peaks are, how the pseudotime plots are generated, and how is the gene activity score aggregated in the methods section, “Genome browser tracks were plotted by plotBrowserTrack function in ArchR. Spatial-ATAC-seq data was normalized by the recommended and default value (normMethod = "ReadsInTSS"), which simultaneously normalizes tracks based on sequencing depth and sample data quality. Blue-colored genes are on the minus strand and red-colored genes are on the plus strand. The loops are the links between a peak and a gene, and the color shows

the Pearson correlation between peak accessibility and gene expression. Peaks were called with macs2 using addReproduciblePeakSet function in ArchR. Gene Score model in ArchR was employed to gene accessibility score.”

- For Figure 3f,g: I am glad that integration with scATAC-seq is now provided. Two things: (1) it is difficult to see how the cells really sit on top of each other with the UMAPs side-by-side. I would rather the points from the other technology are plotted in gray or black as in 2d. This would also be true for the tonsil and hippocampus data. Also, the RNA integration is now too squished to really discern. I would recommend moving this to the Supplement.

Response: Thanks for the comments. According to this suggestion, we replotted figures for the data integration similar to Fig. 2d, and moved Fig. 3g to Extended Data Fig.8a (Please see below).

(1) P21 mouse brain:

(2) Human hippocampus:

(3) Human tonsil:

- The authors were kind enough to provide an estimate of the number of nuclei per pixel in their response. I would like to see these numbers make it into the main text (I would like to see the violin plots of diffusion distance and nuclei per channel size put in a Supplementary figure too – with the y-axes properly labeled).

Response: Thanks for the comments. We added “spatial-ATAC-seq pixels may contain more than one nucleus, which is based on the tissue type and cell size. For E10 mouse embryo, on average, there were 1-2 cells per 10 μm pixel and 25 cells per 50 μm pixel” in the main text. The violin plots of diffusion distance and nuclei per channel size were from our previous study (*Cell*, 183(6), 1665-1681 (2020)), and we cited this paper in the main text for reference.

- For Figure S15c: The authors have replaced the volcano plot with a barplot, but this does not really solve the problem. Most of the p-values near 1 (i.e. $-\log_{10} p\text{-value} = 0$) should be at or around a fold-change of 1 (i.e. $\log_2 \text{fold-change} = 0$) by definition. These points would have no change between the two conditions and therefore no significant difference. The fact that most of the points with a p-value of 1 appear to be somewhere around a \log_2 fold-change of -2 would be interpreted as sites with a 4-fold less accessibility in E11 being the sites with the least difference between the groups. Obviously, this is

logically nonsensical. Therefore, it must be that the volcano plot is not centered properly. One possibility is that the log2 fold-change being plotted is before correcting for sequencing depth or number of cells or something along those lines, while the statistical test is being conducted after correcting for those factors. This might imply that the report of the relative proportion of sites opening and closing is off. It could be even more biased towards opening than the barplot suggests. Whatever the case may be, I would encourage the authors to more deeply investigate the discrepancy regardless of whether they change the plot or not. It may affect some of their interpretations significantly.

Response: Thanks for the comments. The volcano plots were generated by markerPlot function in the ArchR package (*Nature Genetics* 53, 403–411 (2021)), and the shapes of volcano plot might be related to the datasets. For example, the volcano plot from the E13 mouse embryo data was centered properly (Please see below).

Referee #2 (Remarks to the Author):

The authors provide a revised manuscript with more high-resolution data, and additional clarification regarding details of technical validation. I continue to have concerns about the technical characterization of the data they are generating, which I describe below.

Regarding the technical validation: the authors do a feature profile analysis to compare the thickness of a thin layer of vascular cells in fluorescent images and spatial-ATAC images. What is left now to do is to better characterize the false positive noise. In many of these images, one can see "lines" of signal running vertically or horizontally across the image. The authors should explain the source of this noise, and what computational (or experimental) steps might be employed to mitigate it.

Response: Thanks for the comments. Regarding the vertical or horizontal lines, they are not necessarily artifacts but similar to bead-to-bead variation in single cell sequencing. Because our pixels are not completely random but connected in rows or columns, thus we see row-to-row or column-to-column intensity variation. However, using standard data normalization in single cell sequencing data processing, these "issues" are gone and do not affect clustering analysis. This effect was observed in our previous DBiT-seq protocol and has been clarified in our *Cell* paper (2020). However, in our spatial-ATAC-seq technology, we didn't see strong vertical or horizontal line effect.

Referee #3 (Remarks to the Author):

The authors have revised their manuscript adding data, analyses, and explanatory notes. They have competently addressed all of my concerns as well as those of the other reviewers. I congratulate the authors on an exciting new method and insightful analyses.

Response: We would like to thank the reviewer for the positive feedback regarding our manuscript and work!